# Scalable integrated two-dimensional Fourier-transform spectrometry

Hongnan Xu [1] ✉, Yue Qin[1], Gaolei Hu [1] & Hon Ki Tsang [1] ✉

Integrated spectrometers offer the advantages of small sizes and high portability, enabling new applications in industrial development and scientific research. Integrated Fourier-transform spectrometers (FTS) have the potential to realize a high signal-to-noise ratio but typically have a trade-off between the resolution and bandwidth. Here, we propose and demonstrate the concept of the two-dimensional FTS (2D-FTS) to circumvent the trade-off and improve scalability. The core idea is to utilize 2D Fourier transform instead of 1D Fourier transform to rebuild spectra. By combining a tunable FTS and a spatial heterodyne spectrometer, the interferogram becomes a 2D pattern with variations of heating power and arm lengths. All wavelengths are mapped to a cluster of spots in the 2D Fourier map beyond the free-spectral-range limit. At the Rayleigh criterion, the demonstrated resolution is 250 pm over a 200-nm bandwidth. The resolution can be enhanced to 125 pm using the computational method.

Optical spectrometers are widely used in chemical sensing[1], medical analysis[2], astronomical research[3], and optical coherence tomography[4]. Fourier-transform spectrometers (FTS) sample an unknown spectrum with sinusoidal responses at varying periods and reconstruct it in the Fourier domain[5]. FTSs have two principal advantages, compared to the schemes based on diffraction gratings[6,7] or tunable filters[8,9]. First, the spectral information at all wavelengths is simultaneously captured, resulting in a noise reduction proportional to the square root of the number of channels (known as Fellgett's advantage[10]). Second, since light is received by a single photodetector (PD), the etendue is not restricted by the slit size or filter linewidth (known as Jacquinot's advantage[11]). Consequently, FTSs typically exhibit a higher signal-to-noise ratio (SNR). Moreover, unlike speckle spectrometers, which rely on random speckle patterns[12,13], the sinusoidal fringes of FTSs are inherently orthogonal, ensuring an accurate reconstruction.

In conventional benchtop FTSs, the optical path length (OPL) difference is tuned by moving the mirror reflector in a free-space Michelson interferometer (MI), making them sensitive to mechanical vibrations and unsuitable for field deployment. The chip-scale monolithic integration of FTSs improves their robustness and portability. Integrated FTSs are commonly built with planar nanophotonic circuits on various materials, such as silicon, silicon nitride, and lithium niobate. The reported integrated FTSs can be classified into four approaches: spatial heterodyne spectrometers (SHS)[14–24], tunable FTSs (tFTS)[25–27], digital FTSs (dFTS)[28,29], and stationary-wave integrated FTSs (SWIFTS)[30–32]. All these FTS schemes suffer from an inherent trade-off between the resolution and bandwidth (BW) due to the difficulty in achieving a large group-delay variation in a nanophotonic waveguide. The SHS is an assembly of Mach-Zehnder interferometers (MZI) with different arm-length asymmetries[14–24]. It is feasible to attain a fine resolution in a SHS using long delay lines. However, the limited number of monolithically integrated MZIs hinders the scalability of the channel capacity of SHSs. Moreover, SHSs employ multiple physical channels, which reduces the etendue at each port and diminishes the Jacquinot's advantage. For the tFTS, the group delay is scanned by varying the heating power applied to the tunable delay line[25–27]. Due to the continuous nature of thermo-optical (TO) tuning, the number of sweep steps can be scaled up, enabling a large capacity and a broad BW. The primary drawback of tFTSs is the poor resolution that results from the limited TO tuning range. Typically, a power consumption of ≈ 5 W is required to achieve a resolution at the nanometer scale[26]. The dFTS, which uses digitally switchable delay lines in a single MZI, has the potential to realize a fine resolution and a broad BW while preserving the Fellgett's and Jacquinot's advantages. However, scaling to a larger

---

[1]Department of Electronic Engineering, The Chinese University of Hong Kong, Shatin, New Territories, Hong Kong SAR, China.
✉e-mail: hongnanxu@cuhk.edu.hk; hktsang@ee.cuhk.edu.hk

 1

switch capacity makes it increasingly difficult to balance the intensities at two arms and maintain a high extinction ratio. To date, the demonstrated number of switch states is limited to 127[29]. The concept of SWIFTSs is to retrieve a spectrum from the dispersive field of a stationary wave[30–33], according to the Lippmann's principle. However, it remains challenging for a SWIFT to probe the field distribution of a guided mode even with embedded PDs[33]. In ref. 34, a microring resonator (MRR) is utilized to enhance the resolution of the tFTS. However, the MRR-assisted tFTS only captures few wavelengths at each sampling, resulting in a low signal-to-noise ratio of SNR ≈ 10 dB due to the absence of Fellgett's and Jacquinot's advantages. In ref. 35, the SHS is combined with a speckle spectrometer to attain a picometer-scale resolution. In this scheme, however, the speckles are produced in a bulky substrate rather than in the thin-film waveguide, and the fine resolution is mainly supported by speckle spectrometry. In ref. 36, a free-space FTS is combined with a Pelin-Broca prism to disperse the interferogram and expand the BW. However, such a scheme requires a two-dimensional imager to capture the dispersed patterns and is difficult to implement on integrated circuits. Overall, the realization of a monolithically integrated FTS with a fine resolution and a broad BW remains a challenge.

Here, the concept of two-dimensional Fourier-transform spectrometry (2D-FTS) is proposed and demonstrated. The core idea is to use 2D Fourier transform[37] instead of 1D Fourier transform to resolve any spectrum beyond the resolution-bandwidth limit. The structure combines a coarse-resolution, broadband tFTS and a fine-resolution, narrow-band SHS. The interferogram becomes a 2D pattern with varying heating power and arm lengths. The 2D Fourier transform of the interferogram contains a cluster of spots, each of which carries the information at a specific wavelength. The reconstruction is implemented beyond a single free spectral range while maintaining the Fellgett's advantage. The resolution demonstrated at the Rayleigh criterion is 250 pm with a large capacity of 801 channels. The resolution and capacity can be improved to 125 pm and 1601, respectively, using the computational method. These results represent, to the best of our knowledge, the largest channel capacity ever demonstrated in integrated FTSs. The 2D-FTS can be extended to a higher dimension with greater scalability.

This article is structured into four sections: the concept of the 2D-FTS, the design of crucial components, the characterization of the device, and the measurement of spectra. We will focus on the concept, mechanism, and realization of the 2D-FTS. The computational details are covered in Supplementary information.

## Results
### Design principle

The 2D-FTS is a combination of a tFTS and a SHS, as schematically displayed in Fig. 1a. The tFTS and SHS are interfaced by a 1 × 128 power splitter (PS). The broadband edge couplers serve as input and output (IO). The device has two input ports ($IN_{1-2}$) and 129 output ports ($OUT_{0-128}$). Light is injected at $IN_1$ and collected at $OUT_{1-128}$, while $IN_2$ and $OUT_0$ are utilized for monitoring the tFTS. The tFTS is formed by two 2 × 2 adiabatic directional couplers (ADC) and two spiral tunable delay lines. The use of a 2 × 2 coupler allows for additional channels that are connected to $IN_2$ and $OUT_0$. In the SHS, 128 MZIs with different arm-length asymmetries are arranged as a 16×8 array. Each MZI comprises of two Y-branch splitters (YBS) and two folded delay lines. The $i$-th MZI has an arm-length difference of $\Delta L_{SHS,i} = i \cdot \Delta L_{SHS,1}$ and is routed to $OUT_i$. The PS is a seven-layer binary tree formed by YBSs. The input of the PS is connected to the tFTS, while its output is connected to the MZI array.

In Fig. 1b, c, the concepts of 1D- and 2D-FTSs are compared. For the conventional 1D-FTS, the structure is a stand-alone tFTS or SHS. The response of a 1D-FTS can be described by a 2D matrix (denoted as **A**). Given a spike spectrum (denoted as **S**) at a single wavelength ($\lambda$),

the output interferogram (denoted as **O**) is a 1D sinusoidal sequence as a function of heating power ($P$) or $\Delta L_{SHS}$. The 1D fast Fourier transform (FFT) of the interferogram contains a single spike in each quadrant (I - II). Hence, by using 1D discrete cosine transform (DCT), **S** can be reverted from **O**. Nevertheless, 1D-FTSs are limited by the trade-off between the resolution and BW: tFTSs have a broad BW but a coarse resolution, whereas SHSs have a fine resolution but a narrow BW, as discussed in Introduction. For the 2D-FTS, signals are recorded at multiple physical ports of the SHS and modulated simultaneously by the tFTS with varying $P$. Thus, the response of the 2D-FTS can be depicted by a 3D cube with variations of $P$ and $\Delta L_{SHS}$. At a single wavelength, the interferogram becomes a pattern that is sinusoidally modulated in two dimensions. By applying 2D-FFT, a spike in the spectrum is mapped to a spot in each quadrant (I - IV) of FFT(**O**). Figure 1d shows the reconstruction process. The cube can be sliced into a series of fringe patterns (denoted as $a_i$) at varying wavelengths. In the Fourier domain, each fringe is related to a spot at distinct Fourier frequencies ($f_{tFTS}$ and $f_{SHS}$), as discussed in Fig. 1c. Here, $f_{tFTS}$ and $f_{SHS}$ are normalized to ±1/2. The recorded interferogram (**O**) is a linear combination of fringes ($a_i$), with the weight on $a_i$ indicating the spectral intensity at the $i$-th wavelength. Therefore, when a continuous spectrum is launched, the corresponding FFT(**O**) will have a cluster of spots in each quadrant. At varying wavelengths, the spot location shifts "slowly" along $f_{tFTS}$ and "fast" along $f_{SHS}$ since the free spectral range is broad for the tFTS but narrow for the SHS. The shift direction along $f_{tFTS}$ depends on the initial phase of the sinusoidal response of the tFTS at the first sweep step. In this work, the spot on the blue end shifts towards $f_{tFTS} = 0$. The spots are divided into several segments between $f_{SHS} = 0$ and 1/2. Each folded segment contains the spectral information within a single free spectral range ($FSR_{SHS}$) of the SHS. Without the tFTS, all spots will overlap into a single segment. The segments are shifted to distinct $f_{tFTS}$ through tFTS modulation, making it possible to expand the bandwidth beyond a single $FSR_{SHS}$. To identify adjacent $FSR_{SHS}$, the tFTS must have a resolution ($\delta\lambda_{tFTS}$) finer than $FSR_{SHS}$, which yields:

$$\Delta\lambda_{tFTS} = \frac{\lambda^2}{\Delta n_g L_{tFTS}} < FSR_{SHS} = \frac{N_{SHS}\Delta\lambda_{SHS}}{2}, \quad (1)$$

where $\Delta n_g$ denotes the group-index variation induced by TO tuning, $L_{tFTS}$ denotes the length of the tunable delay line, $N_{SHS}$ (=128) denotes the number of MZIs in the SHS, and $\delta\lambda_{SHS}$ denotes the resolution of the SHS. An unknown spectrum can be rebuilt from FFT(**O**) via computational decomposition if the critical condition is met. According to the Rayleigh criterion[38], the resolution ($\delta\lambda_f$) of the 2D-FTS is $\delta\lambda_{SHS}$, while its BW is the free spectral range ($FSR_{tFTS}$) of the tFTS. $\delta\lambda_f$ and BW can be thus formulated as:

$$\delta\lambda_f = \delta\lambda_{SHS} = \frac{\lambda^2}{n_g\Delta L_{SHS,max}}, \quad (2)$$

$$BW = FSR_{tFTS} = \frac{N_{tFTS}\Delta\lambda_{tFTS}}{2}, \quad (3)$$

where $n_g$ denotes the group index, $\Delta L_{SHS,max}$ (=$\Delta L_{SHS,128}$) denotes the maximum arm-length difference in the SHS, and $N_{tFTS}$ denotes the number of power sweep steps. The corresponding channel capacity ($N_f$) is defined as $N_f = BW/\delta\lambda_f + 1$. According to Eq. 3, to obtain a point-to-point mapping (see Fig. 1d), the required number of power sweep steps is $2 \cdot BW/\delta\lambda_{tFTS}$. Notably, the 2D FTS has the potential to reduce sweep steps to $<2 \cdot BW/\delta\lambda_{tFTS}$ using the numerical method since, unlike 1D-FTSs, the folding of a Fourier map only leads to a limited increase in correlation between channels. This issue will be discussed later.

We compare the performance of reported FTSs, as shown in Fig. 1e. It can be found that the proposed 2D-FTS has a record large

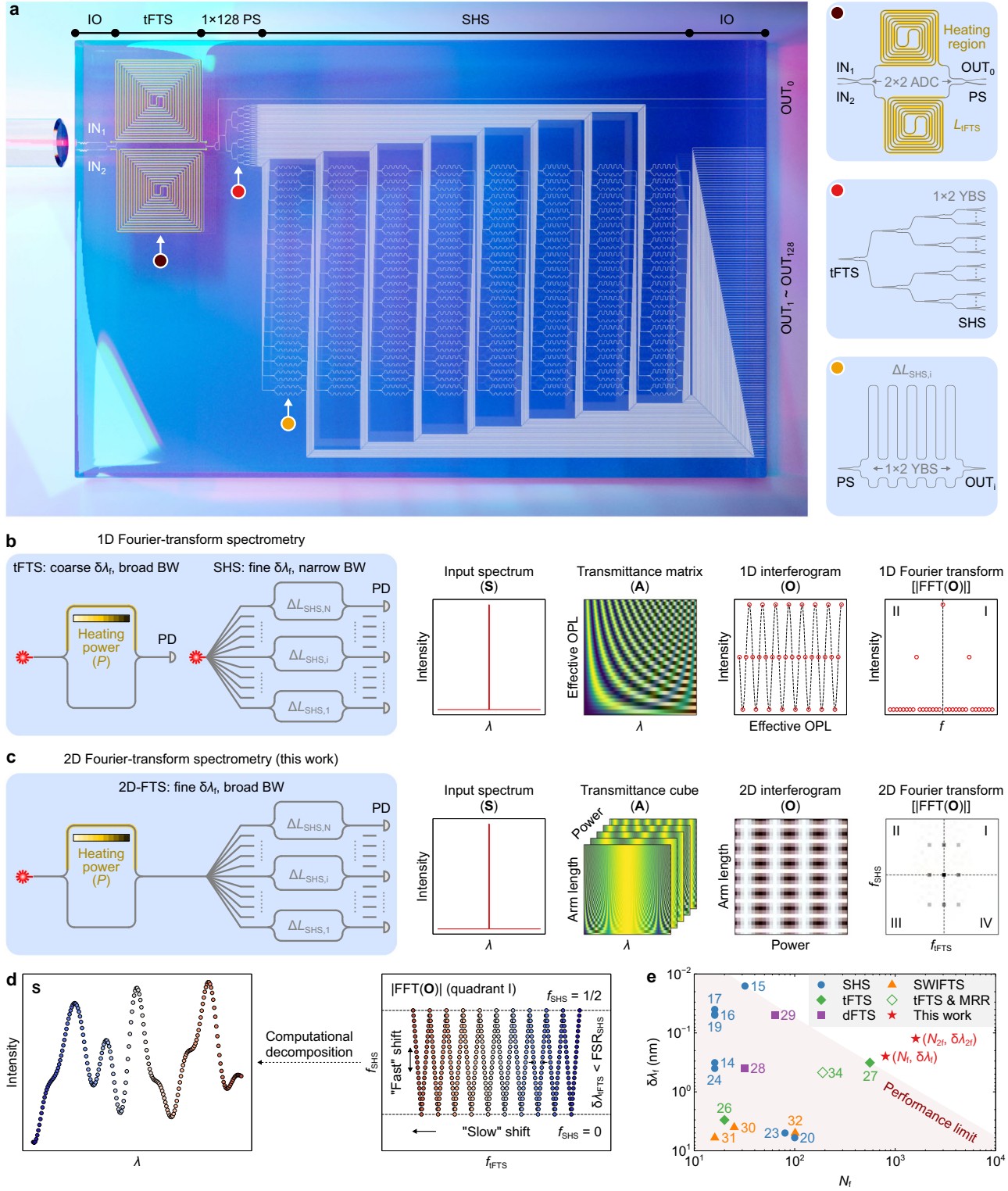

capacity of $N_f = 801$ while retaining a fine resolution of $\delta\lambda_f = 250$ pm. In addition, by leveraging the computational method, the resolution and capacity can be further enhanced to $\delta\lambda_{2f} = \delta\lambda_f/2 = 125$ pm and $N_{2f} = 2N_f = 1601$, respectively, beyond the Rayleigh criterion, which will be discussed later. A more detailed comparison can be found in Supplementary information, Note 1. In this work, all structures were fabricated at a commercial photonic foundry (Applied Nanotools[39]). The simulation and measurement methods are described in Methods. The abbreviations and notations used in this article are summarized in Supplementary information, Note 2.

## Design and characterization of key components

The 2D-FTS has three essential elements: the delay line, ADC, and YBS. The design is implemented on the silicon-on-insulator (SOI) platform with a core thickness of $H_{wg} = 220$ nm, as shown in Fig. 2a. The core width is set as $W_{wg} = 450$ nm to meet the single-mode condition. A titanium-tungsten (TiW) heater is placed atop the SOI waveguide with a spacing of $d_{ht} = 1$ μm. The cross-section dimension of the heater is $W_{ht} \times H_{ht} = 7 \times 0.2$ μm². We utilized a fabricated tFTS with $L_{tFTS} = 1.5$ cm to characterize the TO tunability, as shown in Fig. 2b. Figure 2c shows the measured dispersion curves of transmittances ($|t|^2$) at varying drive

**Fig. 1 | Principle of the two-dimensional Fourier-transform spectrometer (2D-FTS). a** Schematic layout of the 2D-FTS. The insets show the enlarged views of key components. The 2D-FTS comprises of a tunable Fourier-transform spectrometer (tFTS) and a spatial heterodyne spectrometer (SHS) that are connected via a 1×128 power splitter (PS). The edge couplers are utilized as input and output (IO) ports, i.e., IN$_i$ and OUT$_i$. Conceptual illustrations of the **b** 1D and **c** 2D Fourier-transform spectrometry. For a stand-alone tFTS/SHS, the period of the sinusoidal response is tuned by changing the heating power ($P$) or arm-length difference ($\Delta L_{SHS}$). At a single wavelength ($\lambda$), the interferogram (**O**) is a 1D sequence sliced from a 2D transmittance matrix (**A**). The 1D fast Fourier transform (FFT) of the interferogram contains a peak in each quadrant. For the 2D-FTS, **A** is a 3D cube with variations of both $P$ and $\Delta L_{SHS}$. Hence, the interferogram is a 2D diagram that is modulated along two axes. By applying 2D FTT, each wavelength is mapped to a single spot in the

Fourier domain. **d** Reconstruction principle. The intensity information of a continuous spectrum (**S**) is encoded by a cluster of spots in FFT(**O**). At varying wavelengths, the spot location shifts "slowly" along $f_{tFTS}$ and "fast" along $f_{SHS}$. Here, $f_{tFTS}$ and $f_{SHS}$ denote the Fourier frequencies. The shift direction relies on the phases of sinusoidal responses. Any spectrum can be retrieved via decomposition, as long as the resolution ($\delta\lambda_{tFTS}$) of the tFTS is finer than the free spectral range (FSR$_{SHS}$) of the SHS, thereby breaking the inherent limit between the resolution ($\delta\lambda_f$) and bandwidth (BW). **e** Comparison of integrated FTSs in terms of the channel capacity ($N_f$) and $\delta\lambda_f$ at the Rayleigh criterion. $\delta\lambda_f$ and $N_f$ can be improved to $\delta\lambda_{2f}$ and $N_{2f}$ using computational methods. OPL, optical path length. ADC, adiabatic directional coupler. YBS, Y-branch splitter. PD, photodetector. MRR, microring resonator. dFTS, digital FTS. SWIFTS, stationary-wave integrated FTS.

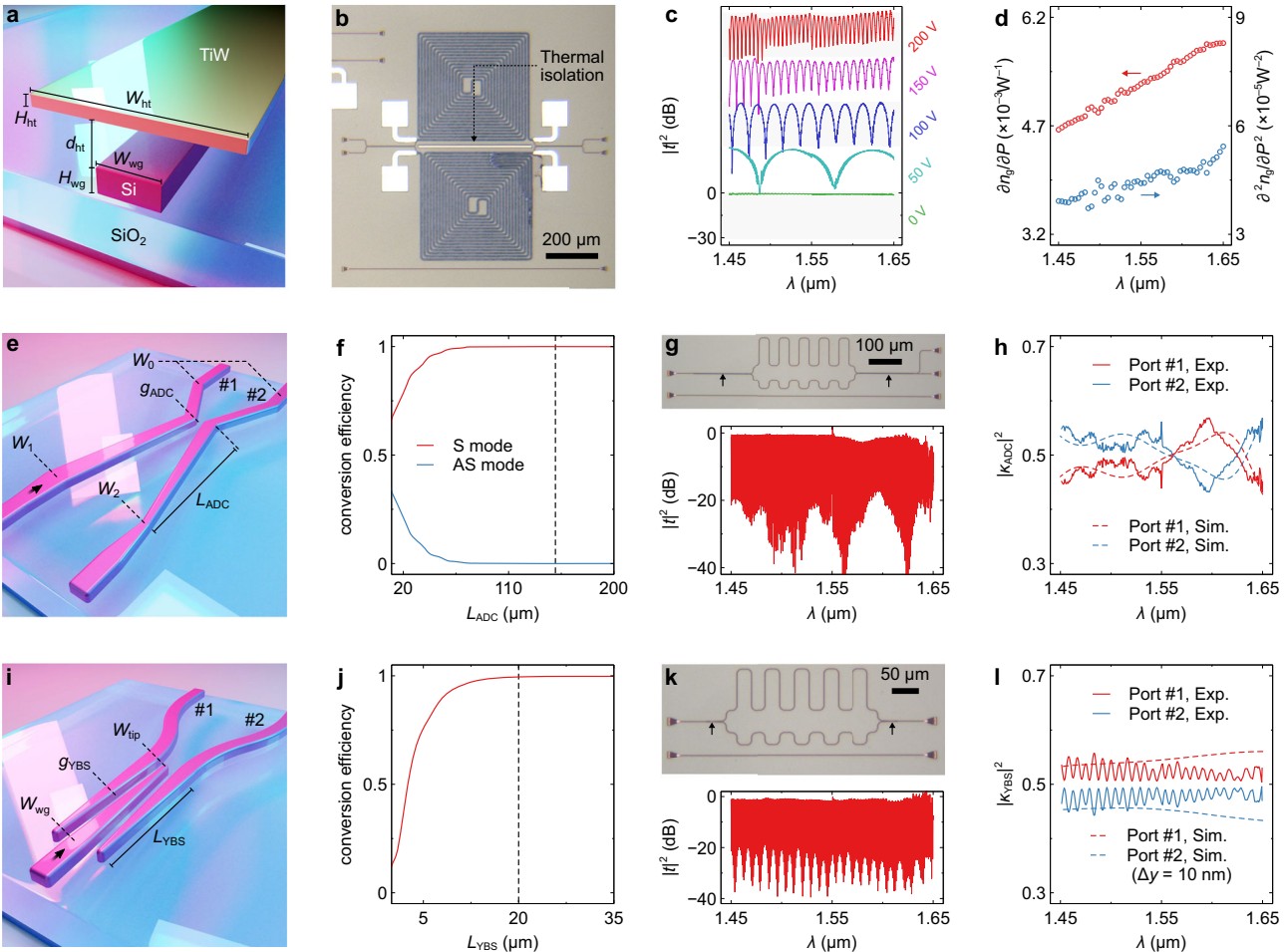

**Fig. 2 | Design and characterization of key components. a** 3D view of the tunable delay line. **b** Microscope image of the testing structure. **c** Measured transmittances ($|t|^2$) at varying wavelengths ($\lambda$) when different voltages were applied. **d** Measured first- and second-order tuning efficiencies ($\partial n_g/\partial P$, $\partial^2 n_g/\partial P^2$). **e** 3D view of the adiabatic directional coupler (ADC). **f** Calculated conversion efficiencies of symmetric (S) and anti-symmetric (AS) modes with varying coupling lengths ($L_{ADC}$). **g** Upper panel: microscope image of the fabricated testing structure. The arrows

indicate the locations of ADCs. Lower panel: Measured $|t|^2$ at varying $\lambda$. **h** Calculated and measured coupling ratios ($|\kappa_{ADC}|^2$) of the ADC. **i** 3D view of the Y-branch splitter (YBS). **j** Calculated conversion efficiencies with varying coupling lengths ($L_{YBS}$). **k** Upper panel: microscope image of the fabricated testing structure. The arrows indicate the locations of YBSs. Lower panel: Measured $|t|^2$ at varying $\lambda$. **l** Calculated and measured coupling ratios ($|\kappa_{YBS}|^2$) of the YBS. In the calculation, the central core is shifted by $\Delta y = 10$ nm.

voltages. TO nonlinearity will arise at a Watt-scale heating power, resulting in higher-order terms in $\Delta n_g$, which can be expressed as:

$$\Delta n_g = \sum_i \frac{\partial^i n_g}{\partial P^i} \cdot P^i. \tag{4}$$

Figure 2d shows the first- and second-order tuning efficiencies ($\partial n_g/\partial P$ and $\partial^2 n_g/\partial P^2$) derived at varying wavelengths. Additional

simulation and measurement results, e.g., the electric response, index dispersions, temperature sensitivity, and propagation losses, can be found in Supplementary information, Note 3. The ADC consists of two cores with varying widths that gradually approach each other[40] (see Fig. 2e). Incident light will excite symmetric (S) and asymmetric (AS) modes in the coupling region. The conversion efficiency of the S mode will reach ≈ 1 when the adiabatic condition is fulfilled. In Fig. 2f, we calculate the conversion efficiencies with varying coupling lengths ($L_{ADC}$). From the curve, the coupling length is optimized to be

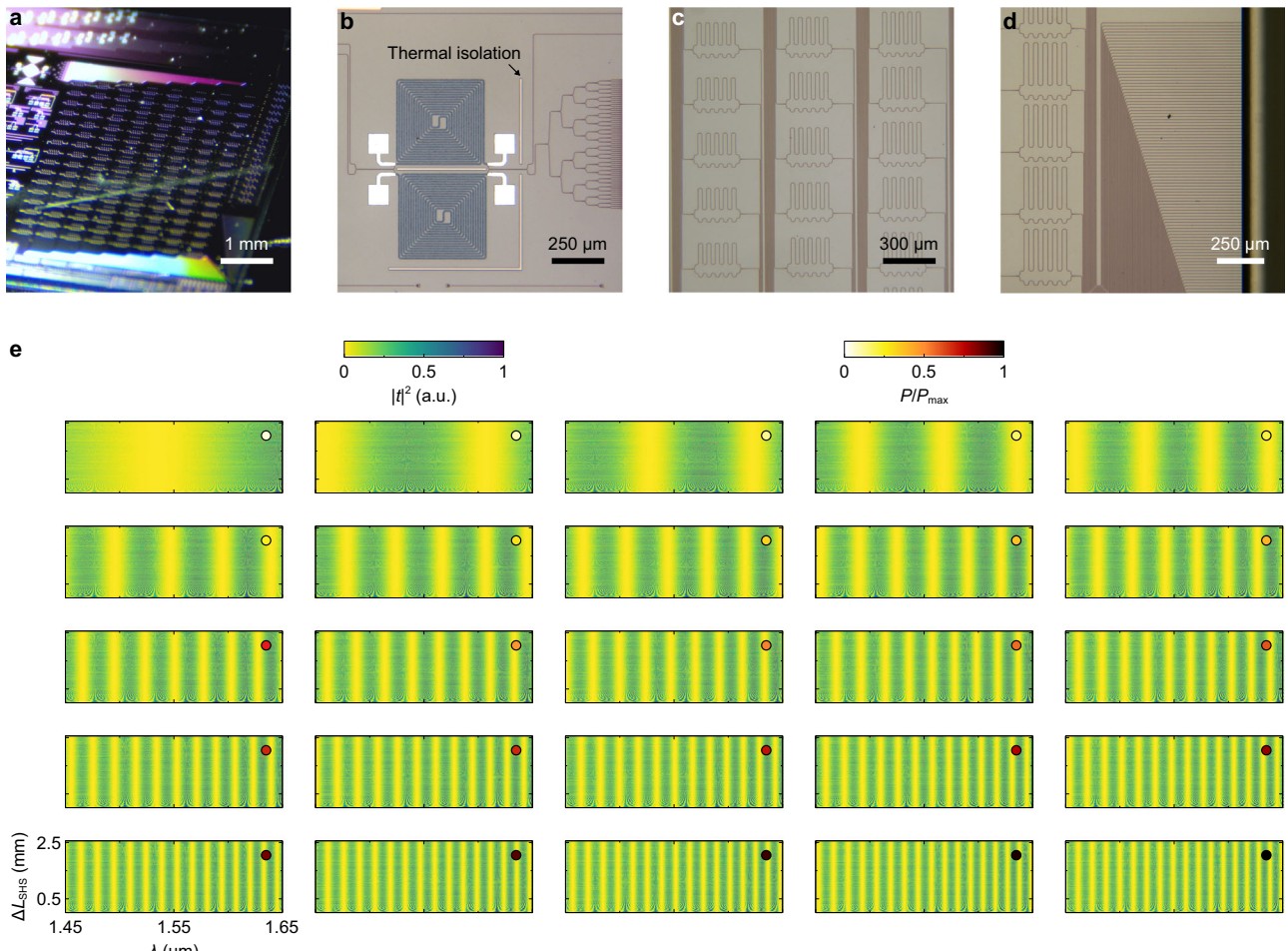

**Fig. 3 | Characterization of the spectrometer.** Microscope images of the fabricated **a** photonic chip, **b** tunable Fourier-transform spectrometer, **c** spatial heterodyne spectrometer, and **d** inverse-taper edge couplers. **e** Measured transmittance ($|t|^2$) cube. The cube is sliced into the matrices with varying heating power ($P$), represented by the colors of dots in the upper right corner. Each matrix contains transmittances with varying arm-length differences ($\Delta L_{SHS}$) and wavelengths ($\lambda$).

$L_{ADC} = 150$ μm, in order to obtain sufficient adiabaticity. Other less critical structural parameters can be found in Supplementary information, Note 4. For testing purposes, we fabricated a MZI using two ADCs as couplers, as shown in the upper panel of Fig. 2g. The measured $|t|^2$ dispersion is shown in the lower panel of Fig. 2g. The coupling ratios ($|\kappa_{ADC}|^2$) can be extracted from the extinction ratios of the curve, as shown in Fig. 2h. The calculated $|\kappa_{ADC}|^2$ are also plotted for comparison. Both simulation and measurement results suggest that the ADC has coupling ratios of $|\kappa_{ADC}|^2 \approx 0.5$ with weak variations over the target bandwidth of BW = 200 nm (see Supplementary information, Note 4 for additional analysis). The YBS is a three-core structure[41] (see Fig. 2i). The widths of the central core and lateral cores vary in-complement over the coupling region. Consequently, the light power in the central core will evenly transfer to two lateral cores through evanescent coupling. Figure 2j shows the calculated conversion efficiency as a function of the coupling length ($L_{YBS}$). The coupling length is chosen as $L_{YBS} = 20$ μm to ensure a complete coupling (see Supplementary information, Note 5 for other parameters). A similar testing MZI was fabricated to measure the coupling ratio ($|\kappa_{YBS}|^2$) of the YBS, as shown in Fig. 2k. The calculated and measured $|\kappa_{YBS}|^2$ dispersion curves are shown in Fig. 2l. In the simulation, the central core is laterally shifted by $\Delta y = 10$ nm. The 3-dB coupling is achieved across a broad wavelength band from $\lambda = 1.45$ μm to 1.65 μm. Additional results, such as the tolerance analysis, can be found in Supplementary information, Note 5.

## Characterization and analysis of the spectrometer

The maximum arm-length difference of the SHS is set as $\Delta L_{SHS,max} = 2.55$ mm to achieve the resolution of $\delta\lambda_f = 250$ pm using Eq. 2 and the results shown in Supplementary information, Fig. S1d. According to Eq. 2, given a fixed $\Delta L_{SHS,max}$, $\delta\lambda_f$ will increase at a longer wavelength. Thus, $\Delta L_{SHS,max}$ is derived at $\lambda = 1.65$ μm to ensure $\delta\lambda_f < 250$ pm over the whole wavelength range. Using Eq. 1, the maximum heating power is set as $P_{max} = 2.4$ W to identify all free spectral ranges of the SHS. The resolution of the tFTS reaches its minimum ($\delta\lambda_{tFTS} \approx 12.36$ nm) at $\lambda = 1.45$ μm, which yields the number of power sweep steps of $N_{tFTS} = 32$. Nevertheless, due to the dislocation of spot trajectories in a folded 2D Fourier map, the sweep steps can be further reduced without compromising much reconstruction accuracy. In this work, the optimal number of sweep steps is set as $N_{tFTS} = 25$ (see Supplementary information, Note 6). The complete optimization flow is discussed in Supplementary information, Fig. S5. Figure 3a–d shows the microscope image of the fabricated 2D-FTS. The TE-pass polarizers were integrated at $IN_{1-2}$ to support a high polarization extinction ratio of 40 dB[42]. Each edge coupler is an inverse taper with an effective spot diameter of ≈ 3 μm. The spacing between output ports is chosen as >15 μm to prevent inter-channel optical crosstalk. The thermal-isolation trenches were used to inhibit the thermal crosstalk between tunable delay lines and the heat transfer from the tFTS to the SHS.

The measured transmittance cube is shown in Fig. 3e. The cube is sliced into matrices with varying $P$, as depicted by the colored dots in

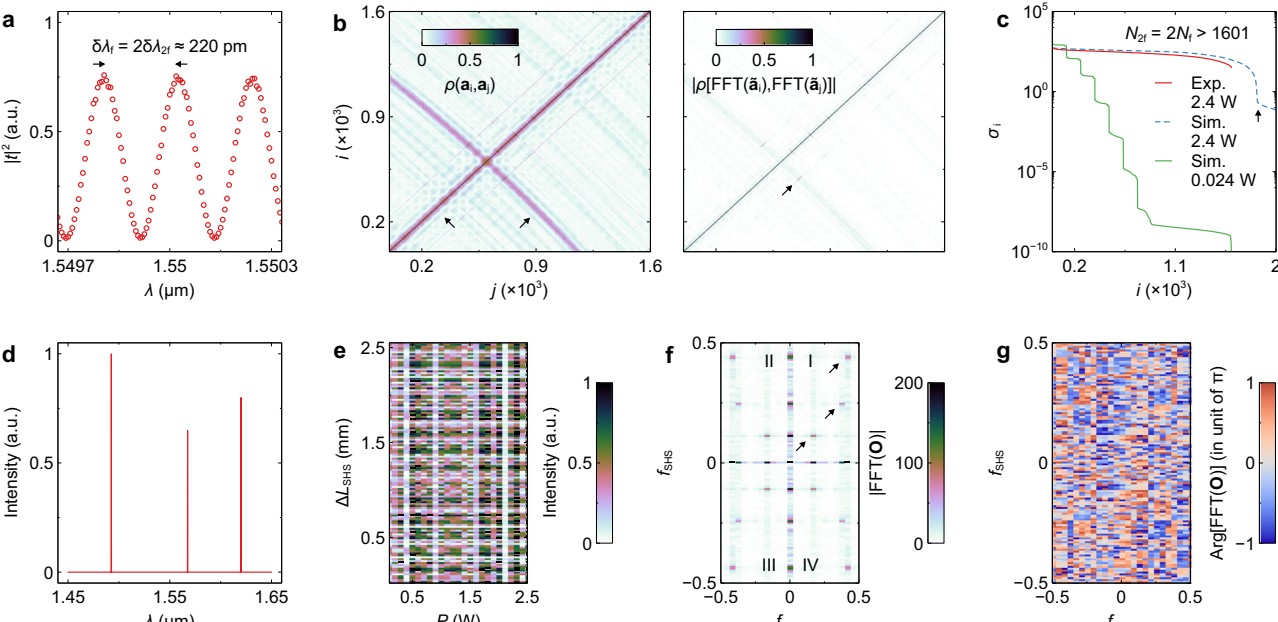

**Fig. 4 | Analysis of the spectrometer. a** Measured transmittance ($|t|^2$) at $OUT_{128}$ and varying wavelengths ($\lambda$). Here, zero electric power was applied to the heater. At the Rayleigh criterion, the resolution is $\delta\lambda_f = 2\delta\lambda_{2f} \approx 220$ pm at $\lambda \approx 1.55\,\mu m$. **b** Correlation matrices derived from the transmittance cube. On the left panel, the correlation [$\rho(\cdot, \cdot)$] is performed between the fringes ($\mathbf{a}_i$) at different $\lambda$. The arrows highlight the high-correlation non-diagonal elements. The right panel shows the correlation of the fast Fourier transform (FFT) of fringes ($\tilde{\mathbf{a}}_i$) with zero-frequency components removed. The arrow highlights the remnant non-diagonal elements with relatively high correlations. **c** Singular values ($\sigma_i$) derived from the calculated and measured cube with the heating power of 2.4 W and 0.024 W. The arrow highlights the kink. The capacity of $N_{2f} = 2N_f = 1601$ is verified. **d** Testing spectrum (**S**) with three spikes. **e** Interferogram (**O**) derived from the measured cube and testing spectrum. **f** Intensity and **g** phase maps of FFT(**O**). The arrows highlight the three spots that are associated with the three spikes.

the upper right corner. For clarity, the displayed cube is normalized to its maximum. In the spectrum reconstruction, the excess losses of the PS, polarizer, and edge couplers will not be deducted from the cube (see Supplementary information, Fig. S1h). The simulation results can be found in Supplementary information, Fig. S6a. The cube is first spectrally channelized into $N_{2f} = 2N_f = 1601$ columns, with a spectral grid size of $\delta\lambda_{2f} = \delta\lambda_f/2 = 125$ pm. $\delta\lambda_f$ and $N_f$ are defined based on the Rayleigh criterion and are related to the fundamental frequency in sampling offered by the FTS response. However, a sinusoidal response must be sampled at more than twice the fundamental frequency; otherwise, the spectral sampling will be ineffective at the highest frequency (i.e., $\Delta L_{SHS} = \Delta L_{SHS,max}$, see Supplementary information, Fig. S7a), as suggested by the Nyquist's theorem. Moreover, it is viable to retrieve the spectral information carried by all $N_{2f}$ grids when the computational method is used, as will be discussed later. The similar effect has been reported in ref. 28. Figure 4a shows the response measured at $OUT_{128}$ and $P = 0$ W. At the Rayleigh criterion, a fine resolution of $\delta\lambda_f = 2\delta\lambda_{2f} \approx 220$ pm is achieved at $\lambda \approx 1.55\,\mu m$. The measured resolution is $\delta\lambda_f = 2\delta\lambda_{2f} < 250$ pm on the red end ($\lambda \approx 1.65\,\mu m$). To implement further analysis, the cube is reshaped into a 2D matrix, in which each column is a vector flattened from the fringe ($\mathbf{a}_i$) at a single wavelength, as detailed in Supplementary information, Note 6. The left panel of Fig. 4b shows the correlation matrix derived from the flattened cube. Here, the element at the $i$-th row and $j$-th column is the Pearson correlation between the $i$-th and $j$-th fringes, i.e., $\rho(\mathbf{a}_i, \mathbf{a}_j)$. Except for the self-correlated diagonal elements, the correlation matrix also contains some "shades" with insufficient decorrelation. Such a phenomenon results from the projection effect of 2D-FFT. As discussed, at a single wavelength, FFT($\mathbf{a}_i$) is mapped to a spot in each quadrant. However, the spot has projections on two axes ($f_{SHS} = 0$ and $f_{tFTS} = 0$, see Fig. 1c). As a consequence, two fringes with distinct spot locations in the Fourier domain may still have similarity since the spot projections may overlap (see Supplementary information, Fig. S7c). This issue can be addressed by omitting the zero-frequency

components of $\mathbf{a}_i$ and **O**, provided that FFT is a linear transform. More details are shown in Supplementary information, Fig. S10. The right panel of Fig. 4b shows the correlation result of FFT($\tilde{\mathbf{a}}_i$). Here, $\tilde{\mathbf{a}}_i$ is the fringe after the removal of the zero-frequency components. The correlation matrix becomes quasi-diagonal, indicating the establishment of decorrelation. Some elements in the correlation matrix still have relatively high values, which results from the crossover of spot trajectories in the folded Fourier map. Throughout the entire capacity, there are only $\approx 5$ ($<N_{tFTS}/2$) pairs of less decorrelated fringes. The residual correlation can be eliminated by increasing the power sweep steps to $> 2 \cdot BW/\delta\lambda_{tFTS}$ and unfolding the Fourier map; however, this will result in an increase in the acquisition period. By using the numerical method, it is viable to identify these fringes without increasing $N_{tFTS}$ or reconstruction errors, provided that the Fourier map is folded only once (i.e., $N_{tFTS} > BW/\delta\lambda_{tFTS}$) and their correlations [$\rho(\tilde{\mathbf{a}}_i, \tilde{\mathbf{a}}_j) \approx 0.5$] are still quite limited. Further discussions can be found in Supplementary Information, Fig. S9. The effectiveness of this operation can be verified using singular value decomposition (SVD). In Fig. 4c, we show the singular values ($\sigma_i$) derived from the calculated and measured cube with the zero-frequency components omitted. When the cube is oversampled into $> 3000$ channels, a kink can be found at $i \approx 1900$ exceeding $N_{2f}$ (= 1601). With a lower heating power of $P_{max} = 0.024$ W, the $\sigma_i$ curve drops rapidly and is segmented by the kinks located at each $FSR_{SHS}$, indicating that Eq. 1 is not satisfied, and decorrelation is insufficient. For the measurement result with $P_{max} = 2.4$ W, the $\sigma_i$ curve is smooth and flat at $N_{2f} = 2N_f = 1601$. Thus, it is conclusively demonstrated that the 2D-FTS has sufficient decorrelation over all fringes. The Fourier analysis is performed on the measured cube. Figure 4d shows a generated testing spectrum with three spikes. The interferogram is then derived from the cube and testing spectrum, as shown in Fig. 4e. The intensity and phase distributions of FFT(**O**) are shown in Fig. 4f, g. Three distinctive spots can be clearly observed in |FFT(**O**)|, validating the effectiveness of the 2D-FTS. The simulation results show that the phase distribution of FFT(**O**)

has an abrupt discontinuity at the spot location (see Supplementary information, Fig. S10b and d, for instance), indicating that the phase map also carries information. Such a phase hopping is less visible in the experimental results (see Fig. 4g) since the environmental perturbation during measurement imposes a chaotic background on the phase map. Nevertheless, the impact of noise components can be mitigated using the iterative optimization method. Both intensities and phases of FFT(**O**) are necessary in the spectrum reconstruction to support the full capacity of $2N_f$, as discussed in Supplementary Information, Fig. S8f.

## Spectrum reconstruction

The measured **A** and recorded **O** both contain noises. In our design, temperature fluctuations dominate as the major source of noises due to the high TO coefficient of silicon and long OPL. The temperature sensitivity is analyzed in Supplementary information, Note 7. Moreover, the fringe patterns are not ideally sinusoidal since $n_g$ and coupling ratios are dispersive. Therefore, the spectrum recovered directly with DCT has large errors (see Supplementary Information, Fig. S12). The pseudo-inverse method also suffers from a poor reconstruction accuracy (see Supplementary Information, Fig. S13). Instead, we use the computational decomposition to rebuild a spectrum:

$$\mathbf{S} = \underset{\mathbf{S}}{\operatorname{argmin}}\left( \|\tilde{\mathbf{A}}_{\mathrm{FFT}}\mathbf{S} - \mathrm{FFT}(\tilde{\mathbf{O}})\|_2^2 + \Omega \right), \qquad (5)$$

where $\tilde{\mathbf{A}}_{\mathrm{FFT}}$ denotes the matrix formed by the column vectors of FFT($\tilde{\mathbf{a}}_i$), $\tilde{\mathbf{O}}$ denotes the output interferogram with zero-frequency components removed, $\Omega$ denotes the regularization term, argmin(·) denotes the global minimum, and $\|\cdot\|_2$ denote the $\ell_2$-norm. Using Eq. 5, FFT($\tilde{\mathbf{O}}$) is decomposed into a linear combination of FFT($\tilde{\mathbf{a}}_i$). The optimal weight on FFT($\tilde{\mathbf{a}}_i$) is the intensity at the $i$-th wavelength of the retrieved spectrum. The decomposition must be operated in the Fourier domain due to the necessity of component removal. $\Omega$ provides both Tikhonov[43] and total variation (TV[44]) regularization, which covers most spectral features. The hyperparameters in $\Omega$ are automatically optimized through cross validation (CV[45]) without manual selection. Remarkably, $\Omega$ only sets a general range of features that may occur in a spectrum, and no specific knowledge of spectral contents are required. We also use the Picard plot[46] to evaluate the solvability of Eq. 5 (see Supplementary information, Fig. S14). The feasibility of reconstructing a spectrum of arbitrary shape is further discussed in Supplementary Information, Fig. S16. In Supplementary Information, Fig. S18, the reconstruction accuracies based on DCT, pseudo inverse, and regularized iterative optimization are compared. The reconstruction method is described in greater detail in Supplementary information, Note 8.

Figure 5 shows the experimental results of spectrum reconstruction. The reference and reconstructed spectra are displayed in red and blue, respectively. The reference spectra were measured using a commercial optical spectrum analyzer (OSA). The reconstruction accuracy is quantified by the relative error[47] ($\varepsilon$) and coefficient of determination[24] ($r^2$). In Fig. 5a, we show the reconstruction of a single spectral line tuned from $\lambda = 1.45\,\mu\mathrm{m}$ to $1.65\,\mu\mathrm{m}$. Here, the spectral line was produced using a tunable laser (TL). A high accuracy of $r^2 > 0.99$ is realized at varying $\lambda$, demonstrating the working bandwidth of BW = 200 nm. A second TL was utilized to produce dual spectral lines with varying spacings, as shown in Fig. 5b. Only a small part of the spectrum is displayed for clarity, but the reconstruction is performed over the entire bandwidth. Even when spaced by only one or two grids, two peaks can still be distinguished. An enhanced resolution of $\delta\lambda_{2f} = \delta\lambda_f/2 < 125\,\mathrm{pm}$ is thus demonstrated, with the corresponding capacity of $N_{2f} = 2N_f = 1601$. In Fig. 5c, we give an example of weak-signal reconstruction. Two peaks with a contrast of 25 dB can be clearly identified from the noise floor at $< -35\,\mathrm{dB}$ (see the arrow). The response of a fiber

Bragg grating (FBG) is then reconstructed, as shown in Fig. 5d. At the rejection band, the retrieved spectrum has a high extinction ratio of $\approx$ 25 dB. Thus, the peak SNR of the fabricated 2D-FTS is characterized as PSNR > 25 dB. The reconstruction results of the responses of an arrayed waveguide grating (AWG) are shown in Fig. 5e and f. All spectral details are properly rebuilt, even for a subtle dip (see the arrow). An amplified spontaneous emission (ASE) source was used to produce a broadband spectrum, as shown in Fig. 5g. Furthermore, we also produced a hybrid spectrum by double injecting ASE and TL emissions (see Fig. 5h). The reconstruction results exhibit small errors ($\varepsilon < 0.02$). The recorded interferograms are shown in Supplementary information, Figs. S19 and S20. More numerical examples and experimental data can be found in Supplementary information, Notes 8 and 9, respectively.

## Discussion

We have proposed and demonstrated an integrated FTS beyond the resolution-bandwidth limit. In this work, the most significant advance is the transition of the working principle from 1D to 2D Fourier transform. For the 2D-FTS, the output interferogram is not a 1D sequence but rather a 2D pattern, which can be realized by combining a tFTS and a SHS. In the Fourier domain, the interferogram is transformed into the spots scattered in two dimensions, which can be decomposed into the linear combination of the FFT of independent fringes, enabling accurate spectrum reconstruction. If $\delta\lambda_{\mathrm{tFTS}}$ is finer than $\mathrm{FSR}_{\mathrm{SHS}}$, the bandwidth can exceed a single $\mathrm{FSR}_{\mathrm{SHS}}$ while maintaining a fine resolution of $\delta\lambda_{\mathrm{SHS}}$. At the Rayleigh criterion, the demonstrated resolution and capacity are $\delta\lambda_f = 250\,\mathrm{pm}$ and $N_f = 801$, respectively. Based on the computational method, the resolution is enhanced to $\delta\lambda_{2f} = 125\,\mathrm{pm}$ with a record large capacity of $N_{2f} = 1601$. The demonstrated bandwidth is BW = 200 nm. The 2D-FTS requires a maximum heating power of $P_{\max} = 2.4\,\mathrm{W}$ and 128 MZIs. By comparison, a conventional 1D-FTS requires $P_{\max} > 100\,\mathrm{W}$ or > 2000 MZIs to achieve the same $\delta\lambda_f$ and $N_f$. The experimental results indicate a signal-to-noise ratio of PSNR > 25 dB, exhibiting an improvement of $\approx$ 15 dB compared to the result reported in ref. 34. The rise-fall time of a single TO tuning step is < 100 μs[48], therefore it is feasible to drive the heater at a high speed (> 1 kHz). Given a small number of sweep steps ($N_{\mathrm{tFTS}} = 25$), the theoretical sampling period is < 0.025 s. In the measurement of a single spectrum, the corresponding energy budget for heating is thus estimated to be 60 mJ. As a proof of concept, we utilize the tFTS and SHS to build the 2D-FTS. Such a scheme has two drawbacks, but they are readily rectifiable. First, the required electric power exceeds one Watt. The TO tuning efficiency can be doubled using a MI instead of a MZI in the tFTS, taking advantage of its folded light path in the reflective interference arm[27]. In addition, by etching thermal-isolation trenches alongside tunable delay lines, it is possible to further reduce the heating power in the tFTS. Second, the SHS is multi-apertured, which retains the Fellgett's advantage but partially diminishes the Jacquinot's advantage. To solve this, we propose to replace the SHS with a dFTS that uses switchable delay lines to change the OPL difference. Thus, launched light will go through a single MZI and be received by a single photodetector (PD). The use of the dFTS also ensures a smaller footprint. A detailed design example of the modified scheme be found in Supplementary information, Note 10, Section A. Moreover, it is viable to extend the concept of 2D-FTS to a higher dimension. In Supplementary information, Note 10, Section B, it is demonstrated that, based on a three-dimensional FTS (3D-FTS), an finer resolution of $\delta\lambda_{2f} < 31.25\,\mathrm{pm}$ and a larger capacity of $N_{2f} > 6401$ are attainable. The concept and fabric of the higher-dimensional FTS (HD-FTS) are discussed in Supplementary information, Note 10, Section C. It should be noted that other schemes, such as filters and speckle spectrometers, can achieve comparable resolutions and bandwidths, but FTSs have the potential for a higher SNR and lower noises (see Supplementary information, Note 11 for further discussions). Overall, the

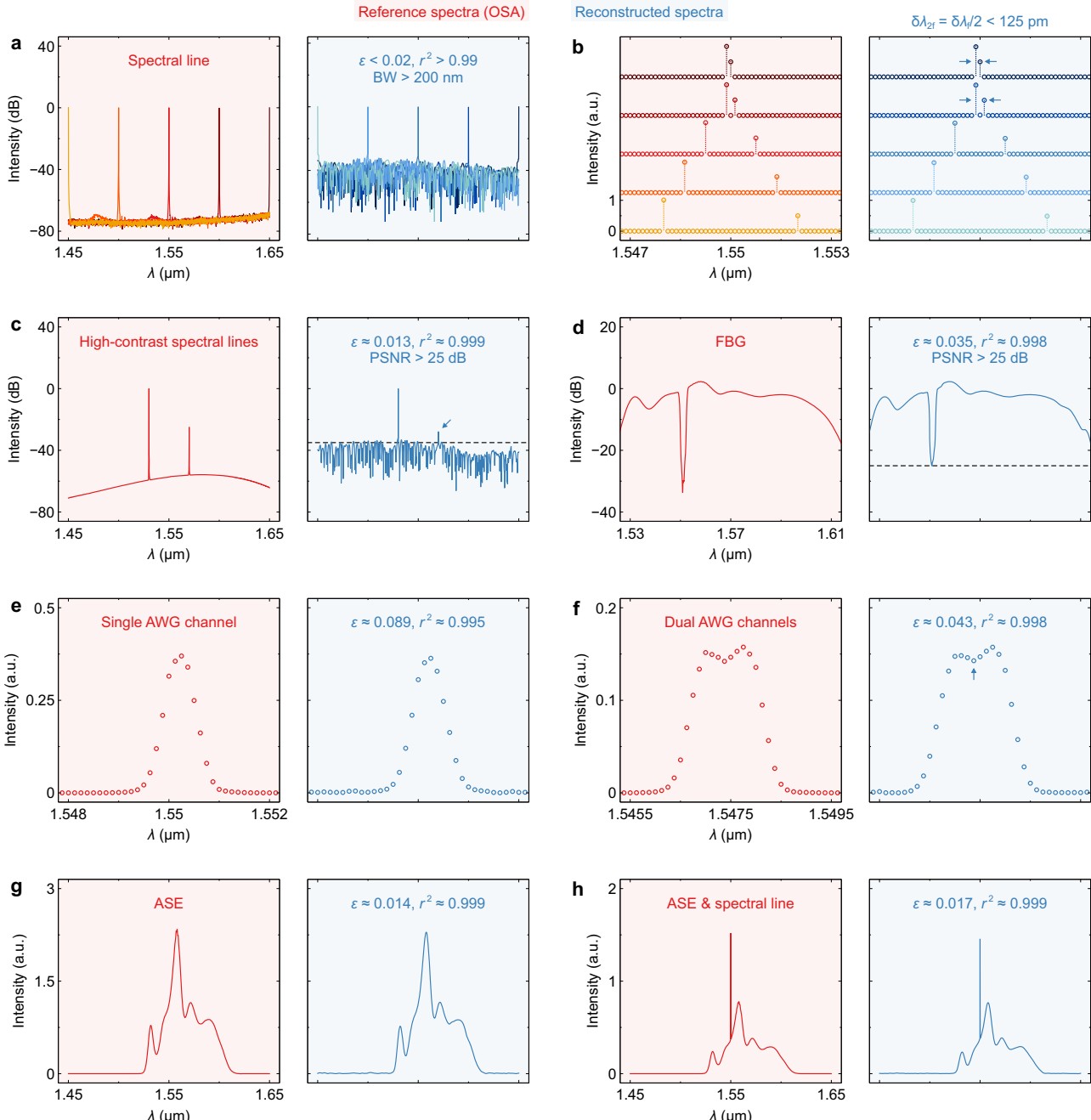

**Fig. 5 | Spectrum reconstruction.** Reconstruction results of the **a** single spectral lines, **b** dual spectral lines, **c** high-contrast spectral lines, **d** response of a fiber Bragg grating (FBG), **e** response of a single channel of an arrayed waveguide grating (AWG), **f** response of dual channels of an AWG, **g** amplified spontaneous emission (ASE), and **h** ASE with a spectral line superimposed. The reference and reconstructed spectra are shaded in red and blue, respectively. The reference spectra were measured using a commercial optical spectrum analyzer (OSA), while the reconstructed spectra are derived from the measured interferograms (see Supplementary information, Note 9, Figs. S19 and S20). The relative errors ($\varepsilon$) and coefficients of determination ($r^2$) are also labeled. In Fig. 5c, d, the dashed lines show the intensity levels of $\approx -35$ dB and $\approx -25$ dB, respectively. The peak signal-to-noise ratio is characterized to be PSNR > 25 dB. The demonstrated resolution and bandwidth are $\delta\lambda_{2f} = \delta\lambda_f/2 < 125$ nm and BW > 200 nm, respectively.

demonstrated 2D-FTS has high performance and great scalability. It is anticipated that the 2D-FTS will find applications in future spectroscopic systems.

## Methods

### Device simulation

The dispersion of the waveguide is simulated using the finite-difference frequency-domain (FDFD) method. The finite-difference time-domain (FDTD) method is utilized to calculate the transmission responses of the ADC and YBS. All these components are modeled as transferring matrices and imported into the Lumerical INTER-CONNECT module to obtain the transmittance cube of the 2D-FTS. Due to the difficulty in simulating TO nonlinearity, the TO tuning efficiencies used in the calculation are derived from the measured data. The entire simulation flow is performed with the Ansys Lumerical simulation suite.

### Measurement details

The transmission responses of the testing structures and 2D-FTS were measured using a narrow-linewidth TL (Keysight 8164B) and a

low-noise PD (Agilent 81532 A). The heating power was supplied by a voltage source (Keithley 2400), which was programmed to sweep over a series of sampling points. The structures shown in Fig. 2 were interconnected with single-mode fibers via grating couplers (GC). Due to the limited bandwidth of GCs, the full wavelength band was divided into two 100-nm bands and measured by modifying the incident angle of fiber probes. Light was injected into and coupled out of the 2D-FTS using lensed fibers and broadband edge couplers. The polarization state of incident light was aligned to TE with a polarization controller. In the measurement of the 2D-FTS, $IN_1$ and $OUT_{1-128}$ were used as input and output, respectively, while $IN_2$ and $OUT_0$ were dummy ports for the in-situ monitoring of the tFTS. The input fiber was aligned to $IN_1$, and the transmission responses at $OUT_{1-128}$ were individually measured. At the $i$-th measurement step, we aligned the output fiber to $OUT_i$. The $i$-th row of interferograms ($\mathbf{O}$) from all testing spectra and the $i$-th row of the transmittance cube ($\mathbf{A}$) were successively recorded with varying heating power. The input/output fibers were not relocated during the measurement of each port, ensuring an accurate mapping between $\mathbf{A}$ and $\mathbf{O}$. This proof-of-concept method has been used in prior works on SHSs[23] and speckle spectrometers[49]. Practical multiport acquisition can be realized by integrating silicon-germanium PDs to each output channel and reading their signals under a synchronized clock[50]. The monolithic integration of PDs can be supported by most commercial silicon photonic foundries[51]. Integrated PDs typically have an electric bandwidth of >20 GHz[51], so it is feasible to capture signals at all ports within the theoretical time span of a tuning step (<1 ms). With 25 tuning steps, the theoretical sampling period per spectrum is <0.025 s. The TL was also used to produce a single spectral line at varying wavelengths. We utilized a second TL (Keysight 8163B) and a fiber 3-dB coupler to produce dual spectral lines with varying spacings. The peak intensities were modified with variable optical attenuators. Various continuous spectra were produced with an ASE source (Fiberlake) and extra optical filters, such as the FBG and AWG. The emissions from the TL and ASE were simultaneously launched into the chip to produce the hybrid spectrum. All testing spectra were monitored by a commercial OSA (Yokogawa AQ6370D). The raw data from the OSA has a resolution of 20 pm, which is reorganized and fitted into the same grid as the 2D-FTS. During the measurement, the chip was mounted on a thermo-electric cooler (TEC) to stabilize the ambient temperature.

### Reporting summary

Further information on research design is available in the Nature Portfolio Reporting Summary linked to this article.

## Data availability

The data that support the findings of this study are included in the paper and its supplementary document. Other data are available from the corresponding authors upon reasonable request.

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

## Acknowledgements

We acknowledge Applied Nanotools for device fabrication. H.K.T. acknowledges the funding from the Innovation and Technology Fund (MRP/066/20). H.X. acknowledges ITF Research Talent Hub for financial support.

## Author contributions

H.X. initiated the project, designed the device, developed the reconstruction method, and prepared the mask layout. H.X, Y.Q., and G.H. built the measurement setup and performed the experiment. H.K.T. obtained the funding and supervised the project. H.X. and H.K.T. wrote the manuscript with support from all authors.

## Competing interests

The authors declare no competing interests.
