## [Peer Review File · Nature Communications]

Scalable integrated two-dimensional Fourier-transform spectrometryREVIEWER COMMENTS

Reviewer #1 (Remarks to the Author):

In the paper "Scalable integrated two-dimensional Fourier-transform spectrometry", the authors present a novel approach for realising simultaneously high-resolution and high-bandwidth integrated Fourier-transform spectrometry (FTS) devices. They combine the concepts of tunable FTS (tFTS) and spatial heterodyne spectrometer (SHS) to implement a device that benefits of the high bandwidth inherently given by the former, and the high resolution given by the latter. The key aspect resides on the fact that they combine the degrees of freedom provided by the two approaches to map the interferogram on a 3D space, which is then sliced to produce 2D maps at each value of the applied thermo-optic tuning power and relative optical path length. These maps are then fed to a 2D Fourier-transform algorithm for spectral reconstruction which is used to retrieve spectral content of the input signal out of unconventional power-delay maps, opportunely pre-processed.

The device features 250 pm resolution and 200 nm bandwidth, figures that can improve by a factor of two via computational methods. The method is interesting and well designed, though it mostly consists of numerical methods rather than a physical mechanism per se.

There are some important open questions and clarifications that need to be addressed before I can recommend the paper for publication in Nature Communications. A main concern is the data reconstruction process that seems to require an 'a priori' knowledge of the spectral content (according to Eq. 5). If so, the device is of poor applicability as a spectrometer device should be able to reconstruct arbitrary spectra without knowing the shape in advance. It is really required that the authors show the spectrum as directly obtained after Fourier-transforming the datasets with no optimisation applied for a convincing proof-of-concept.

General style comment: the paper is hard to follow due to the choice of referring constantly to quantities labelled at the beginning of the manuscript (A, S, O, etc). This makes it difficult to read and fully follow the reasoning.

Please address all the comments below:

- Line 51, it is said that SWIFTS devices cannot be integrated with monolithic PDs. This is partially true but there has been a recent publication (Nat. Photon, 17, 59-64, 2023) showing that electrical readouts are possible on SWIFTS and this, although the field is not yet mature, opens up a whole new set of opportunities and greatly boosts the potential of SWIFTS. This should be taken into account as the pixel size of the camera does not limit the bandwidth that much anymore. Second, SWIFTS have been shown to have among the largest bandwidths for integrated spectrometers, so the statement is incorrect.
- Line 88, probably the authors wanted to write "as long as the resolution...".
- Figure 2 (h,l) describing characterisation of coupler and Y-splitter. The experimental traces deviate from the simulated ones with additional features, for instance there seem to be some kind of weak resonances in the coupler's traces, is there an explanation as to why those features appear? What is their origin? Similar for the oscillations in the Y-splitters traces, which in the supplementary material seem to be attributed to the reflections at the grating couplers. Can the authors prove this? Do they have a reference trace for a straight waveguide? Also, the tolerance analysis doesn't seem to really support the statement that "coupling ratios are virtually constant". Was some kind of statistical study done on these components, since there are so many in their device?
- Figure 3, in particular the FFT phase is not commented nor described if not with a bare introduction and no explanation. Spectral analysis heavily relies on phase, which is the component carrying most of the information and is often overlooked. This pattern is not intuitive at all and, if shown, needs to be commented. Can the authors give a meaning to such a figure?
- Line 234, "It is conclusively demonstrated that the 2D-FTS has full orthogonality over all fringes". I would rather say that the discussed procedure can be applied to the data to provide

orthogonality. However, a question remains: is this true for any spectrum? To support the robustness of their approach, they should test the device with an arbitrary spectrum and/or comment on the feasibility of it.

- I understand that the paper is mostly about data processing and computational method, but the reported spectra are so close to the reference ones that there seems to be something artificial. For instance, equation 5, am I right in saying that the data reconstruction process seems to require a priori knowledge of the spectral content to then run an optimisation task? If so, the device is of poor applicability as a spectrometer device should be able to reconstruct arbitrary spectra without knowing the shape in advance. Also, the "regularization term Ω " should be better commented. Can the authors show the spectrum as directly obtained after Fourier-transforming the datasets with no optimisation applied? As it seems to be written in line 245, the reconstructed spectrum has large errors.

- Regarding the reconstructed spectra, a few comments:

- o Why changing the scale from dB and a.u.? Spectra can be rescaled to any peak value with this approach.

- o Spectrum 4b looks a bit superficial.

- o Again, reference and measured spectra are so similar that there doesn't seem to be almost any experimental error. It is a main concern and rises some doubts about the applicability of such a technique.

- o Is waveguide dispersion compensated for during data analysis? I see that it's considered in the supplementary, especially referring to the delay line, but for instance in the SHS there are many bends that will introduce further dispersion besides that of a straight waveguide.

- Line 273, I guess the authors made a typo and wanted to write pm instead of nm.

- Methods section, regarding measurements, it is not clear how the experiment was conducted.

Are the two inputs used at the same time, with a single lensed fibre? How can this give good coupling? What is the working distance of the fibre and its mode field diameter?

- Still regarding measurements. Line 323 "The transmission at 128 output ports was individually measured". This means the fibre needs to be moved across the outputs, which in turn requires to make sure to have good coupling and stable output. How was this done? How long does this take? Is it done manually or with an automatic feedback system of some sort? The inputs seem to be separated by 5-10 μm , was crosstalk between ports considered at the output? A spectrometer is useful if it can acquire and process data in a fast and reliable way, ideally seconds. Naturally, for an integrated spectrometer one cannot expect to have a measurement lasting as short as a commercial bulky system would do, but this sounds a long and cumbersome procedure. Additionally, was instability of light source and heater considered? The statement that the procedure "ensures precise data acquisition" sounds odd to me and is not enough to claim stable operation.

- Related to the comment above, it seems like a priori knowledge of the spectral content is required for such a precise operation. In line 109 there is written that the matrix is a collection of slices obtained by weighting the output with the spectral intensity, and if there is no control over the coupling efficiency and/or stability of the source, the device becomes unreliable.

Reviewer #2 (Remarks to the Author):

This paper presents a novel Fourier Transform Spectrometer (FTS) approach that combines two popular implementations of FTS: tunable FTS and spatial heterodyne FTS. The principle is clearly explained, and the characterization is comprehensive. The performance shows visible improvements in terms of bandwidth and resolution, demonstrating certain novelty and warranting publication. However, considering the lack of fundamental breakthroughs in the technical or theoretical aspects, I would suggest that the authors consider submitting to a more specialized journal instead of NC. The tunable FTS and spatial heterodyne FTS used in this work share the same principle as previous demonstrations, and the drawbacks associated to previous demonstrations mentioned by the authors are still present in their own work. For instance, in line 40, the authors criticize conventional spatial heterodyne FTS, stating that it reduces the etendue at

each port and diminishes the Jacquinet's advantage due to the use of multiple physical channels. But it is exactly the same situation in this work as the signal needs to be split into 128 MZIs. Similarly, in line 44, the authors criticize tunable FTS for its power consumption of approximately 5 W. However, this work still requires a power consumption of 2.5 W, which is at a similar level. No practical applications can tolerate this amount of power consumption. Another reason why I believe it is not suitable for NC is the performance. It seems that the authors may have unintentionally neglected some recently demonstrated spectrometers that have similar or even better performance. For example, a spectrometer with over 100 nm bandwidth and 30 pm resolution has been reported. In summary, this work appears to be more engineering-oriented rather than a fundamental breakthrough in theory or technique for realizing spectrometers. I believe it would be a good candidate for other photonics journals, such as IEEE JLT or Photonics Research.

Reviewer #3 (Remarks to the Author):

This is a review of the manuscript entitled "Scalable integrated two-dimensional Fourier-transform spectrometry" by Hongnan Xu, Yue Qin, Gaolei Hu, and Hon Ki Tsang submitted to the journal Nature Communications. In the manuscript the authors present a design of an integrated hybrid Fourier transform (FTS) spectrometer composed of a thermally tuned Mach-Zehnder interferometer (MZI) section followed by a fan out into a series of imbalanced passive MZI filters. The principle of operation essentially combines that of a thermally driven FTS (represented by the tunable MZI) with that of a spatial heterodyne spectrometer (represented by the passive MZI filters). The manuscript includes numerical and experimental validation, including multiple combinations of broadband and narrowband spectra. Overall the manuscript is well written, and the fusion of spectrometer concepts is novel. Consequently, I recommend that the manuscript be published in Nature Communications subject to minor revisions.

I have the following recommendations to strengthen the manuscript:

- Although your introductory overview is thorough, there is one additional type of FTS that you should mention, particularly as it has parallels to your device by operating in a second dimension. It is called a channel dispersed FTS (see reference [1] below for details).
- I think that some clarification could be used pertaining to the discussion of device efficiency, both in the manuscript and around table S1 in the supplementary material. You only discuss this in terms of power consumption, however this is a somewhat misleading measure because the devices don't generally operate continuously. The thermal relaxation coefficient of SOI devices is around 10 us, and a full sweep of the thermal FTS should take maybe 100us. When you consider the energy per spectrum measurement the devices look a lot better (and arguably this is the more important figure of merit). For context, this is a very small amount of energy, compared to the processor used for the taking the inverse transform. I think some consideration of the total energy budget would provide very useful context for your readers. I also recommend adding this as a column in table S1 (although you will have to estimate it).
- There is some imprecise terminology that I think you should consider revising. You use the term "arm length" to refer both to changes in the refractive index of an MZI, as well as to actual changes in MZI arm length. It would be more accurate to refer to changes in refractive index using something like optical path length to help distinguish the two cases. This could be very confusing to readers who are not familiar with the various FTS systems.
- The figures in the manuscript are very busy. I recommend breaking them up and reorganizing them, ideally so that they contain no more than 4 sub-images. This will help the scale tick marks become more visible, which will significantly help the readability.
- In figure 4 the spectral reconstruction seems very good, except in the case of high contrast spectral lines. Is there a theoretical reason for this? If so, can it be mitigated somehow?
- You mention in the supplementary information that the spectral reconstruction takes around 30s of calculation time with a 24 core CPU. Is this per spectrum, or is this for a sort of calibration after which the reconstruction occurs much faster?

References:

- [1] Hong, B., Monifi, F. & Fainman, Y. Channel dispersed Fourier transform spectrometer. *Commun Phys* 1, 34 (2018). <https://doi.org/10.1038/s42005-018-0036-1>

**To Reviewer #1's comments:**

**General remarks:**

In the paper "Scalable integrated two-dimensional Fourier-transform spectrometry", the authors present a
novel approach for realizing simultaneously high-resolution and high-bandwidth integrated Fourier-
transform spectrometry (FTS) devices. They combine the concepts of tunable FTS (tFTS) and spatial
heterodyne spectrometer (SHS) to implement a device that benefits of the high bandwidth inherently given
by the former, and the high resolution given by the latter. The key aspect resides on the fact that they
combine the degrees of freedom provided by the two approaches to map the interferogram on a 3D space,
which is then sliced to produce 2D maps at each value of the applied thermo-optic tuning power and relative
optical path length. These maps are then fed to a 2D Fourier-transform algorithm for spectral reconstruction
which is used to retrieve spectral content of the input signal out of unconventional power-delay maps,
opportunely pre-processed.

The device features a 250-pm resolution and a 200-nm bandwidth, figures that can improve by a factor
of two via computational methods. The method is interesting and well designed, though it mostly consists
of numerical methods rather than a physical mechanism per se.

There are some important open questions and clarifications that need to be addressed before I can
recommend the paper for publication in *Nature Communications*.

*Reply:*

We thank the reviewer for his/her careful reading and insightful comments. The reviewer expresses
his/her concerns mainly in the generality of the reconstruction method utilized in this work. There are three
prevalent reconstruction approaches in Fourier-transform spectrometry (FTS): discrete cosine transform
(DCT), pseudo inverse, and regularized iterative optimization. DCT is applicable typically to the integrated
FTSs based on low-dispersion large-mode-area platforms (e.g. silica^{R1} and laser-written^{R2} waveguides) or
the schemes^{R3} that are easy to calibrate. For silicon FTSS, however, the large dispersions and imperfect light
extinction will cause frequency leakage in the Fourier domain, making it challenging to precisely reconstruct
spectra via direct DCT. Pseudo inverse can be used to improve the accuracy of FTSS with non-ideal sinusoidal
responses, provided that all dispersion information is contained in a calibrated matrix. For instance, the first
silicon spatial heterodyne spectrometer (SHS) is based on pseudo inverse rather than DCT^{R4}. However, with
pseudo inverse, the reconstruction result is still sensitive to measurement noises. The regularized iterative
method used in this work offers higher robustness against dispersions, fabrication flaws, and environmental
perturbations. The core idea is to solve an inverse problem with calibrated transmissions and regularization
penalties. The so-called "priors" is actually a general range of possible features that may occur in a spectrum.
No specific knowledge of spectral contents is required prior to the measurement. The iterative optimization
is completely automatic, without any manual selection of parameters. The regularized iterative method is a

well-established, generic approach that has been widely applied to SHSs^{R5}, digital FTSs (dFTS^{R6,R7}) and some
similar applications, such as computed tomography (CT^{R8}) and radar detection^{R9}.

In this response letter, all remarks pertaining to the reconstruction method are compiled in **Comments**
**1-1. We will give a comprehensive discussion on this topic.** Alternatively, the reviewer can read the modified
Note 8 in **Supplementary information**. To improve readability, other technical comments are replied in a
reorganized order. The modifications to the manuscript are highlighted in red. The critical descriptions are
underlined.

**Comments 1-1 (about the reconstruction method):**

All the comments regarding the reconstruction method are listed below:

**(i)** It is really required that the authors show the spectrum as directly obtained after Fourier-transforming
the datasets with no optimization applied, for a convincing proof of concept.

**(ii)** Can the authors show the spectrum as directly obtained after Fourier-transforming the datasets with no
optimization applied? As it seems to be written in line 245, the reconstructed spectrum has large errors.

**(iii)** A main concern is the data reconstruction process that seems to require an “a priori” knowledge of the
spectral content (according to Equation 5). If so, the device is of poor applicability, as a spectrometer device
should be able to reconstruct arbitrary spectra without knowing the shape in advance.

**(iv)** For instance, Equation 5, am I right in saying that the data reconstruction process seems to require a
priori knowledge of the spectral content to then run an optimization task?

**(v)** Also, the “regularization term” Ω should be better commented.

**(vi)** However, a question remains: is this true for any spectrum? To support the robustness of their approach,
they should test the device with an arbitrary spectrum and/or comment on the feasibility of it.

**(vii)** Additionally, was instability of light source and heater considered?

**(viii)** Related to the comment above, it seems like a priori knowledge of the spectral content is required for
such a precise operation. In line 109, it is written that the matrix is a collection of slices obtained by
weighting the output with the spectral intensity, and if there is no control over the coupling efficiency and/or
stability of the source, the device becomes unreliable.

**(ix)** I understand that the paper is mostly about data processing and computational method, but the
reported spectra are so close to the reference ones that there seems to be something artificial.

**(x)** Again, reference and measured spectra are so similar that there does not seem to be almost any
experimental error. It is a main concern and rises some doubts about the applicability of such a technique.

*Reply and modifications:*

A comprehensive and detailed description of the spectrum-reconstruction method is provided here. As
the discussion is somewhat lengthy, we will begin by replying briefly to each of the above points:

To points **(i)** and **(ii)**:

With ideal sinusoidal responses, any arbitrary spectrum can be recovered via Fourier transform. However,
a real-world integrated Fourier-transform spectrometer (FTS) typically has strong dispersion and imperfect
light extinction, resulting in spectral leakage in the Fourier domain and severe degradation in reconstruction
accuracy. Actually, for most FTSs reported on the silicon nanophotonic platform, spectrum reconstruction is
implemented via iterative optimization rather than direct Fourier transform^{R6,R7}. We will provide additional
simulation and experimental results to explain this issue.

The related discussions can be found in lines 101-144, 328-332, 354-355 and Figure R1, R7.

To points **(iii)**, **(iv)**, and **(v)**:

Ω is a regularization term that sets a *general range* of possible characteristics that may occur in a spectrum.
It does *not* require specific knowledge of spectral contents before measurement, as most naturally occurring
features can be covered by Ω . In addition, the hyperparameters in Ω can be automatically optimized via cross
validation. The optimization process does *not* require any manual parameter selection. Similar regularized
iterative methods have been used in prior studies^{R6,R7}.

The related discussions can be found in lines 178-263, 269 and Figures R3, R4.

To points **(vi)**:

Generally, a spectrum is either continuous (or derivable), discrete (or underivable), or exhibiting hybridized
features. The proposed Ω can cover all these naturally occurring spectral features. To prove this, we will give
additional reconstruction examples for spectra with a great amount of randomly distributed high-frequency
components and a seemingly chaotic response.

The related discussions can be found in lines 270-298 and Figure R5.

To points **(vii)** and **(viii)**:

The instability of input light and power source will *not* significantly affect the reconstruction accuracy since
the undulation in a spectrum can be automatically filtered out during regularized iterations. We will provide
a numerical example to support this viewpoint.

The related discussions can be found in lines 299-308, 313-315 and Figure R6.

To points **(ix)** and **(x)**:

We thank the reviewer for his/her positive evaluation for the reconstruction accuracy demonstrated in this
work. Nevertheless, the reconstructed spectra are certainly *not* identical to the reference ones. We suppose
such a misunderstanding might result from the unclear visualization of reconstruction results. To enhance
the visibility of reconstruction errors, an additional figure with extended experimental data has been added.

The related discussions can be found in lines 356-358 and Figure R8.

**Discussion on the reconstruction method**

There are three prevalent approaches to spectrum reconstruction in Fourier-transform spectrometry (FTS):
discrete cosine transform (DCT), pseudo inverse, and regularized iterative optimization. For FTSs with ideal

Figure R1 Reconstruction with discrete cosine transform (DCT). (a) 1D interferograms (denoted as **O**) with an ideal sinusoidal response, non-uniform periods, non-uniform extinction ratios (ER), and background noises. (b) Spectra (denoted as **S**) reconstructed by 1D-DCT. The insets show the enlarged views around the spike. (c) Testing spectrum with discrete features. (d) 2D-DCT and (e) fast Fourier transform (FFT) of the interferogram. (f) Spectra reconstructed by 2D-DCT. The arrows highlight the correspondence of spikes in the input/reconstructed spectra and the spots in the DCT/FFT map. (g) Testing spectrum with continuous features. (h) 2D-DCT and (i) FFT of the interferogram. (j) Spectra reconstructed by 2D-DCT. The arrows highlight the correspondence of bumps in the input and reconstructed spectra.

sinusoidal responses, any arbitrary spectrum can be recovered from the DCT of the recorded interferogram. However, a real-world interferogram typically has deviations from its ideal form, especially for integrated FTSs with strong dispersion and imperfect light extinction. For clarity, we first consider a simplified 1D-FTS model. Given an input spectrum (denoted as **S**) with a single spike, the output interferogram (denoted as **O**) is sinusoidal-like, as shown in the 1st column of Figure R1a. Here, the number of sampling steps is set as 2^{10}

= 1024, while the period of \mathbf{O} is set as $2^6 = 64$, as an example. In the ideal case, the spectrum can be accurately
recovered, as shown in the 1st column of Figure R1b. According to Equation S7, however, the free spectral
range (FSR) of an MZI is wavelength-dependent, thereby resulting in non-uniform periods in interferograms,
as shown in the 2nd column of Figure R1a. The chirp in \mathbf{O} will broaden and split the retrieved spike, due to
spectral leakage^{R10} (see the 2nd column of Figure R1b). To be specific, DCT is the discrete Fourier transform
(DFT) of the continuation of a finite sequence; hence, a fractional period will inevitably cause discontinuity
and create new frequency components. Non-uniform extinction ratios (ERs) also result in the splitting at a
lower frequency, as shown in the 3rd columns in Figures R1a and R1b. In addition, the measurement noise
in \mathbf{O} will raise the noise floor in the rebuilt spectrum (see the 4th columns in Figures R1a and R1b). Next, we
provide some numerical examples of reconstruction through 2D-DCT. Here, the reconstruction is performed
based on the measured transmittance cube and computer generated spectra. The power sweep steps are set
as $N_{\text{FTS}} = 50$, in order to fully unfold the DCT map and avoid channel obscurity (see Note 6 for explanations).
A white Gaussian noise of $\pm 1\%$ is imposed to the interferograms to emulate the errors in the measurement.
Figure R1c shows a discrete input spectrum with four spikes. The DCT result is shown in Figure R1d. Four
distinctive spots, corresponding to four spikes in the spectrum, can be observed from the DCT map. For cross
reference, we also show the fast Fourier transform (FFT) of the interferogram (see Figure R1e). According
to the trajectories shown in Figure S8b, the 2D-DCT map is transformed into a 1D vector. Since N_{FTS} exceeds
the critical value, the raw vector encompasses redundant elements. The effective spectral information can
be obtained by truncating the vector at two ends of the bandwidth, as shown in Figure R1f. The displayed
spectrum is normalized with Parseval's theorem^{R10}. Four spikes are discernible in the result. A spot in a 2D-
DCT map has leakage in two dimensions (see Figure R1d). The broadening and splitting along f_{FTS} will lead
to the false spikes that are distant from the original one, as the map is flattened in columns (see Figure S8b).
Therefore, the distribution of noise-like errors is not limited to the vicinity of spikes but is spread across the
entire bandwidth. Also, due to the leakage of integral power, the intensities at spike locations are inaccurate.
Since errors accumulate over all wavelengths, the reconstruction accuracy of a continuous spectrum is even
worse (see Figures R1g-R1j). From these results, it is challenging to implement reconstruction with DCT due
to the large errors. In previous studies, the DCT method is applicable typically to the integrated FTSs using
low-dispersion large-mode-area platforms, such as silica^{R1} and laser-written^{R2} waveguides, or some simple
schemes^{R3} that are easy to calibrate.

As demonstrated in the early research on silicon spatial heterodyne spectrometers (SHS^{R11}), it is also
possible to reconstruct a spectrum from an interferogram using pseudo inverse. This method requires the
calibration of the transmittance matrix (in this work, a cube) that depicts the transmission of an FTS at all
wavelengths. In the matrix (or cube, denoted as \mathbf{A}), each column is a distinctive fringe pattern (denoted as
\mathbf{a}_i) at a specific wavelength. The output interferogram is thus a linear combination of fringes:

Figure R2 Reconstruction with pseudo inverse. Reconstruction of (a) a discrete spectrum and (b) a continuous spectrum utilizing pseudo inverse. White Gaussian noises [denoted as wgn(-)] of different levels are imposed to the interferograms (denoted as \mathbf{O}). The input and reconstructed spectra (denoted as \mathbf{S}) are displayed in red and blue, respectively. The relative errors (ϵ) and coefficients of determination (r^2) are also labeled.

$$\mathbf{O} = \sum_{i=1}^{N_{2f}} s_i \mathbf{a}_i, \quad (\text{R1})$$

where s_i denotes the intensity at the i -th wavelength channel, and N_{2f} denotes the channel capacity. This is
equivalent to a matrix multiplication:

$$\mathbf{O} = \mathbf{A}\mathbf{S}. \quad (\text{R2})$$

In essence, the reconstruction of a spectrum is to determine the weight (i.e., s_i) of each fringe (i.e., \mathbf{a}_i), which
can be realized by the inverse operation of Equation R2:

$$\mathbf{S} = \mathbf{A}^{-1}\mathbf{O}, \quad (\text{R3})$$

where \mathbf{A}^{-1} denotes the Moore-Penrose inverse of \mathbf{A} . Compared to DCT, the pseudo-inverse method does not
require ideal sinusoidal responses or any phase compensation since the dispersions of FSRs and ERs have
been incorporated into the calibrated matrix/cube. If the recorded interferogram is completely accurate and
noiseless, then any arbitrary spectrum can be precisely reverted with Equation R3. Due to the presence of
temperature fluctuation (see Note 7) and fiber jittering, however, the interferogram usually contains errors:

$$\mathbf{O} = \mathbf{O}_{\text{ext}} + \Delta\mathbf{O}, \quad (\text{R4})$$

where \mathbf{O}_{ext} denotes the error-free interferogram, and $\Delta\mathbf{O}$ denotes the measurement error. As a consequence,
the reconstruction results also involve errors ($\Delta\mathbf{S}$):

$$\Delta \mathbf{S} = \mathbf{A}^{-1} \Delta \mathbf{O}. \quad (\text{R5})$$

Figure R2 shows the numerical examples of reconstruction with pseudo inverse. The white gaussian noises [denoted as $\text{wgn}(\cdot)$] with varying strength are applied to the interferogram. Here, the transmittance cube is based on the measurement with $N_{\text{FTS}} = 25$. The reconstruction accuracy is evaluated with relative error (ϵ) and coefficient of determination (r^2). All spectra are displayed with their absolute values. The errors become increasingly severe at a higher noise level. The reconstruction of a continuous spectrum is more susceptible to noises since an FTS collects information of all channels at each sampling and $\Delta \mathbf{O}$ will therefore influence the entire spectrum. Overall, the main drawback of the pseudo-inverse method is its sensitivity to noises.

It is possible to solve the inverse problem of Equation R2 via iterative optimization instead of pseudo inverse:

$$\mathbf{S} = \underset{\mathbf{s}}{\text{argmin}} \left(\|\mathbf{A}\mathbf{S} - \mathbf{O}\|_2^2 \right), \quad (\text{R6})$$

where $\text{argmin}(\cdot)$ denotes global minimum, and $\|\cdot\|_2$ denotes ℓ_2 -norm. Nevertheless, the noises in \mathbf{O} will still affect the convergence in the search for the optimal \mathbf{S} . A generic solution is to add a regularization term (Ω) to Equation R6^{R12}:

$$\mathbf{S} = \underset{\mathbf{s}}{\text{argmin}} \left(\|\mathbf{A}\mathbf{S} - \mathbf{O}\|_2^2 + \Omega \right). \quad (\text{R7})$$

Ω is a penalty that imposes cost to the optimization function (i.e., $\|\mathbf{A}\mathbf{S} - \mathbf{O}\|_2^2$), in order to bias the solution towards preconditioned features. With the use of Ω , the reconstruction error (i.e., $\Delta \mathbf{S}$) becomes^{R13}:

$$\Delta \mathbf{S} = (\mathbf{I} - \mathbf{V}\Psi\mathbf{V}^T) \mathbf{S} - \mathbf{V}\Psi\Sigma^{-1}\mathbf{U}^T \Delta \mathbf{O}, \quad (\text{R8})$$

where \mathbf{I} denotes the identity matrix. \mathbf{U} , Σ , and \mathbf{V} are the matrices produced via singular value decomposition (SVD, $\mathbf{A} = \mathbf{U}\Psi\mathbf{V}^T$), and Ψ denotes the filtering matrix determined by Ω ^{R13}. In Equation R8, the first [i.e., $(\mathbf{I} - \mathbf{V}\Psi\mathbf{V}^T)\mathbf{S}$] and second (i.e., $\mathbf{V}\Psi\Sigma^{-1}\mathbf{U}^T \Delta \mathbf{O}$) parts represent the regularization penalty and noise perturbation, respectively. The proper selection of Ω will therefore balance the penalty and perturbation and mitigate the influence of measurement noises. This novel method has been applied to SHSs^{R5} and digital FTSs (dFTS^{R6,R7}) with various forms of Ω (e.g., compressed sensing and elastic networks). As a well-established approach, it has also been applied to computed tomography (CT^{R8}) and radar detection^{R9}. In this work, Ω is formulated as:

$$\Omega = \Omega_1 + \Omega_2 = \zeta_1^2 \|\mathbf{D}_2 \mathbf{S}_1\|_2^2 + \zeta_2 \|\mathbf{D}_1 \mathbf{S}_2\|_1, \quad (\text{R9})$$

where ζ_i denotes the regularization parameter, \mathbf{D}_i denotes the i -th order derivative operator, \mathbf{S}_1 and \mathbf{S}_2 denote the continuous and discrete components in the spectrum. The complete reconstruction formula can be thus expressed as:

$$\mathbf{S} = \underset{\mathbf{s}}{\text{argmin}} \left(\|\tilde{\mathbf{A}}_{\text{FFT}} \mathbf{S} - \text{FFT}(\tilde{\mathbf{O}})\|_2^2 + \zeta_1^2 \|\mathbf{D}_2 \mathbf{S}_1\|_2^2 + \zeta_2 \|\mathbf{D}_1 \mathbf{S}_2\|_1 \right), \quad (\text{R10})$$

where $\tilde{\mathbf{A}}_{\text{FFT}}$ denotes the matrix formed by the column vectors of $\text{FFT}(\tilde{\mathbf{a}}_i)$, $\tilde{\mathbf{a}}_i$ denotes the flattened fringe with zero-frequency components removed, and $\tilde{\mathbf{O}}$ denotes the interferogram after the component removal. More

explanations of component removal and Fourier-domain operations can be found in Note 6. In essence, Ω is
 a precondition (or a priori) for the reconstruction result. Remarkably, Ω only sets a *general* range of possible
 characteristics that may occur in a spectrum. It does *not* require specific knowledge of spectral details before
 measurement. From Equation R9, the regularization term encompasses two parts, i.e., Ω_1 with ℓ_2 -norm and
 Ω_2 with ℓ_1 -norm. Ω_1 is commonly known as Tikhonov regularization^{R14} that provides smoothening to the
 spectrum. Specifically, during iterations, Ω_1 will decrease when the derivative of the updated spectrum has
 a smaller mean square; as a consequence, with a larger ζ_1 , the spectrum will become more continuous. On
 the other hand, Ω_2 provides total-variation (TV^{R8}) regularization that imposes more discrete features to the
 outcome of the optimization function. Generally, a spectrum is either continuous (or derivable) or discrete
 (or underivable); therefore, the proposed Ω can cover most naturally occurring spectral features. A hybrid
 spectrum can also be rebuilt using this method with non-zero values for both ζ_1 and ζ_2 . The reconstruction
 of a spectrum with chaotic features will be discussed later. The weights of Ω_1 and Ω_2 are associated with ζ_1
 and ζ_2 , respectively. By using cross validation (CV), these hyperparameters can be automatically optimized
 without any manual selection. For K -fold CV, the basic concept is to divide the interferogram into K sets, one
 of which is used to produce a “reduced” solution to predict the elements in other sets. The prediction error
 reaches minimum with the optimal ζ_1 and ζ_2 ^{R15}:

$$219 \quad (\zeta_1, \zeta_2) = \underset{\zeta_1, \zeta_2}{\operatorname{argmin}} \left[\sum |\tilde{\mathbf{a}}_i \mathbf{S}_i - \tilde{o}_i|^2 \right], \quad (\text{R11})$$

where \mathbf{S}_i denotes the solution with $\tilde{\mathbf{a}}_i$ left out in the cube, and \tilde{o}_i denotes the i -th element in the interferogram.
 For a smooth spectrum, ζ_2 will descend to zero, and only Ω_1 will function in the regularization term. Similarly,
 when the input spectrum is spike- or step-like, the penalty will be dominated by Ω_2 . If multiple features are
 hybridized in a spectrum, then both ζ_1 and ζ_2 will be non-zero, with their optimal values selected by CV. In
 this work, the number of subsets is chosen as $K = 5$. The search for global optimum is enabled by a standard
 least-squares solver^{R16}. The CV procedure is embedded within the iterative optimization. The search for the
 optimal ζ_1 and ζ_2 is performed over an 8×8 space arranged in the log scale. From Equation R10, the spectrum
 is decomposed into two components in accordance with their spectral features, i.e., $\mathbf{S} = \mathbf{S}_1 + \mathbf{S}_2$. To ensure a
 single-vector input, the formula is modified as:

$$229 \quad \mathbf{S} = \underset{\mathbf{s}}{\operatorname{argmin}} \left(\left\| \begin{bmatrix} \tilde{\mathbf{A}}_{\text{FFT}} \\ \tilde{\mathbf{A}}_{\text{FFT}} \end{bmatrix} \begin{bmatrix} \mathbf{S}_1 \\ \mathbf{S}_2 \end{bmatrix} - \tilde{\mathbf{O}} \right\|_2^2 + \zeta_1^2 \left\| \begin{bmatrix} \mathbf{D}_2 \\ 0 \end{bmatrix} \begin{bmatrix} \mathbf{S}_1 \\ \mathbf{S}_2 \end{bmatrix} \right\|_2^2 + \zeta_2 \left\| \begin{bmatrix} 0 \\ \mathbf{D}_1 \end{bmatrix} \begin{bmatrix} \mathbf{S}_1 \\ \mathbf{S}_2 \end{bmatrix} \right\|_1 \right), \quad (\text{R12})$$

Thus, the searching vector becomes $[\mathbf{S}_1, \mathbf{S}_2]$, and the transmittance cube becomes $[\tilde{\mathbf{A}}_{\text{FFT}}, \tilde{\mathbf{A}}_{\text{FFT}}]^T$. In this work,
 spectrum reconstruction is realized using the least-squares QR-decomposition (LSQR) module in IRtools^{R16}.
 It is also easy to reproduce our results with other open-source packages (e.g., Pylops^{R17}) since Equation R12
 is in the standard form of a regularization problem.

**Figure R3 Picard plots.** (a) Generated testing spectra. Picard plots for the (b) simulation and (c) experimental results.

To further verify the effectiveness of the proposed method, the solvability of Equation R7 is assessed
with the Picard plot^{R18}. Based on SVD, the “naïve” solution to the inverse problem can be written as:

$$238 \quad \mathbf{S} = \sum_{i=1}^{N_{2f}} \frac{\mathbf{u}_i^T \mathbf{O}}{\sigma_i} \mathbf{v}_i, \quad (\text{R13})$$

where \mathbf{u}_i denotes the i -th left singular vector, \mathbf{v}_i denotes the i -th right singular vector, σ_i denotes the singular
value, and N_{2f} denotes the capacity. From Equation R13, it is revealed that an ideal reconstruction result is
formed on the basis of right singular vectors (i.e., \mathbf{v}_i) that are weighted by SVD coefficients (i.e., $\mathbf{u}_i^T \mathbf{O}/\sigma_i$). It is
thus essential to ensure that the SVD coefficient levels off to a finite value (commonly known as the Picard
condition^{R18}); otherwise, the integral of N_{2f} channels will be infinite and the iterative process will suffer from
a poor convergence. Four different types of spectra (i.e., plateau, Gaussian, spike, and random functions) are
used for testing, as shown in Figure R3a. In Figures R3b and R3c, we calculate the absolute values of SVD
coefficients, sampling weights ($|\mathbf{u}_i^T \tilde{\mathbf{O}}|$), and singular values in a semi-log plot. The simulated and measured
SVD coefficients do not overall increase even at a high index, demonstrating that the Picard condition is met,
and that a convergent solution can always be obtained in Equation R7.

 **Figure R4 Numerical test.** Numerical spectrum reconstruction of (a) smooth spectra, (b) sparse spectra, (c) step spectra, and (d)
 hybrid spectra. The relative errors (ϵ) and coefficients of determination (r^2) are labeled.

Figure R4 shows some numerical reconstruction examples. Here, we mainly consider the spectra with
 smooth, sparse, and step features, as shown in Figures R4a-R4c. The hybrid spectra with multiple features
 are also discussed, as shown in Figure R4d. To emulate environmental perturbations, we use the recorded
 temperature fluctuations (see Note 7) and calculated temperature sensitivities (see Figure S1f) to generate
 noises in the interferogram. A high reconstruction accuracy is attained for all different types of spectra. The
 reconstruction was implemented with MATLAB on a 24-core 3-GHz Intel Xeon Gold CPU. If all the optimal
 hyperparameters (i.e., ζ_i) are known, then the fixed-parameter reconstruction time (FPRT) will be as short
 as < 1 s. When ζ_1 and ζ_2 are free to optimize, the time cost will increase due to the CV procedure. In the worst
 case, when the full 8×8 searching space must be traversed, the total reconstruction time is < 60 s. There are
 several strategies to expedite the reconstruction. First, since the iterative least-squares solver relies heavily
 on matrix multiplication, the reconstruction can be significantly accelerated by employing a GPU. Second, it
 is possible to train a deep-learning network to identify spectral features directly from the interferogram and

 **Figure R5 Reconstruction of random spectra.** (a-d) Testing functions and their discrete Fourier transform (DCT) with different Λ .
 Here, Λ is a parameter that determines the fraction of random responses in the Fourier domain. (e) Input and (f) reconstructed spectra
 with varying Λ . The relative errors (ϵ) and coefficients of determination (r^2) are also labeled. Calculated (g) ϵ and (h) r^2 with varying Λ .
 Normalized deviations $[\Delta S / \max(S)]$ with (i) $\log_{10}(\Lambda) = -0.9$ and (j) $\log_{10}(\Lambda) = -2.15$.

determine ζ_i without CV^{R19}, which may reduce the reconstruction period to a single FPRT.

From above analysis, we have numerically and experimentally demonstrated the feasibility in precisely
 retrieving spectra using the regularized iterative method. The applications in spectroscopy, communications,
 and imaging typically require recovering a spectrum that is either smooth (e.g., spontaneous emission and
 near-infrared spectroscopy), sparse (e.g., WDM signals and atomic spectroscopy), or step-like (e.g., rejection
 band of an optical filter). All these scenarios can be covered by the proposed 2D-FTS scheme. Nevertheless,
 it is possible that a spectrum may have a great amount of randomly distributed high-frequency components

and a seemingly chaotic response. Here, we will explore the capability in retrieving a spectrum with a high
 degree of randomness. It is difficult to experimentally produce random testing spectra. As a proof of concept,
 we use the measured transmittance cube and generated random functions to perform a numerical test. The
 Fourier-Wiener series are leveraged to create sequences with controllable randomness^{R20}:

$$280 \quad \mathbf{S} = \sum_{i=-m}^m c_i D(x - ih), \quad (\text{R14})$$

$$281 \quad D(x) = \frac{\sin[(2m+1)\pi x]}{(2m+1)\sin(\pi x)}, \quad (\text{R15})$$

$$282 \quad m = \lfloor 1/\Lambda \rfloor, \quad (\text{R16})$$

where x is a N_{2f} -point sequence ranging from 0 to 1, c_i is the i -th element in a computer generated random
 sequence, and Λ is a parameter that tunes the randomness of \mathbf{S} . Given a relatively large Λ , the function output
 resembles a 1D random walk, as shown in Figure R5a. The reduction of Λ will increase the proportion of
 random responses in the DCT domain (see Figure R5d), making the generated spectrum more chaotic (see
 Figure R5c). By using Fourier-Wiener series, we can reveal the evolution of reconstruction accuracy when
 the input spectrum transitions from a smooth function to a chaotic function. Figure R5e shows the testing
 spectra with the parameter ranging from $\log_{10}(\Lambda) = -0.9$ to -2.15 . The corresponding reconstruction results
 are listed in Figure R5f. Figures R5g and R5h respectively show the relative errors (ε) and coefficients of
 determination (r^2) as functions of Λ . In Figures R5i and R5j, we also calculate the normalized deviations [i.e.,
 $\Delta\mathbf{S}/\max(\mathbf{S})$] with $\log_{10}(\Lambda) = -0.9$ and -2.15 . The calculated ε and r^2 curves are still quite flat even with a high
 degree of randomness in the spectrum. At $\log_{10}(\Lambda) = -2.15$, the input spectrum becomes chaotic, but a high
 reconstruction accuracy of $\varepsilon < 0.1$ and $r^2 > 0.85$ can still be achieved. The slight degradation in accuracy may
 results from the rivalry between Tikhonov and TV regularizations during the automatic CV procedure, as a
 chaotic spectrum is neither continuous nor discrete, and the regularization penalty cannot fully compensate
 for the noise perturbation. Form these results, it is demonstrated the regularized iterative optimization has
 the potential to cover a wide range of spectral features and serve as a generic reconstruction method.

Due to platform vibration and fiber jittering, the input spectrum can be instable during measurement.
 In Figure R6, we give an example of spectrum reconstruction with varying levels of intentionally introduced
 instability. Here, a white Gaussian noise [i.e., $\text{wgn}(\cdot)$] is employed to emulate the undulation in the spectrum,
 as shown in Figure R6a. The reconstruction results are shown in Figure R6b. It can be found that the noises
 are filtered out and all rebuilt spectra are “smoothened”. This effect can be explained as follows. According
 to the Picard plot shown in the 4th column of Figure R3, the perturbation component in \mathbf{S} will not result in
 the divergence of the solution. By using CV, the input spectrum will be automatically recognized as a smooth
 one. The Tikhonov-regularization term (i.e., Ω_1 , see Equation R9) will thus cancel out the non-smooth part.
 This effect holds valid as long as the instability of a spectrum does not overwhelm its key features; otherwise,
 the undulation will also appear in the reconstructed spectrum, as discussed in Figure R5. Our measurement

**Figure R6 Reconstruction under input instability.** (a) Testing input spectra with intentionally introduced instability. Here, $\text{wgn}(\cdot)$
 denotes white Gaussian noise. (b) Reconstructed spectra under varying levels of instability. The relative errors (ϵ) and coefficients of
 determination (r^2) are also labeled.

setup (e.g., positioning stage and fiber holder) is mechanically stable. All testing spectra were produced with
 fiber-connected commercial optical sources and filters. The input instability was well controlled and limited
 to a relatively low level during the experiment, while the remnant perturbation can be numerically resolved.

Table R1. Comparison of reconstruction methods in integrated Fourier-transform spectrometry.

Design	Method	Formula	Iterative	Noise sensitivity	Solvable spectral feature			
					Smooth	Sparse	Step	Hybrid
SHS ^{S21}	DCT	$\mathbf{S} = \text{DCT}(\mathbf{O})$	N ^(a)	High	Y	Y	Y	Y
SHS ^{S4}	Pseudo inverse	$\mathbf{S} = \mathbf{A}^{-1}\mathbf{O}$	N	High	Y	Y	Y	Y
SHS ^{S22}	Reg. (LASSO)	$\mathbf{S} = \text{argmin}(\zeta \ \mathbf{S}\ _1)$	Y ^(b)	Low	Y	Y	N	N
dFTS ^{S6}	Reg. (elastic-net)	$\mathbf{S} = \text{argmin}(\zeta_1^2 \ \mathbf{S}\ _2^2 + \zeta_2 \ \mathbf{S}\ _1)$	Y	Low	Y	Y	N	N
This work	Reg. (Tikhonov/TV)	$\mathbf{S} = \text{argmin}(\ \tilde{\mathbf{A}}_{\text{FFT}}\mathbf{S} - \tilde{\mathbf{O}}\ _2^2 + \zeta_1^2 \ \mathbf{D}_2\mathbf{S}_1\ _2^2 + \zeta_2 \ \mathbf{D}_1\mathbf{S}_2\ _1)$	Y	Low	Y	Y	Y	Y

DCT, discrete cosine transform.

LASSO, least absolute shrinkage and selection operator.

TV, total variation.

Reg., regularization iterative method.

321 ^(a)No.

322 ^(b)Yes.

 **Figure R7 Comparison of reconstruction with different methods.** Reference and reconstruction results of (a-d) a single spectral
 line, (e, f) amplified spontaneous emission (ASE), and (g, h) the response of a fiber Bragg grating (FBG). The 1st, 2nd, and 3rd columns of
 the right panel show the reconstruction results based on discrete cosine transform (DCT), pseudo inverse, and the regularized iterative
 method, respectively. The relative errors (ϵ) and coefficients of determination (r^2) are also labeled.

In Figure R7, we compare the experimental reconstruction results based on three methods discussed
 above: DCT, pseudo inverse, and regularized iterative optimization. The testing spectra include spectral lines,
 amplified spontaneous emission, and the response of a fiber Bragg grating, as an example. The DCT results
 have a high noise floor and a great amount of false peaks. By using pseudo inverse, a single spectral line can
 be retrieved with a peak signal-to-noise ratio of PSNR \approx 15 dB. However, it is still impossible to reconstruct

Figure R8 Extended experimental data. Reconstruction results of (a) a single spectral line, (b) dual spectral lines, (c) high-contrast spectral lines, (d) the response of a FBG, (e) the response of a single channel of an AWG, (f) the response of dual channels of an AWG, (g) the response of ASE, and (h) an ASE superimposed with a single spectral line. In each figure, the left panel shows the experimental reconstruction result and reference spectrum from a commercial OSA, while the right panel shows normalized deviations $[\Delta\mathbf{S}/\max(\mathbf{S})]$. Here, $\Delta\mathbf{S}$ denotes the difference between the rebuilt and reference spectra, and $\max(\mathbf{S})$ denotes the maximum element in the reference spectrum. The insets show the enlarged views of spectra around the wavelength ranges indicated by the arrows. For the reconstruction of a sparse spectrum, errors typically appear at the peak locations, which can be explained as follows. The initial guess of an unknown spectrum is an all-zero sequence. In a sparse spectrum, most elements are close to zero, except for the peaks. Consequently, most near-zero elements will reach its optimum after a few iterations, but will continue to be updated, resulting in a noise-like background. In the meantime, the solving of peak values requires more iterations. The cumulative errors from other elements will thus affect the accuracy of peak reconstruction. The reconstruction of an ideally sparse or smooth spectrum typically has a high accuracy of $\Delta\mathbf{S}/\max(\mathbf{S}) < 2\%$. However, the background noise will increase when reconstructing the sharp but spreading response of an AWG (see Figures R8e and R8f). Albeit being derivable, the AWG response has a fast-changing derivative around the resonant wavelength and a majority of near-zero elements. During cross validation, the ℓ_1 -norm term will therefore compete with the ℓ_2 -norm term and result in a small but non-zero ζ_2 , which imposes a higher noise floor to the retrieved spectrum and slightly reduces the peak signal-to-noise ratio to $\text{PSNR} \approx 20$

349 dB. Relatively larger errors [$\Delta S/\max(S) > 5\%$] can also be found in the reconstruction of the hybrid spectra shown in Figures R8d and
350 R8h, which is also caused by the imperfect selection of two hyperparameters. Nevertheless, for all tested spectra with various features,
small relative errors of $\varepsilon < 0.1$ and high coefficients of determination of $r^2 > 0.99$ can be attained. A higher accuracy can be obtained by
transforming the spectrum with a basis (e.g., DCT^{R23} and Lorentzian decomposition^{R24}) that aligns better with pre-conditioned features.
FBG, fiber Bragg grating. AWG, arrayed waveguide grating. ASE, amplified spontaneous emission.

a smooth or step spectrum. Our proposed method, in contrast, supports small errors and low noises for all
different types of spectra. Table R1 compares different reconstruction methods in reported integrated FTSS.
The reference and experimentally reconstructed spectra are displayed in the same subplot of Figure R8. The
normalized deviations [$\Delta S/\max(S)$] are also plotted to enhance the visibility of errors. More discussions on
reconstruction accuracy are provided in the caption.

In the revised manuscript, we have added the following sentences:

“Therefore, the spectrum recovered directly with DCT has large errors (see **Supplementary**
**Information**, Figure S12). The pseudo-inverse method also suffers from a poor reconstruction accuracy
(see **Supplementary Information**, Figure S13).” (**Spectrum reconstruction**, lines 272-274)

“The hyperparameters in Ω are automatically optimized via cross validation (CV⁴⁵) without manual
selection. Remarkably, Ω only sets a general range of features that may occur in a spectrum, and no specific
knowledge of spectral contents are required.” (**Spectrum reconstruction**, lines 282-284)

“The feasibility of reconstructing a spectrum of arbitrary shape is further discussed in **Supplementary**
**Information**, Figure S16. In **Supplementary Information**, Figure S18, the reconstruction accuracies based
on DCT, pseudo inverse, and regularized iterative optimization are compared.” (**Spectrum reconstruction**,
lines 286-289)

“More numerical reconstruction examples and extended experimental data can be found in
**Supplementary information**, Notes 8 and 9, respectively.” (**Spectrum reconstruction**, lines 317-318)

The reference list is updated accordingly:

“45. Golub GH, Heath M, Wahba G. Generalized cross-validation as a method for choosing a good ridge
parameter. *Technometrics* **21**, 215-223 (1979).” (**References**, lines 487-488)

An inaccurate expression has been deleted:

“ ~~N_f also indicates the number of points in the DCT map.~~” (**Characterization and analysis of the**
**spectrometer**, line 218)

All the discussions and figures in this section have been added to **Supplementary information**, Notes
8 and 9.

**Comment 1-2 (about fringe decorrelation):**

Line 234, “It is conclusively demonstrated that the 2D-FTS has full orthogonality over all fringes”. I would
rather say that the discussed procedure can be applied to the data to provide orthogonality.

*Reply and modifications:*

We presume that the reviewer may have a confusion regarding the terms “interferogram” and “fringe”.

In this work, a spectrum is reconstructed by solving a linear inverse problem:

$\mathbf{O} = \mathbf{A}\mathbf{S}$. (R17)

where \mathbf{O} denotes the recorded signal, or “interferogram”, \mathbf{A} denotes the transmittance cube, and \mathbf{S} denotes
the input spectrum. Each column of \mathbf{A} is a 2D pattern serving as a unique signature for a specific wavelength.
The term “fringe” (denoted as \mathbf{a}_i) refers to the signature pattern sliced at a single channel of the cube:

$\mathbf{A} = [\mathbf{a}_1, \mathbf{a}_2, \mathbf{a}_3, \dots, \mathbf{a}_{N-2}, \mathbf{a}_{N-1}, \mathbf{a}_N]$. (R18)

These terms are commonly used in previous research on Fourier-transform spectrometry (e.g., see Ref. R22).

It is essential to ensure that all fringes are virtually decorrelated, or orthogonal. If there are channels with
highly correlated fringe patterns, it will be impossible to find a convergent solution to Equation R17 since a
matrix (flattened from the transmittance cube) with identical columns is noninvertible. The orthogonality,
an inherent property of the cube, is irrelevant to the recorded data (i.e., “interferogram”) or reconstruction
technique. Nevertheless, the term “full orthogonality” might be less rigorous in this sentence since there are
still some remnant correlations and the singular-value curve is not ideally flat (see Figure 4). In the revised
manuscript, we have replaced the term “full orthogonality” with “sufficient decorrelation”:

“Thus, it is conclusively demonstrated that the 2D-FTS has sufficient decorrelation over all fringes.”

**(Characterization and analysis of the spectrometer, line 261-262)**

The following sentences have also been modified:

“The cube can be sliced into a series of fringe patterns (denoted as \mathbf{a}_i) at varying wavelengths. In the
Fourier domain, each fringe is related to a spot at distinct Fourier frequencies (f_{FTS} and f_{SHS}), as discussed in
Figure 1c.” **(Design principle, line 112-114)**

“The recorded interferogram (\mathbf{O}) is a linear combination of fringes (\mathbf{a}_i), with the weight on \mathbf{a}_i indicating
the spectral intensity at the i -th wavelength” **(Design principle, line 115-116)**

“The correlation matrix is quasi-diagonal when the heating power reaches the critical value of $P_{\text{max}} =$
2.4 W, demonstrating that all the wavelength channels are highly decorrelated.” **(Supplementary**
**information, Note 6, line 342-344)**

It should be noted that, compared to 1D-FTSs, the Fourier map of the 2D-FTS is more complex, and the
decorrelation of fringes becomes less intuitive. In addition, the 2D-FTS has the potential to reduce the sweep
steps and support a folded Fourier map. This issue was not thoroughly discussed in the original manuscript.
Herein, we provide a comprehensive discussion on fringe decorrelation.

**Discussion on fringe decorrelation**

The fringe decorrelation and feasibility of reducing sweep steps are discussed here. The critical number of
power sweep steps is $N_{\text{FTS}} = 2 \cdot \text{BW} / \delta\lambda_{\text{FTS}}$. When N_{FTS} exceeds this critical value, the Fourier map will be fully
unfolded, as shown in Figure R9b. As the wavelength increases, the “vertical” location of a spot in the Fourier

Figure R9 Folding of the Fourier map. Illustrations of an (a) arbitrary input spectrum and the (b-c) fast Fourier transform (FFT) of output interferograms with different power sweep steps (N_{FTS}). When the number of sweep steps is chosen as $N_{\text{FTS}} = 2 \cdot \text{BW} / \delta\lambda_{\text{FTS}}$, the locations of spots are fully unfolded along f_{FTS} , allowing for an ideal point-to-point mapping. The reduction of sweep steps will lead to two dislocated trajectories in a folded Fourier map. The similarity between fringes can be categorized into three cases (i.e., i-iii). (d-f) Intensity and phase maps of FFT results at different wavelengths (λ) derived from the measured transmittance cube.

map oscillates between $f_{\text{SHS}} = 0$ and $1/2$. In the meantime, its “lateral” location shifts from $f_{\text{FTS}} = 1/2$ to 0 . Here, the direction of spot shift is determined by the initial phase of the 2D-FTS at the first sampling. When fewer sweep steps (i.e., $N_{\text{FTS}} < 2 \cdot \text{BW} / \delta\lambda_{\text{FTS}}$) are utilized, the Fourier map will become folded, and a spot will undergo an additional backtracking. To be specific, the spot shifts from $f_{\text{FTS}} = 1/2$ to 0 when the wavelength ranges from minimum (λ_{min}) to medium (λ_{mid}), and then rebounds from $f_{\text{FTS}} = 0$ for the remaining bandwidth (i.e., from λ_{mid} to λ_{max}). Consequently, the Fourier map contains two zig-zag trajectories, as shown in Figure R9c. Due to the “mirror flip” at $f_{\text{FTS}} = 0$, the forward and backward trajectories are inverted and dislocated,

making it possible to obtain decorrelation between the folded spots with similar f_{FTS} . Such a phenomenon
 is unique to 2D-FTSs. For 1D-FTSs, any folding of a 1D Fourier map will cause unsolvable obscurity between
 channels. Next, we will demonstrate the feasibility of reconstructing spectra from a folded 2D Fourier map
 with a reduced N_{FTS} . From Figure R9c, the origin of inter-channel correlations is categorized into three cases:

 **Figure R10 Correlation analysis.** Correlation matrices derived from the measured cube with power sweep steps of (a) $N_{\text{FTFS}} = 25$ and
 (b) $N_{\text{FTFS}} = 50$. The left panels show the correlations $[\rho(\cdot, \cdot)]$ between the fringes (a_i) at different wavelengths. The right panels show the
 correlations of the fast Fourier transform (FFT) of fringes (\tilde{a}_i) with zero-frequency components removed. The green and blue arrows
 show the high-correlation elements with $i \approx j$ and $i + j \approx \text{Const}$, respectively. Here, i and j denote channel indices, and Const is a constant
 related to the channel index at $f_{\text{FTS}} = 0$. The dashed lines represent two slices of the matrix at $\lambda \approx 1.559 \mu\text{m}$ and $1.511 \mu\text{m}$. Correlations
 between the fringes at $\lambda \approx 1.559 \mu\text{m}$ and other wavelengths (c) before and (d) after the component removal. Correlations between the
 fringes at $\lambda \approx 1.511 \mu\text{m}$ and other wavelengths (e) before and (f) after the component removal. The left and right panels show the results
 with $N_{\text{FTFS}} = 25$ and 50 , respectively. Intensity maps of FFT results at (g) $\lambda \approx 1.511 \mu\text{m}$ and (h) $\lambda \approx 1.539 \mu\text{m}$ with $N_{\text{FTFS}} = 50$. The dashed
 line represents $f_{\text{FTS}} = 0.25$. (i) Singular values (σ_i) derived from the measured transmittance cubes with $N_{\text{FTFS}} = 25$ and 50 . (j) σ_i curves
 derived from the oversampled cubes. The dashed line represents the location of the kink.

- (i) Two spots have similar f_{FTS} but different f_{SHS} ;
- (ii) Two spots have similar f_{SHS} but different f_{FTS} ;
- (iii) Two spots locate closely at the crossover of two trajectories.

Some examples are provided in Figures R9d-R9f. For instance, at the wavelengths of $\lambda \approx 1.559 \mu\text{m}$ and 1.488
 450 μm , the residual correlations mainly result from their similar projections on the f_{FTS} axis. In the second case,
 the channels at $\lambda \approx 1.559 \mu\text{m}$ and $1.615 \mu\text{m}$ have similar projections on the f_{SHS} axis. Figure R9f shows two
 spots at the crossover point with both similar f_{FTS} and f_{SHS} (i.e., type **iii**). Hence, it is challenging to directly
 read out a spectrum from the DCT map with $N_{\text{FTS}} < 2 \cdot \text{BW} / \delta\lambda_{\text{FTS}}$, as some spots are closely located. Next, we
 will give a rigorous analysis on the correlation in a folded 2D Fourier map and show the feasibility of solving
 this issue with the proposed numerical method.

Figures R10a and R10b show the correlation matrices derived from the measured cube with $N_{\text{FTS}} = 25$
 ($< 2 \cdot \text{BW} / \delta\lambda_{\text{FTS}}$) and 50 ($> 2 \cdot \text{BW} / \delta\lambda_{\text{FTS}}$), respectively. In each figure, we compare the correlation properties
 before and after the removal of zero-frequency components. With $N_{\text{FTS}} = 50$, the original correlation matrix
 contains a shade in the vicinity of diagonal elements ($i \approx j$, see the blue arrow), which is caused by the overlap
 of projections of adjacent spots on the f_{FTS} axis in the Fourier map. In addition, a “noisy” background can be
 found over the non-diagonal region, which results from the similar projections of different spots on the f_{SHS}
 axis in the Fourier map (type **ii** in Figure R9c). Since both cases result from the projection effect, correlations
 can be inhibited by omitting the zero-frequency components. After the component removal, the correlation
 matrix becomes diagonal, as shown in the right panel of Figure R10a. If the sweep steps are reduced to N_{FTS}
 $= 25$, an additional shade will appear in the correlation matrix (see the green arrow in Figure R10b). Such a
 shade is caused by the similar projections of folded trajectories on the f_{FTS} axis in the Fourier map (type **i** in
 Figure R9c). This is evidenced by the fact that each element in this shade has a virtually constant sum of row
 and column indices, i.e., $i + j = \text{Const}$, corresponding to the folding point at a medium wavelength. These high-
 correlation elements can be eliminated via component removal, as shown in the right panel of Figure R10b.
 However, some correlations still remain in the cube after the removal. For clarity, the correlation matrix is
 sliced at two typical wavelengths. The right panel of Figure R10c shows the correlations between $\lambda \approx 1.559$
 472 μm and other wavelengths with $N_{\text{FTS}} = 25$. The near-unity peak is associated with the self-correlation. The
 473 bump around the self-correlated peak (see the blue arrow) relates to the shade with $i \approx j$. The bump at $\lambda \approx$
 $1.488 \mu\text{m}$ (see the green arrow) is induced by the type-**i** correlation. Other peaks (e.g., $\lambda \approx 1.615 \mu\text{m}$) are tied
 to the type-**ii** correlation. The linewidths of bumps are wider than peaks due to the “slower” shift along f_{FTS}
 in the Fourier map. For most wavelengths, all the superfluous bumps and peaks can be eradicated after the
 removal operation, as shown in the right panel of Figure R10d. However, the removal operation is applicable
 solely in cases **i** and **ii**. In the right panel of Figure R10e, we show the correlation function sliced at $\lambda \approx 1.511$
 479 μm . The peak resides on a bump at $\lambda \approx 1.539 \mu\text{m}$ (see the green arrow), suggesting that this wavelength is
 480 mapped to a crossover in the Fourier domain (type **iii** in Figure R9c). Such a high correlation cannot be fully
 depressed by omitting the zero-frequency components (see the right panel of Figure R10f). As discussed in

Figure R9b, the type-iii correlation issue can be addressed by increasing the number of sweep steps. The
left panel of Figure R10f shows the correlation function sliced at $\lambda \approx 1.511 \mu\text{m}$ with $N_{\text{tFTS}} = 50$. By comparison,
the peak at $\lambda \approx 1.539 \mu\text{m}$ diminishes, leaving only the self-correlated peak at $\lambda \approx 1.511 \mu\text{m}$. In Figures R10g
and R10h, we show the obtained Fourier maps at $\lambda \approx 1.511 \mu\text{m}$ and $1.539 \mu\text{m}$. Compared to the results shown
in Figure R9f, two spots are laterally shifted to $f_{\text{tFTS}} < 0.25$ and > 0.25 . It should be noted that, with $N_{\text{tFTS}} = 25$,
the majority of correlations are sufficiently inhibited and there are merely ≈ 5 ($< N_{\text{tFTS}}/2$) pairs of channels
suffering from the crossover problem. Moreover, even at a crossover point, the correlation is still limited to
$\rho(\tilde{\mathbf{a}}_i, \tilde{\mathbf{a}}_j) \approx 0.5$ after component removal, as shown in the right panel of Figure R10f. This is the result of two
factors: first, due to spectral leakage (see Note 8), each channel occupies multiple rather than a single pixel
in the 2D Fourier map, and two spots do not precisely coincide at a crossover; and second, the difference in
phase distributions of FFT results also contributes to the improvement of decorrelations (see the 2nd and
4th columns of Figure R9f), as both intensity and phase information are useful in numerical reconstruction.
To assess the solvability, we calculate the singular values (σ_i) of the measured cubes with $N_{\text{tFTS}} = 25$ and 50,
as shown in Figure R10i. The decay rates of σ_i curves are quite close, validating that reducing N_{tFTS} will not
cause notable degradation in decorrelation. We also calculate the σ_i curves when the cubes are oversampled
into > 3000 channels. The locations of kinks do not change, which is a direct proof that the channel capacity
is not affected. From above analysis, it is conclusively demonstrated that sufficient decorrelation has been
established at $N_{\text{tFTS}} = 25$ and that all fringes can be numerically identified. It is viable to further reduce N_{tFTS}
to support an even shorter sampling period, as long as the Fourier map is folded only once (see Figure R9c)
and the following condition is met:

$$502 \quad \frac{2 \cdot \text{BW}}{\delta\lambda_{\text{tFTS}}} > N_{\text{tFTS}} \geq \frac{\text{BW}}{\delta\lambda_{\text{tFTS}}}. \quad (\text{R19})$$

In this work, we choose to use $N_{\text{tFTS}} = 25$ to leave a margin and improve robustness, as the selected value is
already quite small. When scaling to a higher dimension, the type-iii correlation can also be fully eliminated
since a high-dimensional space provides a higher degree of freedom in arranging wavelength channels and
circumventing crossover between spots. This issue is detailed in our response to **Comment 2-2**.

In the revised manuscript, we have added the following sentences:

“According to Equation 3, to obtain a point-to-point mapping (see Figure 1d), the required number of
power sweep steps is $2 \cdot \text{BW} / \delta\lambda_{\text{tFTS}}$. Notably, the 2D FTS has the potential to reduce sweep steps to $<$
$2 \cdot \text{BW} / \delta\lambda_{\text{tFTS}}$ using the numerical method since, unlike 1D-FTSs, the folding of a Fourier map only leads to a
limited increase in correlation between channels. This issue will be discussed later.” (**Design principle**, lines
136-139)

“Using Equation 1, the maximum heating power is set as $P_{\text{max}} = 2.4 \text{ W}$ to identify all free spectral ranges
of the SHS. The resolution of the tFTS reaches its minimum ($\delta\lambda_{\text{tFTS}} \approx 12.36 \text{ nm}$) at $\lambda = 1.45 \mu\text{m}$, which yields
the number of power sweep steps of $N_{\text{tFTS}} = 32$. Nevertheless, due to the dislocation of spot trajectories in a

folded 2D Fourier map, the sweep steps can be further reduced without compromising much reconstruction
accuracy. In this work, the optimal number of sweep steps is set as $N_{\text{FSTS}} = 25$ (see Note 6)."

**(Characterization and analysis of the spectrometer, lines 196-201)**

"The arrow highlights the remnant non-diagonal elements with relatively high correlations."

**(Characterization and analysis of the spectrometer, caption of Figure 4, lines 227-228)**

"Some elements in the correlation matrix still have relatively high values, which results from the
crossover of spot trajectories in the folded Fourier map. Throughout the entire capacity, there are only ≈ 5
($< N_{\text{FSTS}}/2$) pairs of less decorrelated fringes. The residual correlation can be eliminated by increasing the
power sweep steps to $> 2 \cdot \text{BW}/\delta\lambda_{\text{FSTS}}$ and unfolding the Fourier map; however, this will result in an increase
in the acquisition period. By using the numerical method, it is viable to identify these fringes without
increasing N_{FSTS} or reconstruction errors, provided that the Fourier map is folded only once (i.e., $N_{\text{FSTS}} >$
$\text{BW}/\delta\lambda_{\text{FSTS}}$) and their correlations [$\rho(\tilde{\mathbf{a}}_i, \tilde{\mathbf{a}}_j) \approx 0.5$] are still quite limited. Further discussions can be found in
**Supplementary Information, Figure S9**. The effectiveness of this operation can be verified using singular
value decomposition (SVD)." **(Characterization and analysis of the spectrometer, lines 248-256)**

"From Figure S5c, $\delta\lambda_{\text{FSTS}}$ increases at a longer wavelength. From the minimum resolution at $\lambda = 1.45 \mu\text{m}$,
we derive the critical number of sweep steps as $N_{\text{FSTS}} = 32$. Notably, due to the unique folding property of 2D
Fourier map, it is feasible to reduce the sweep steps to $< 2 \cdot \text{BW}/\delta\lambda_{\text{FSTS}}$ while maintaining sufficient
decorrelation and a high reconstruction accuracy, as will be discussed later. In this work, the number of
power sweep steps is optimized to be $N_{\text{FSTS}} = 25$." **(Supplementary information, Note 6, lines 314-318)**

"The origin of correlations is more complex when N_{FSTS} is reduced to $< 2 \cdot \text{BW}/\delta\lambda_{\text{FSTS}}$, as will be discussed
later." **(Supplementary information, Note 6, lines 383-384)**

We have also modified some descriptions regarding 2D Fourier mapping:

"All wavelengths are mapped to a cluster of spots in the 2D Fourier map beyond the free-spectral-range
limit." **(Abstract, lines 13-14)**

"The intensity information of a continuous spectrum (**S**) is encoded by a cluster of spots in **FFT(O)**."
**(Design principle, caption of Figure 1, lines 86-87)**

"When the Fourier map is unfolded along f_{FSTS} (see Figure S8), the locations of spots will be distinct at
varying λ , as discussed in Figure 1d." **(Supplementary information, Note 6, lines 378-379)**

All the discussions and figures in this section have been added to **Supplementary information, Note**
6. A higher degree of decorrelation can be achieved using higher-dimensional FTS (HD-FTS). We invite the
reviewer to read our reply to **Comment 2-2**, in which the technical details of HD-FTS are discussed.

**Comment 1-3 (about phase maps):**

Figure 4, in particular the FFT phase is not commented nor described if not with a bare introduction and no
explanation. Spectral analysis heavily relies on phase, which is the component carrying most of the
information and is often overlooked. This pattern is not intuitive at all and, if shown, needs to be commented.

Can the authors give a meaning to such a figure?

*Reply and modifications:*

As stated, in this work, the reconstructed is realized using the following formula:

$$554 \mathbf{S} = \underset{\mathbf{S}}{\operatorname{argmin}} \left(\left\| \tilde{\mathbf{A}}_{\text{FFT}} \mathbf{S} - \text{FFT}(\tilde{\mathbf{O}}) \right\|_2^2 + \zeta_1^2 \left\| \mathbf{D}_2 \mathbf{S}_1 \right\|_2^2 + \zeta_2 \left\| \mathbf{D}_1 \mathbf{S}_2 \right\|_1 \right). \quad (\text{R20})$$

In Equation R20, both transmittance cube (i.e., \mathbf{A}) and recorded interferogram (i.e., \mathbf{O}) are transformed into
the Fourier space. Notably, $\tilde{\mathbf{A}}_{\text{FFT}}$ and $\text{FFT}(\tilde{\mathbf{O}})$ are complex cube (or vector) with both real and imaginary parts,
as they are transformed via FFT rather than DCT. Apparently, both intensities and phases of $\tilde{\mathbf{A}}_{\text{FFT}}$ and $\text{FFT}(\tilde{\mathbf{O}})$
are necessary in solving Equation R20. The phase and intensity maps of $\text{FFT}(\mathbf{O})$ are separately displayed in
Figures 4f and 4g simply because it is impossible to depict a complex matrix with a single plot. As discussed
in our reply to **Comments 1-1**, when using DCT, the transformed interferogram becomes a real map, without
imaginary part or phase information. The intensities of a FFT map are quite similar to the DCT distribution
(see Figures R1d and R1e for instance), indicating that the phase map carries some information, but not the
majority. According to the correlation analysis (see our reply to **Comment 1-2**), phase information improves
the decorrelation between channels located at the crossover point in a folded Fourier domain.

In the revised manuscript, we have added the following sentence:

**“Both intensities and phases of $\text{FFT}(\mathbf{O})$ are necessary in the spectrum reconstruction to support the**
**full capacity of $2N_f$, as discussed in **Supplementary Information, Figure S8f**” (**Characterization and**
**analysis of the spectrometer**, lines 266-267)**

Other modifications have been specified in our reply to **Comment 1-2**.

**Comment 1-4 (about waveguide dispersions):**

Is waveguide dispersion compensated for during data analysis? I see that it is considered in the
**Supplementary information**, especially referring to the delay line, but for instance, in the SHS there are
many bends that will introduce further dispersion besides that of a straight waveguide.

*Reply and modifications:*

Again, in this work, any arbitrary spectrum is recovered by solving a linear inverse problem:

$$576 \mathbf{S} = \underset{\mathbf{S}}{\operatorname{argmin}} \left(\left\| \mathbf{A} \mathbf{S} - \mathbf{O} \right\|_2^2 + \Omega \right). \quad (\text{R21})$$

A spectrum (i.e., \mathbf{S}) is retrieved from the recorded interferogram (i.e., \mathbf{O}) and calibrated transmittance cube
(i.e., \mathbf{A}). Each column of the calibrated cube is a fringe pattern at a specific wavelength. Hence, the dispersion
information of waveguides has already been contained in the transmittance matrix, and extra compensation
of dispersions is unnecessary. The calculated dispersions of effective indices and group indices in Figure R1
are solely for numerical modeling.

Figure R11 Bends in the spatial heterodyne spectrometer (SHS). The upper and lower arms of the SHS have the same number of bends, so their effective optical path lengths (OPL) will cancel each other out.

In addition, the upper and lower arms of the SHS have the same number of bends, as shown in Figure R11, so their effective optical path lengths (OPL) will cancel each other out. Consequently, the effective group delay is dependent solely on the length of the straight sections.

In the revised **Supporting information**, we have added the following sentences:

“The simulation results are derived from a straight waveguide, provided that each SHS has the same number of bends in two arms and its dispersion is provided solely by the straight sections. In addition, for the high-contrast SOI waveguide, the variation of dispersion is minimal at a modest bending radius of > 10 μm in the design.” (Supplementary information, Note 3, lines 178-181)

“Compared to DCT, the pseudo-inverse method does not require ideal sinusoidal responses or any phase compensation since the dispersions of FSRs and ERs have been incorporated into the calibrated matrix/cube.” (Supplementary information, Note 8, lines 590-592)

Comment 1-5 (about coupler measurement):

Figures 2h and 2l describe the characterization of coupler and Y-splitter. The experimental traces deviate from the simulated ones with additional features, for instance, there seem to be some kind of weak resonances in the couplers’ traces, is there an explanation as to why those features appear? What is their origin? Similar for the oscillations in the Y-splitters traces, which in the supplementary material seem to be attributed to the reflections at the grating couplers. Can the authors prove this? Do they have a reference trace for a straight waveguide? Also, the tolerance analysis does not seem to really support the statement that “coupling ratios are virtually constant”. Was some kind of statistical study done on these components since there are so many in their device?

Reply and modifications:

Some Mach-Zehnder interferometers (MZI) were fabricated to characterize the coupling ratios of the adiabatic directional coupler (ADC) and Y-branch splitter (YBS), as shown in Figures 2g and 2k. The coupling ratios ($|\kappa|^2$) can be derived from the extinction ratios (ER) of the MZI interference curve^{R25}:

$$|\kappa|^2 = \frac{1}{2} \pm \frac{1}{2} \sqrt{\frac{1}{10^{ER}}} \tag{R22}$$

Figure R12 Explanation on ripples. (a) Illustration of a testing Mach-Zehnder interferometer (MZI) with defects in one interference arm. (b) Calculated transmittance ($|t|^2$) curve of the MZI with $L_{FP} = 30 \mu\text{m}$ and $|r|^2 = 0.03$.

Some ripples appear in the obtained coupling-ratio curve, as shown in **Supplementary information**, Notes 4 and 5. For the coupling ratios for the ADC, these additional features mainly results from the measurement inaccuracy at a low level of received light power. This is evidenced by the fact that ripples become stronger at $|\kappa|^2 \approx 0.5$, as a coupling ratio of 0.5 leads to an infinite ER. On the other hand, the coupling-ratio curve of the YBS has variations with a nearly uniform period of $\approx 9 \text{ nm}$. In the original manuscript, this phenomenon was attributed to the reflection at input/output grating couplers. However, after meticulous investigation, we found that the ripples may actually result from the fabrication flaws in the testing MZI rather than grating couplers. Here, we will provide a simulation result to support this viewpoint.

According to the tolerance analysis (see Figures S3d and S3e), the deviations in YBS parameters cannot induce such a periodicity. In addition, the periodic variation in ERs is not due to the reflection at two grating couplers since the period would be much smaller (close to the MZI's FSR) if that were the case. We assume that the ripples may result from the fabrication defects in the interference arm. For instance, if there are two

Figure R13 Characterization of couplers. Microscope images of (a) cascading adiabatic couplers (ADC) and (c) cascading Y-branch splitters (YBS). Measured coupling ratios of the (b) ADC and (d) YBS. The coupling ratios measured with Mach-Zehnder interferometers (MZI) are also plotted for comparison.

defects (e.g., sidewall discontinuity or air void) in one arm, the slight reflection (denoted as $|r|^2$) will induce
 a weak Fabry-Perot (FP) resonance that sinusoidally varies the interference contrast and ERs, as illustrated
 in Figure R12a. To verify this, we calculate the dispersion curves of the MZI under defects, as shown in Figure
 R12b. Here, the spacing between defects is set as $L_{FP} = 30 \mu\text{m}$ and their reflectivity is set as $|r|^2 \approx 0.03$. Similar
 periodicity ($\approx 9 \text{ nm}$) can be observed from the curve. These defects are fully stochastic and the transmission
 of most MZIs in the SHS does not have noticeable ripples in their ERs (see Figures 3e and 4a).

From above analysis, the additional features in the measured coupling-ratio curves mainly result from
 measurement inaccuracy or imperfection of testing MZIs. To validate the uniformity of coupling ratios, some
 testing structures with 1 ~ 9 cascading ADCs or YBSs were fabricated on the same chip, as shown in Figure
 R13a and R13c. The output port of each coupler was connected to the input port of the preceding coupler.
 The coupling ratios can be thus derived by averaging the transmittance difference between 1 and 9 couplers.
 Figure R13b and R13d compare the measured coupling-ratio curves and MZI results. The results measured
 from the cascading structures exhibit negligible undulations, verifying that the ripples in Figure 2h and 2l
 are not caused by the couplers.

In the revised Supplementary information, we have added the following sentences:

“Figure 2g shows the measured interference curve of the MZI, from which the coupling ratios ($|\kappa_{\text{ADC}}|^2$)
 can be derived with the following formula:

$$646 \quad |\kappa_{\text{ADC}}|^2 = \frac{1}{2} \pm \frac{1}{2} \sqrt{\frac{1}{10^{\text{ER}}}}, \quad (\text{S6})$$

where ER denotes the extinction ratio of the coupling-ratio curve.” (Supplementary information, Note 4,
 lines 216-219)

“The ripples in the curve results from the noise in the measurement of weak signals.” (Supplementary
 information, Note 4, lines 221-222)

“As a cross reference, some end-to-end cascading structures were also fabricated on the same chip (see
 Figure S2f). Here, we mainly consider the transmission at the port #1, while the port #2 was connected to a
 tapered waveguide to prevent reflection. The coupling ratio at the port #1 can be derived by averaging the
 difference in the transmittances of 1 and 9 cascading ADCs, as shown in Figure S2g. The measured $|\kappa_{\text{ADC}}|^2$
 agrees well with the result obtained from the MZI and has fewer disturbances, demonstrating that the
 ripples in Figure 2h result mainly from measurement inaccuracy.” (Supplementary information, Note 4,
 lines 224-230)

“The $|\kappa_{\text{YBS}}|^2$ curve measured in Figure 2l contains ripples with a uniform period of $\approx 9 \text{ nm}$, which could
 be the consequence of fabrication defects in one arm of the MZI (for explanations, see Figure S4). To validate
 the uniformity of $|\kappa_{\text{YBS}}|^2$, some testing structures with 1 ~ 9 cascading YBSs were fabricated, as shown in
 Figure S3f. The input port of each YBS was connected to the port #1 of the preceding YBS, while the port #2
 was left unused and connected to a short taper. Thus, the transmission at the port #1 can be derived by

averaging the transmittance difference between 1 and 9 YBSs. Figure S3g shows a comparison between the
measured $|k_{YBS}|^2$ curve and MZI result. The result measured with the cascading structures shows negligible
variations, demonstrating that the ripples in Figure 2l are not caused by the YBS.” (Supplementary
information, Note 5, lines 256-263)

The results shown in Figure R13 have been added to Supplementary information, Figures S2 and S3.
Figure R12 and the corresponding discussions have been added to Supplementary information, Figure S4
and its caption.

**Comments 1-6 (about spectrometer measurement):**

In **Methods** section, regarding measurements, it is not clear how the experiment was conducted. Are the
two inputs used at the same time, with a single lensed fiber? How can this give good coupling? What is the
working distance of the fiber and its mode field diameter?

Line 323 “The transmission at 128 output ports was individually measured”. This means the fiber needs
to be moved across the outputs, which in turn requires to make sure to have good coupling and stable output.
How was this done? How long does this take? Is it done manually or with an automatic feedback system of
some sort? The inputs seem to be separated by 5-10 μm . Was crosstalk between ports considered at the
output? A spectrometer is useful if it can acquire and process data in a fast and reliable way, ideally seconds.
Naturally, for an integrated spectrometer, one cannot expect to have a measurement lasting as short as a
commercial bulky system would do, but this sounds a long and cumbersome procedure. The statement that
the procedure “ensures precise data acquisition” sounds odd to me and is not enough to claim stable
operation.

*Reply and modifications:*

The reviewer’s comments involve two aspects: the edge-coupling method and multi-port measurement.

A lensed fiber was used to launch light into the two-dimensional Fourier-transform spectrometer (2D-
FTS). There are two input ports in the fabricated 2D-FTS (denoted as IN_1 and IN_2 , see Figure 1a). Light was
injected at IN_1 and collected at 128 output ports (denoted as OUT_{1-128}). The dummy port IN_2 is designed for
monitoring the tunable FTS. IN_2 was not used during measurement. The edge couplers are based on inverse
tapers with an effective spot diameter of $\approx 3 \mu\text{m}$. The spacing between adjacent output ports is $> 15 \mu\text{m}$. The
lensed fiber was positioned in close proximity to the output facet ($< 1 \mu\text{m}$, almost touching it). The optical
crosstalk was measured to be $< -40 \text{ dB}$, as the spot diameter is sufficient smaller than the channel spacing.
The measured end-to-end coupling efficiencies can be found in Supplementary information, Figure S1h.

In the revised manuscript, we have modified and added the following sentences:

“Light is injected at IN_1 and collected at OUT_{1-128} , while IN_2 and OUT_0 are utilized for monitoring the
tFTS.” (Design principle, lines 93-94)

“Each edge coupler is an inverse taper with an effective spot diameter of $\approx 3 \mu\text{m}$. The spacing between

output ports is set as $> 15 \mu\text{m}$ to prevent inter-channel optical crosstalk.” (Characterization and analysis
 of the spectrometer, lines 209-210)

“In the measurement of the 2D-FTS, IN_1 and OUT_{1-128} were used as input and output, respectively, while
 IN_2 and OUT_0 were dummy ports for the in-situ monitoring of the tFTS.” (Measurement details, lines 369-
 370)

**Figure R14 Multiport measurement.** (a) Illustration of the relation between the spectrum (\mathbf{S}), interferogram (\mathbf{O}), and transmittance
 cube (\mathbf{A}). The i -th row vector in \mathbf{O} (i.e., $\mathbf{o}_{\text{row},i}$) is just the multiplication of i -th row slice in \mathbf{A} (i.e., $\mathbf{a}_{\text{row},i}$) and the spectrum vector (i.e. \mathbf{S}).
 (b) Illustration of the multiport measurement process. The output fiber was aligned to the i -th output port at the i -th step. The i -th row
 of the interferograms (i.e., $\mathbf{o}_{\text{row},i}$) for all testing spectra and the i -th row of the transmittance cube (i.e., $\mathbf{a}_{\text{row},i}$) were successively measured
 with varying heating power. Spec., spectrum.

The proposed 2D-FTS has 128 output physical ports, and their responses were individually measured.
 The key is to acquire the transmittance cube and interferograms in succession, rather than separately. Here,
 we first discuss the reliability of this method. The i -th row vector of the interferogram (denoted as $\mathbf{o}_{\text{row},i}$) and
 the i -th row slice of the transmittance cube (denoted as $\mathbf{a}_{\text{row},i}$) are captured at the i -th output port (i.e., OUT_i),
 as shown in Figure R14a. Moreover, $\mathbf{o}_{\text{row},i}$ is just the multiplication of $\mathbf{a}_{\text{row},i}$ and the spectrum vector (denoted
 as \mathbf{S}). Therefore, the measurement is viable as long as each $\mathbf{o}_{\text{row},i}$ and $\mathbf{a}_{\text{row},i}$ have an accurate mapping relation.
 At the i -th measurement step, we aligned the output fiber to OUT_i . The i -th row of the interferograms for all

testing spectra and the i -th row of the transmittance cube were *successively* captured with varying power, as
shown in Figure R14b. The input/output fibers were *not* relocated during each step. The used commercial
optical sources and filters are highly stable. The whole setup was connected with fibers and controlled by a
host computer. The only manual operation required is the relocation of the output fiber from i -th to $(i + 1)$ -
th step. Each step, including the acquisitions of the cube and interferograms, was completed in few minutes.
The edge-coupling efficiency maintains stable within such a short period, ensuring the accurate mapping
between $\mathbf{o}_{\text{row},i}$ and $\mathbf{a}_{\text{row},i}$. The high reconstruction accuracy shown in Figure 5 also validates the effectiveness
of this method.

The individual measurement of a multiport device has been widely applied in prior research on spatial-
heterodyne spectrometers (SHS^{R26R,27}) and speckle spectrometers^{R28,R29}. In this work, such a method is used
only as a proof of concept. Nevertheless, as the reviewer pointed out, the proposed method is not applicable
to practical multiport acquisition. A straightforward solution to this problem is to monolithically integrate
photodetectors at each output port and acquire the multi-stream signals from the PDs under a synchronized
clock. We also invite the reviewer to read our reply to **Comment 2-1** or **Supplementary information, Note**
10, in which we have proposed a modified 2D-FTS scheme that only requires a single output port.

In the revised manuscript, we have added the following sentences:

*“The input fiber was aligned to IN_1 , and the transmission responses at OUT_{1-128} were individually*
*measured. At the i -th measurement step, we aligned the output fiber to OUT_i . The i -th row of interferograms*
*(**O**) from all testing spectra and the i -th row of the transmittance cube (**A**) were successively recorded with*
*varying heating power. The input/output fibers were not relocated during the measurement of each port,*
*ensuring an accurate mapping between **A** and **O**. This proof-of-concept method has been used in prior works*
*on SHS²³ and speckle spectrometers⁴⁹. Practical multiport acquisition can be accomplished by using*
*monolithically integrated PDs.” (Measurement details, lines 370-376)*

**Comment 1-7 (about introductory overview):**

In line 51, it is said that SWIFTS devices cannot be integrated with monolithic PDs. This is partially true, but
there has been a recent publication [*Nat Photon* **17**, 59-64 (2023)] showing that electrical readouts are
possible on SWIFTS and this, although the field is not yet mature, opens up a whole new set of opportunities
and greatly boosts the potential of SWIFTS. This should be taken into account, as the pixel size of the camera
does not limit the bandwidth that much anymore. Second, SWIFTS have been shown to have among the
largest bandwidths for integrated spectrometers, so the statement is incorrect.

*Reply and modifications:*

We thank the reviewer for his/her careful reading and useful advice. The mentioned research has been
cited in the revised manuscript:

*“33. Grotevent MJ, et al. Integrated photodetectors for compact Fourier-transform waveguide*

spectrometers. *Nat Photon* **17**, 59-64 (2023).” (**References**, lines 464-465)

In the revised manuscript, the following sentence has been modified:

“The SWIFTS has two main disadvantages: first, it is still challenging to probe the field of a guided mode
even with monolithic PDs³³; and second, it requires a dense sampling of a stationary wave to attain a large
capacity in a SWIFTS.” (**Introduction**, lines 50-52)

**Comments 1-8 (about language and figures):**

**(i)** The paper is hard to follow due to the choice of referring constantly to quantities labelled at the beginning
of the manuscript (**A**, **S**, **O**, etc.). This makes it difficult to read and fully follow the reasoning.

**(ii)** In line 88, probably the authors wanted to write “as long as the resolution...”.

**(iii)** Why changing the scale from “dB” and “a.u.”? Spectra can be rescaled to any peak value with this
approach.

**(iv)** The spectra in Figure 5b look a bit superficial.

**(v)** In line 273, I guess the authors made a typo and wanted to write “pm” instead of “nm”.

*Reply and modifications:*

We thank the reviewer for his/her careful reading and useful advice. Our replies are as follows:

**(i)** In the revised manuscript, we have replaced some of the notations (e.g., **A**, **S**, and **O**) with words:

“At a single wavelength, the interferogram becomes a pattern that is sinusoidally modulated in two
dimensions.” (**Introduction**, lines 110-111)

“The recorded interferogram (**O**) is a linear combination of fringes (\mathbf{a}_i), with the weight on \mathbf{a}_i indicating
the spectral intensity at the i -th wavelength.” (**Introduction**, lines 115-116)

Not all notations are replaced since some terminologies can be quite redundant. To improve readability,
we have added Note 2 to **Supplementary information**, in which all abbreviations and notations used in the
main manuscript are summarized.

**(ii)** This typo has been corrected in the revised manuscript:

“Any spectrum can be retrieved via decomposition, as long as the resolution ($\delta\lambda_{\text{tFTS}}$) of the tFTS is finer
than the free spectral range (FSR_{SHS}) of the SHS, thereby breaking the inherent limit between the resolution
($\delta\lambda_f$) and bandwidth (BW).” (**Design principle**, caption of Figure 1, lines 88-90)

**(iii)** In unit of “a.u.”, more subtle details can be observed (see Figure 5f), whereas in unit of “dB”, signal-
to-noise ratios can be plainly displayed (see Figure 5c). In the revised **Supplementary information**, Note
9, we have a figure with all spectra plotted in “a.u.”. The normalized deviations have also been calculated in
this figure, in order to improve the visibility of errors. For more details, please find Figure R8 in our reply to
**Comment 1-1.**

**(iv)** It is a bit confusing why the reviewer found the spectra in Figure 5b (originally 4b) superficial. We
presume that it is because the reference and reconstructed spectra are not shown in their entirety. For clarity,

only a small part of the spectrum is displayed in Figure 5b, but the reconstruction is implemented over the
whole bandwidth. In the revised manuscript, we have modified the following sentence:

“Only a small part of the spectrum is displayed for clarity, but the reconstruction is performed over the
entire bandwidth.” (Spectrum reconstruction, lines 305-306)

The full spectra can be found in Supplementary information, Note 9.

(v) This typo has been corrected in the revised manuscript:

“An enhanced resolution of $\delta\lambda_{2f} = \delta\lambda_f/2 < 125$ pm is thus demonstrated, with the corresponding capacity
of $N_{2f} = 2N_f = 1601$.” (Spectrum reconstruction, lines 307-308)

**To Reviewer #2's comments:**

**General remarks:**

This paper presents a novel Fourier-transform spectrometer (FTS) approach that combines two popular
implementations of FTS: tunable FTS and spatial heterodyne FTS. The principle is clearly explained, and the
characterization is comprehensive. The performance shows visible improvements in terms of bandwidth
and resolution, demonstrating certain novelty and warranting publication. However, considering the lack of
fundamental breakthroughs in the technical or theoretical aspects, I would suggest that the authors consider
submitting to a more specialized journal instead of *Nature Communications*.

*Reply:*

We thank the reviewer for his/her careful reading and useful comments. The reviewer's main concerns
pertain to the concept novelty and performance advance. Prior to engaging in a comprehensive discussion,
we would want to give a concise recapitulation of the progress achieved in this work:

- 1. The primary breakthrough in this work is the progression of the principle from one-dimensional to two-
dimensional Fourier transform, which represents a paradigm shift in the research of Fourier-transform
spectrometry (FTS). For conventional one-dimensional FTS (1D-FTS), efforts have been made to extend
the variation range of group delay. However, the difficulty in extensive refractive-index tuning imposes
an inherent limit between the resolution and bandwidth. Our proposed two-dimensional FTS (2D-FTS),
in contrast, circumvents this limit by mapping any arbitrary spectrum to a 2D Fourier map. For the first
time, we reveal the mathematical connection between cascaded interferometers and high-dimensional
Fourier transform. Remarkably, this scheme provides a flexible framework applicable to all existing FTS
designs, leading to a new group of spectrometer devices.
- 2. We report a record large channel capacity (> 1601) among all integrated FTSS. The signal-to-noise ratio
of > 25 dB and a noise floor of < -35 dB have been experimentally achieved. It is thus demonstrated that
the resolution-bandwidth limit in integrated FTSS can be broken, while preserving a high reconstruction
accuracy.
- 3. The study of 2D-FTS paves the path towards higher-dimensional FTSS (HD-FTS).

In the following sections, we will respond to the technical comments about the Jacquinot's advantage, power
efficiency, and device performance. As the discussions are somewhat lengthy, here, we provide a list of major
viewpoints and related contents in our replies:

- 1. As a proof of concept, the proposed 2D-FTS is formed by a tunable FTS (tFTS) and a spatial heterodyne
spectrometer (SHS) with multiple ports. As a result, the Fellgett's advantage can be maintained, but the
Jacquinot's advantage is *partially* diminished. With 128 physical ports, the etendue is still improved by
around one order of magnitude, compared to a 1601-channel filter. Furthermore, the 2D-FTS is a flexible
design framework, and the etendue issue can be resolved by replacing the SHS with a single-port digital

FTS (dFTS). In addition, by using a Michelson interferometer and etching isolation trenches, the heating
power required by the tFTS can be reduced from $P_{\max} = 2.4$ W to < 120 mW. In our response to **Comment**
**2-1**, we provide detailed simulation results about a modified 2D-FTS design enabling a single port and
low power consumption. This modified scheme is built using the components that are well-established
and demonstrated in the main manuscript.

The related discussions can be found in lines 855-990 and Figures R15-R18.

2. The demonstrated 2D-FTS has a fine resolution of < 125 pm across a 200-nm bandwidth, supporting a
capacity of 1601. As agreed by the reviewer, these results indicate performance advances over previous
FTSs. In addition to FTSs, other schemes, e.g., filters and speckle spectrometers (e.g., Ref. R30 mentioned
by the reviewer), can support comparable resolutions and bandwidths. Our reply to this point involves
two aspects. First, it is imperative to assess signal-to-noise ratios (SNR) in the performance comparison,
as reconstruction accuracy is crucial for spectrometry. The importance of SNRs has been overlooked in
many prior studies. We will show that, compared to filters and speckle spectrometers, FTSs has greater
potential to attain a higher SNR. Second, a finer resolution of < 31.25 pm and a larger capacity of > 6401
can be realized by extending the concept of 2D-FTS to three dimensions. The design of 3D-FTS will be
detailed in our response to **Comment 2-2**. We will also discuss the concept of higher-dimensional FTSs
(HD-FTS).

The related discussions can be found in lines 1009-1073, 1092-1227 and Figures R19-R24.

The modifications to the manuscript are highlighted in red. The critical descriptions are underlined.

**Comment 2-1:**

The tunable FTS and spatial heterodyne spectrometer used in this work share the same principle as previous
demonstrations, and the drawbacks associated to previous demonstrations mentioned by the authors are
still present in their own work. For instance, in line 40, the authors criticize conventional spatial heterodyne
FTS, stating that it reduces the etendue at each port and diminishes the Jacquinot's advantage due to the use
of multiple physical channels, but it is exactly the same situation in this work as the signal needs to be split
into 128 MZIs. Similarly, in line 44, the authors criticize tunable FTS for its power consumption of
approximately 5 W. However, this work still requires a power consumption of 2.4 W, which is at a similar
level. No practical applications can tolerate this amount of power consumption.

*Reply and modifications:*

As a proof of concept, the 2D-FTS is realized by cascading a tunable FTS (tFTS) with $P_{\max} \approx 2.4$ W and a
spatial heterodyne spectrometer (SHS) with 128 physical ports. By comparison, a stand-alone tFTS requires
a heating power of $P_{\max} > 100$ W to attain the same resolution. Also, when using a stand-alone SHS to realize
the same bandwidth, it is essential to integrate > 2000 Mach-Zehnder interferometers (MZI) on a single chip.
Hence, based on the proposed 2D-FTS, a substantial reduction in power consumption and port number has

already been achieved. In addition, even with 128 ports, the demonstrated peak signal-to-noise ratios (> 25
861 dB) and noise floor (< -35 dB) are already superior to most prior results (see our reply to **Comment 2-2**,
as the received etendue is already improved by around one order of magnitude compared to a 1601-channel
filter. However, it is essential to improve power efficiency and signal detectivity of the 2D-FTS when scaling
to a finer resolution and a larger capacity. Notably, the 2D-FTS is a flexible scheme that is not limited to the
combination of a tFTS and a SHS but can be realized with any form of FTS designs. Actually, the power and
etendue issues are readily rectifiable. In the following section, we will provide a modified 2D-FTS scheme to
prove the viability of attaining the same resolution and bandwidth with a single output port and a drastically
reduced drive power (< 120 mW). All components in this modified scheme have been well-established and
demonstrated in the main manuscript.

**Modified two-dimensional Fourier-transform spectrometer (2D-FTS)**

Figure R15a shows the layout of the modified 2D-FTS. The device consists of a tFTS and a digital FTS (dFTS).
The tFTS is formed by a balanced Michelson interferometer (MI) rather than a Mach-Zehnder interferometer
(MZI), as shown in Figure R15b. The use of an MI doubles the optical path length (OPL) change from $\Delta n_g L_{\text{tFTS}}$
to $2\Delta n_g L_{\text{tFTS}}$ since its interference arm is reflective. Here, the arm length is set as $L_{\text{tFTS}} = 1.5$ cm, the same as
in the original design. The tuning efficiency is further enhanced by etching isolation trenches alongside the
tunable waveguide. In the spiral, thermal isolation is inserted between adjacent straight sections (see Figure
R15b), whereas the short bends are not isolated, in order to release structural stress. At each arm, the output
ports of a Y-branch splitter (YBS) are connected to each other to serve as a looper reflector. Two interference
arms are directed to an adiabatic directional coupler (ADC). One port of the ADC is used as input, while the
other one is routed to the dFTS. The dFTS is composed of two YBSs and two interference arms (denoted as
#1 and #2). Each arm is formed by several repeating units with switchable effective OPLs, as shown in Figure
R15c. Each unit has two MZI switches and a pair of asymmetric delay lines between them. The MZI switch
is based on two broadband ADCs and two balanced arms. In the arm #1, the upper delay lines are longer
than the lower delay lines (i.e., $L_{U,i} > L_{L,i}$, where i is an even number), whereas in the arm #2, the upper delay
lines are shorter than the lower delay lines (i.e., $L_{U,i} < L_{L,i}$, where i is an odd number). The lower delay lines
in the arm #1 and the upper delay lines in the arm #2 have a constant length (denoted as L_{ref}), as a reference:

$$887 \quad L_{L,i} = L_{\text{ref}}, \quad \text{mod}(i, 2) = 0, \quad (\text{R23})$$

$$888 \quad L_{U,i} = L_{\text{ref}}, \quad \text{mod}(i, 2) = 1. \quad (\text{R24})$$

For the delay lines cascaded after the i -th switch (denoted as SW_i), the length asymmetry (denoted as $\Delta L_{\text{dFTS},i}$)
is:

$$891 \quad \Delta L_{\text{dFTS},i} = L_{L,i} - L_{U,i} = (-1)^i \cdot 2^{i-1} \Delta L_0, \quad (\text{R25})$$

where ΔL_0 denotes the base length. The number of usable states (denoted as N_{dFTS}) is thus tied to the number

Figure R15 Modified two-dimensional Fourier-transform spectrometer (2D-FTS). Schematic layout of the (a) modified 2D-FTS. The enlarged views of the (b) tunable FTS (tFTS) and (c) a repeating unit in the digital FTS (dFTS). In the modified scheme, the spatial heterodyne spectrometer (SHS) is replaced by a dFTS, in order to ensure single-port detection and reduce the footprint. Furthermore, a Michelson interferometer (MI) instead of a Mach-Zehnder interferometer (MZI) is employed in the tFTS, so that the effective optical path length (OPL) can be doubled. Some isolation trenches are etched alongside the heating region of the spiral waveguides to improve the thermo-optical tuning efficiency. ADC, adiabatic directional coupler. YBS, PD, photodetector. Y-branch splitter. WG, waveguide. SW, switch.

of switches (denoted as N_{sw}):

$$N_{dFTS} = 2^{N_{sw}}. \quad (R26)$$

We impose additional asymmetry of ΔL_0 to arm #1, so the variation range of overall arm-length differences

(denoted as ΔL_{dFTS}) becomes:

$$\Delta L_{dFTS} = \Delta L_{dFTS,min} \sim \Delta L_{dFTS,max} = \Delta L_0 \sim 2^{N_{sw}} \Delta L_0, \quad (R27)$$

where $\Delta L_{dFTS,min}$ and $\Delta L_{dFTS,max}$ respectively denote the minimum and maximum differences. The π phase shift

induced by the switch process can be compensated for with an additional tuning section (not displayed in

the figure).

In such a device, light is modulated by the tFTS and dFTS in succession. The interferogram is thus a 2D pattern with variations of heating power in the tFTS and switch steps in the dFTS. The 2D-FTS response can be depicted as a 3D cube formed by a series of 2D fringes at varying wavelengths. In principle, this scheme is equivalent to the tFTS/SHS scheme proposed in the main manuscript, but it is implemented in a different manner. The modified scheme offers several advantages:

1. Due to the use of the MI and isolation trenches, the power consumption can be reduced to < 120 mW.

2. The dFTS has the capability to support a great number of switch states with a single output port, thereby
 resolving the etendue issue. The increase in N_{dFTS} also contributes to the expansion of FSR of the dFTS
 and the reduction of the heating power required in the tFTS.

3. Due to the folding of light paths in the tFTS and dFTS, the device footprint can be minimized.

Next, we will give the optimization flow of this design. From above analysis, the resolutions of the tFTS
 and dFTS can be expressed as:

$$921 \quad \delta\lambda_{\text{tFTS}} = \frac{\lambda^2}{2\Delta n_g L_{\text{tFTS}}}, \quad (\text{R28})$$

$$922 \quad \delta\lambda_{\text{dFTS}} = \frac{\lambda^2}{n_g \Delta L_{\text{dFTS,max}}}, \quad (\text{R29})$$

where λ denotes the wavelength, $\delta\lambda_{\text{tFTS}}$ denotes the tFTS resolution, $\delta\lambda_{\text{dFTS}}$ denotes the dFTS resolution, n_g
 denotes the group index, and Δn_g denotes the change in n_g . The factor “1/2” in Equation R28 results from
 the light-path folding in the MI. The corresponding FSRs can be written as:

$$926 \quad \text{FSR}_{\text{tFTS}} = \frac{N_{\text{tFTS}} \delta\lambda_{\text{tFTS}}}{2} = \frac{N_{\text{tFTS}} \lambda^2}{4\Delta n_g L_{\text{tFTS}}}, \quad (\text{R30})$$

$$927 \quad \text{FSR}_{\text{dFTS}} = \frac{N_{\text{dFTS}} \delta\lambda_{\text{dFTS}}}{2} = \frac{N_{\text{dFTS}} \lambda^2}{2n_g \Delta L_{\text{dFTS,max}}}, \quad (\text{R31})$$

where N_{tFTS} denotes the number of power sweep steps. From Equations R28-S31, we derive the resolution,
 bandwidth, and critical condition for the modified 2D-FTS:

$$930 \quad \delta\lambda_f = \delta\lambda_{\text{dFTS}} = \frac{\lambda^2}{n_g \Delta L_{\text{dFTS,max}}}, \quad (\text{R32})$$

$$931 \quad \text{BW} = \text{FSR}_{\text{tFTS}} = \frac{N_{\text{tFTS}} \lambda^2}{4\Delta n_g L_{\text{tFTS}}}, \quad (\text{R33})$$

$$932 \quad \delta\lambda_{\text{tFTS}} = \frac{\lambda^2}{\Delta n_g L_{\text{tFTS}}} < \text{FSR}_{\text{dFTS}} = \frac{N_{\text{dFTS}} \lambda^2}{2n_g \Delta L_{\text{dFTS,max}}}, \quad (\text{R34})$$

where $\delta\lambda_f$ denotes the resolution at the Rayleigh criterion, and BW denotes the operation bandwidth. The
 target resolution is $\delta\lambda_f = 250$ pm, which can be enhanced to $\delta\lambda_{2f} = \delta\lambda_f/2 = 125$ pm using the computational
 method. According to Equation R32 and the results shown in Note 6, the maximum arm-length difference is
 chosen as $\Delta L_{\text{dFTS,max}} = 2.55$ mm. Each interference arm in the dFTS contains four stages, yielding the number
 of switch states of $N_{\text{dFTS}} = 2^8 = 256$. Thus, in the design, the unknown parameters are the maximum heating
 power (P_{max}) required to meet Equation R34 and the number of power sweep steps required to achieve the
 target bandwidth (i.e., $\text{BW} = 200$ nm, see Equation R33).

Figure R16 Design of key components. Calculated temperature (T) distributions (a) with and (b) without isolation trenches. Here, the heating power is set as $P = 100$ mW. (c) Calculated tuning efficiencies ($\partial n_g / \partial P$) at varying wavelengths (λ). Here, the $\partial n_g / \partial P$ curves are compared with different heater widths (W_{ht}) and isolation conditions. (d) Calculated coupling ratios ($|k_{sw}|^2$) of the switch at on and off states. (e) Calculated transmittance ($|t|^2$) matrix of the tunable Fourier-transform spectrometer (tFTS) with varying P . (f) Calculated $|t|^2$ dispersions of the tFTS at $P = P_{max}$. The tFTS resolution is $\delta\lambda_{tFTS} = 25.5$ nm. (g) Calculated $|t|^2$ matrix of the digital Fourier-transform spectrometer (dFTS) with varying arm-length differences (ΔL_{dFTS}). (h) Calculated $|t|^2$ dispersions of the dFTS at $\Delta L_{dFTS} = \Delta L_{dFTS,max}$. The dFTS resolution is $\delta\lambda_{dFTS} = 220$ pm.

Two major modifications are made to the tunable waveguide to improve its tunability: first, the width of the heater is reduced from $W_{ht} = 7 \mu\text{m}$ to $2.5 \mu\text{m}$; and second, the heating region is isolated by the trenches in the SiO_2 cladding and the partial undercut in the Si substrate. Figures R16a and R16b show the calculated temperature distributions with the electric power of $P = 100$ mW applied to the heater. It can be found that the energy is condensed with the downsize and isolation of the heating region. The isolation width is set as $W_{iso} = 3 \mu\text{m}$ to ensure an easy fabrication. The undercut technology has been demonstrated in Ref. R31. In Figure R16c, we show the calculated tuning-efficiency ($\partial n_g / \partial P$) dispersions with different W_{ht} and isolation conditions. At the central wavelength, the tuning efficiency is improved from $\partial n_g / \partial P \approx 5.25 \times 10^{-3} \text{ W}^{-1}$ to $29.9 \times 10^{-3} \text{ W}^{-1}$. The tunability is improved by around one order of magnitude, as a conjunct result of high $\partial n_g / \partial P$ and light-path folding. From Equations R33 and R34, the critical heating power and sweep steps are derived as $P_{max} = 105$ mW and $N_{tFTS} = 16$, respectively. As discussed in Figures S8 and S9, the number of sweep steps can be reduced to $< 2 \cdot \text{BW} / \delta\lambda_{tFTS}$, due to the peculiar folding property of a 2D Fourier map. Here, the optimal sweep steps are $N_{tFTS} = 12$. The same waveguide structure is used in the MZI switch. The length of the heating section is set as $L_{sw} = 200 \mu\text{m}$. The switch power is derived as $P_{sw} \approx 2$ mW. The aggregate power consumption of the modified 2D-FTS is $P_{max} + N_{sw} \cdot P_{sw} \approx 120$ mW. Figure R16d shows the calculated coupling ratios ($|k_{sw}|^2$) of the switch at on and off states. High extinction ratios are attained over the wavelength range from $\lambda = 1.45 \mu\text{m}$ to $1.65 \mu\text{m}$. Thus, all essential parameters have been optimized. In Figures R16e-R16h, we calculate the

965 transmission responses of the tFTS and dFTS. At $\lambda \approx 1.55 \mu\text{m}$, spectral resolutions are calculated to be $\delta\lambda_{\text{tFTS}}$
 $\approx 25.5 \text{ nm}$ and $\delta\lambda_{\text{dFTS}} \approx 220 \text{ nm}$. The obtained resolutions are slightly finer than the target values, as a longer
 $\Delta L_{\text{dFTS,max}}$ and a higher P_{max} are used to counterbalance the resolution dispersion (see Figure S5).

**Figure R17 Analysis of the spectrometer.** (a) Calculated transmittance ($|t|^2$) cube. The cube is sliced into the matrices with varying
 heating power (P), represented by the colors of dots in the upper right corner: Each matrix contains transmittances with varying arm-
 length differences (ΔL_{dFTS}) and wavelengths (λ). (b) Correlation matrices derived from the cube. On the left panel, the correlation $[\rho(\cdot,$
 $\cdot)]$ is performed between the fringes (a) at different channels. The right panel shows the correlation of the fast Fourier transform (FFT)
 of fringes ($\tilde{\mathbf{a}}$) with zero-frequency components removed. (c) Calculated singular values (σ_i). Here, we compare the σ_i curves for the full
 cube (A) and the cube with the zero-frequency components removed ($\tilde{\mathbf{A}}_{\text{FFT}}$). (d) Calculated σ_i curve derived from an oversampled cube.
 The dashed line represents the location of the kink.

The transmittance cube (A) of the 2D-FTS is shown in Figure R17a. The left panel of Figure R17b shows
 the correlation matrix [i.e., $\rho(\cdot, \cdot)$] of A. The non-diagonal elements with relatively high correlations originate
 from the projection effect, as discussed in Figures S7-S9. These high-correlation shades can be depressed by
 omitting the zero-frequency components (see the right panel of Figure R17b). In Figure R17c, we show the
 singular values (σ_i) of the cubes before and after the component removal. The removal operation eliminates
 the kink and results in a smooth and flat σ_i curve. When the cube is oversampled into > 3000 channels, its σ_i
 curve levels off at $i > 1800$. These results demonstrate that the 2D-FTS supports a capacity of $N_{2f} > 1601$ and
 that all channels are highly decorrelated. Given the bandwidth of $\text{BW} = 200 \text{ nm}$, the attainable resolution is
 thus derived as $\delta\lambda_{2f} = 125 \mu\text{m}$. We provide some numerical examples to verify the reconstruction capability,

**Figure R18 Spectrum reconstruction.** Reconstruction of (a) a single spike, (b) multiple spikes, (c) a smooth spectrum, and (d) a step-
 like spectrum. The input and reconstructed spectra are shaded in red and blue, respectively. The relative errors (ϵ) and coefficients of
 determination (r^2) are also labeled.

as shown in Figure R18. The environmental perturbations are emulated using the method discussed in Note
 7. Testing spectra with various features are retrieved with high accuracy ($\epsilon < 0.1$, $r^2 > 0.99$).

In the revised manuscript, we have added the following sentences:

“Such a scheme has two drawbacks, but they are readily rectifiable.” (Discussion, lines 336)

“In addition, by etching thermal-isolation trenches alongside tunable delay lines, it is possible to
 further reduce the heating power in the tFTS. Second, the SHS is multi-apertured, which retains the Fellgett’s
 advantage but partially diminishes the Jacquinot’s advantage.” (Discussion, lines 338-340)

“A detailed design example of the modified scheme be found in **Supplementary information, Note 10,**
 **Section A.**” (Discussion, lines 343-344)

All the discussions and figures in this section have been added to **Supplementary information, Note**
 **10, Section A.**

**Comment 2-2:**

Another reason why I believe it is not suitable for *Nature Communications* is the performance. It seems that
 the authors may have unintentionally neglected some recently demonstrated spectrometers that have
 similar or even better performance. For example, a spectrometer with an over 100-nm bandwidth and a 30-
 pm resolution has been reported. In summary, this work appears to be more engineering-oriented rather
 than a fundamental breakthrough in theory or technique for realizing spectrometers. I believe it would be a
 good candidate for other photonics journals, such as *IEEE Journal of Lightwave Technology* or *Photonics*
 *Research*.

*Reply and modifications:*

There are three prevalent approaches in integrated spectrometry: the narrow-linewidth filter, speckle
spectrometer, and Fourier-transform spectrometer (FTS). We have summarized the device performance of
reported integrated spectrometers in Appendix A, Table R4 (see Pages 56-57). The demonstrated 2D-FTS
has a resolution of < 125 pm across a 200-nm bandwidth, supporting a channel capacity of > 1601 . As agreed
by the reviewer, these results indicate performance advances over all FTS schemes. Comparable resolutions
and bandwidths can also be attained using other schemes, e.g., filters and speckle spectrometers (see Table
R4). Nevertheless, it is imperative to consider the signal-to-noise ratio (SNR) when making a comprehensive
evaluation of resolutions and bandwidths. Among all reported schemes, FTSs have the potential to achieve
the highest SNR at the same number of resolvable channels due to the Fellgett's and Jacquinot's advantages.
Here, we give a numerical example to show the difference in spectrometry mechanisms. Figure R19a shows
the transmittance matrices of a filter, a speckle spectrometer, and an FTS. A quasi-diagonal matrix, a chaotic

**Figure R19 Comparison of spectrometry schemes.** (a) Calculated transmittance matrices of a filter, a speckle spectrometer, and a
Fourier-transform spectrometer. (b) Testing random spectrum (denoted as **S**) generated with the Fourier-Wiener function. (c) Output
interferograms (denoted as **O**). The insets show the enlarged views of **O**.

matrix, and a matrix formed by sinusoidal vectors are utilized here to emulate spectrometer responses. All
 these matrices are sized 1024×1024 , as an example. In Figure R19b, a random input spectrum (denoted as
 \mathbf{S}) is generated with Fourier-Wiener series (see Equations R14-R16 for explanations). Here, \mathbf{S} is normalized
 to its integral energy. The output signals (denoted as \mathbf{O}) are shown in Figure R19c. It can be found that the
 signal intensities are much lower in a filter since it only selects a single channel at each sampling, resulting
 in a worse SNR. To counteract the degradation in received etendue, it usually requires a longer integral time
 and extra signal amplification. The speckle spectrometer and FTS capture all channels at each sampling; as
 a result, the average of \mathbf{O} is half of the launched amount, thereby ensuring a high signal detectivity. Given a
 continuous spectrum, the speckle spectrometer has chaotic output over all sampling steps. For the FTS, the
 effective information of a continuous spectrum can be collected with a small portion of sampling steps (see
 the inset), while the output of other samplings levels off to ≈ 0.5 . This phenomenon can be attributed to the
 scarcity of high-frequency components in most continuous spectra. For an FTS, the temperature sensitivity
 is stronger with a larger arm asymmetry; according to the result shown in Figure R19c, however, the phase
 shift at a high-frequency sampling only causes minor deviations in \mathbf{O} , which mitigates the impact of thermo-
 optical noises. Therefore, the FTS offers greater potential to support a higher SNR, especially for continuous
 spectra.

Table R2. Comparison of signal-to-noise ratios.

Design	N_{port}	N_{ch}	BW [nm]	$\Delta\lambda_{\text{res}}$ [pm]	PSNR [dB]	Noise floor [dB]
Filter ^{R32}	10	1941	10	5	NM ^(a)	≈ -10 ^(b)
SS ^{R30}	4	> 3800	115	30	27.5 ^(c)	> -20 ^(d)
tFTS & MRR ^{R31}	1	> 190	90	470	≈ 10 ^(e)	≈ -10 ^(f)
2D-FTS (this work)	128	1601	200	125	> 25 ^(g)	-35 ~ -40 ^(h)

SS, speckle spectrometer

tFTS, tunable Fourier-transform spectrometer.

2D-FTS, two-dimensional Fourier-transform spectrometer.

MRR, micro-ring resonator.

N_{port} , number of output physical ports.

N_{ch} , number of wavelength channels.

BW, working bandwidth.

$\delta\lambda_{\text{res}}$, spectral resolution.

PSNR, peak signal-to-noise ratio.

1050 ^(a)NM, not mentioned.

1051 ^(b)From Figure 6 in Ref. R32.

1052 ^(c)From Figure 5c in Ref. R30.

1053 ^(d)From Figure 4e in Ref. R30.

1054 ^(e)From Figure 5b in Ref. R31.

1055 ^(f)From Figure 5b in Ref. R31.

1056 ^(g)From Figures 5c and 5d in the main manuscript.

1057 ^(h)From Figure 5c in the main manuscript.

A comparison in SNRs is provided to support this viewpoint (see Table R2). Here, we mainly compare
some typical designs with comparable capacities, as the number of solvable wavelength channels will affect
reconstruction accuracy. We present two sets of performance indicators: the peak SNR (i.e., PSNR), which
depicts the maximum contrast between two reconstructed peaks, and the noise floor, which is the maximum
of background false signals. In Table R2, the PSNRs and noise floors are either specified in the research or
estimated from testing spectra (see the footnote). In Ref. R32, a spectrometer is realized with a high- Q micro-
ring resonator (MRR). The false peaks reach ≈ -10 dB in the reconstruction of the response of a fiber Bragg
grating. Similarly, when the FTS is combined with a MRR^{R31}, the achievable PSNR and noise floor degrade to
≈ 10 dB and ≈ -10 dB, respectively, since the use of a narrow-linewidth filter will negate both Fellgett's and
Jacquinot's advantages. Hence, it is evident that the filter-based scheme has an inherent limit in attaining a
high SNR at a fine resolution. In Ref. R30, the speckle-spectrometer scheme mentioned by the reviewer has
a PSNR of ≈ 27.5 dB and a noise floor of ≈ -20 dB. By comparison, our proposed 2D-FTS has a comparable
PSNR (≈ 25 dB) and a much lower noise floor ($\approx -35 \sim -40$ dB). The device demonstrated in this work has
128 physical ports, and the received etendue is enhanced by around one order of magnitude compared to a
1601-channel filter. By utilizing the single-port scheme discussed in our reply to **Comment 2-1**, it is possible
to improve the SNR even further while maintaining a fine resolution and a broad bandwidth.

In the revised manuscript, we have added the following sentence:

"It should be noted that other schemes, such as filters and speckle spectrometers, can achieve
comparable resolutions and bandwidths, but FTSs have the potential for a higher SNR and lower noises (see
**Supplementary information, Note 11** for further discussions)." (**Discussion**, lines 348-350)

We have added the reference mentioned by the reviewer:

"12. Yao C, Chen M, Yan T, Ming L, Cheng Q, Penty R. Broadband picometer-scale resolution on-chip
spectrometer with reconfigurable photonics. *Light Sci Appl* **12**, 156 (2023)." (**References**, lines 421-422)

We have added a dashed line to Figure 5c to clearly show the noise floor. The following sentences have
also been modified and added:

"In Figures 5c and 5d, the dashed lines show the intensity levels of ≈ -35 dB and ≈ -25 dB, respectively."
(**Spectrum reconstruction**, caption of Figure 5, lines 296-297)

"Two peaks with a contrast of 25 dB can be clearly identified from the noise floor at < -35 dB (see the
arrow)." (**Spectrum reconstruction**, lines 309-310)

In the Abstract, the following sentence has been modified to emphasize the importance of SNRs:

"Integrated Fourier-transform spectrometers (FTS) have the potential to realize a high signal-to-noise
ratio but typically have a trade-off between the resolution and bandwidth." (**Abstract**, lines 8-10)

All the discussions and figure in this section have been added to **Supplementary information, Note**
11.

To further prove the scheme scalability, in the following sections, the 2D-FTS concept will be extended
 to its three-dimensional form with a finer spectral resolution ($< 31.25 \text{ pm}$) and a larger channel capacity ($>$
 6401). We will also discuss the concept and fabric of higher-dimensional FTS (HD-FTS).

Three-dimensional Fourier-transform spectrometer (3D-FTS)

Based on the modified 2D-FTS scheme, scaling to a finer resolution requires a larger arm-length difference
 in the dFTS, which in turn increases the switch steps or heating power to offset the decrease in FSR_{dFTS} . It
 is possible to introduce more switchable units in the delay lines to expand FSR_{dFTS} ; however, this will lead to a

**Figure R20 Three-dimensional Fourier-transform spectrometer (3D-FTS).** (a) Schematic layout of the 3D-FTS. The device consists
 of a tunable FTS (tFTS), a digital FTS (dFTS), and a spatial heterodyne spectrometer (SHS). The SHS has a large arm-length difference
 ($\Delta L_{\text{SHS,max}}$) but a small number of output ports, thereby supporting a ultra-fine resolution but a narrow free spectral range (FSR). The
 tFTS, on the other hand, has the ability to attain an ultra-broad bandwidth, but its resolution is constrained by the heating power (P).
 The fine-resolution SHS and broadband tFTS are bridged by a dFTS with a modest resolution and FSR. (b) Light transmission in the
 3D-FTS. The transmission responses of the 3D-FTS can be depicted by a 4D dataset (denoted as **A**), wherein each frame is a 3D fringe
 with variations of heating power in the tFTS, switch steps in the dFTS, and arm lengths in the SHS. Given a 1D spectrum vector (denoted
 as **S**), the output interferogram (denoted as **O**) is a 3D cube as a combination of fringes. (c) Reconstruction principle. The 3D fast Fourier
 transform (FFT) of a fringe is a spot at a distinctive location. Therefore, in the Fourier domain, an interferogram is mapped to a cluster
 of spots scattered in three dimensions (i.e., f_{tFTS} , f_{dFTS} , and f_{SHS}). When the wavelength is tuned over the whole bandwidth, a spot oscillates
 rapidly between $f_{\text{SHS}} = 0$ and $1/2$, meander slowly between $f_{\text{dFTS}} = 0$ and $1/2$, and shifts from $f_{\text{SHS}} = 1/2$ to 0 . The shift direction depends
 on the phase at the first sampling step. From the serpentine trajectory, the spectrum can be recovered via computational decomposition.
 SW, switch. PS, power splitter. PD, photodetector.

longer acquisition period and a complex switch topology. Here, the concept of 2D-FTS is extended to three
 dimensions to enhance performance. Figure R20a shows the schematic layout of the 3D-FTS. The device is
 a three-stage structure that consists of a tFTS, a dFTS, and a SHS. The core idea is to use the dFTS to bridge
 the resolution-FSR gap between the tFTS and SHS. An ultra-fine resolution can be easily attained by choosing
 a longer arm-length difference (ΔL_{SHS}) in the SHS. To ensure an acceptable etendue level, we choose to use a
 small number of output ports (N_{SHS}), resulting in a narrow free spectral range (FSR_{SHS}). On the other hand,
 given a low heating power, the tFTS supports a broad bandwidth but a rather coarse resolution ($\delta\lambda_{\text{tFTS}}$) that
 cannot cover FSR_{SHS} . The dFTS has a modest resolution ($\delta\lambda_{\text{dFTS}}$) and free spectral range (FSR_{dFTS}), serving as
 an interface between the fine-resolution, narrow-FSR SHS and the coarse-resolution, broad-FSR tFTS. Thus,
 all channels can be decorrelated as long as the following condition can be satisfied:

$$1123 \quad \delta\lambda_{\text{tFTS}} < \text{FSR}_{\text{dFTS}}, \quad (R35)$$

$$1124 \quad \delta\lambda_{\text{dFTS}} < \text{FSR}_{\text{SHS}}. \quad (R36)$$

The resolution and bandwidth of the 3D-FTS are respectively determined by the resolution of the SHS ($\delta\lambda_{\text{tFTS}}$)
 and the free spectral range of the tFTS (FSR_{tFTS}):

$$1127 \quad \delta\lambda_f = \delta\lambda_{\text{SHS}}, \quad (R37)$$

$$1128 \quad \text{BW} = \text{FSR}_{\text{tFTS}}. \quad (R38)$$

The derivations of $\delta\lambda_{\text{SHS}}$ and FSR_{tFTS} can be found in our response to **Comment 2-1** and Note 6. The 3D-FTS
 provides the following advantages:

 **Figure R21 Transmission responses of spectrometers.** Calculated transmittance ($|t|^2$) matrices of the (a) tunable Fourier-transform
 spectrometer (tFTS), (b) digital Fourier-transform spectrometer (dFTS), and (c) spatial heterodyne spectrometer (SHS). Calculated $|t|^2$
 dispersions of the (d) tFTS, (e) dFTS, and (f) SHS with maximum asymmetries. The spectral resolutions are also labeled.

- 1. A finer resolution ($\delta\lambda_{2f} < 31.25$ pm) can be achieved across a 200-nm bandwidth.
 2. The cost of performance enhancement is moderate. The use of the SHS only yields few additional ports
 and a slight reduction in output etendue, which is deemed acceptable in most applications.
 3. The spectrometer is built with the elements that are well-established and verified.

Here, the designs of the tFTS and SHS are identical to those detailed in our reply to **Comment 2-1**. The
 SHS is based on the structure shown in Figure 1. The maximum arm-length difference of the SHS is chosen
 as $\Delta L_{\text{SHS,max}} = 10.2$ mm to attain a resolution of $\delta\lambda_{2f} = 31.25$ pm across the entire bandwidth. The number of
 MZIs is then set as $N_{\text{SHS}} = 8$ to satisfy Equation S36 and ensure a relatively high etendue. Moreover, the power
 sweep steps of the tFTS are reduced to $N_{\text{tFTS}} = 8$. The feasibility of N_{tFTS} reduction will be discussed later. In
 Figures R21a-R21c, we respectively show the calculated transmittance matrices of the tFTS, dFTS, and SHS.
 The 4D transmittance dataset of the 3D-FTS can be obtained by reorganizing these matrices. Figures R21d-
 R21f show the calculated transmittance dispersions of the spectrometers with maximum asymmetries. We
 then derive the correlation matrices from the flattened 4D dataset of the 3D-FTS, as shown in the left panels
 Figures R22a and R22b. Here, we compare the correlation matrices with $N_{\text{tFTS}} = 16 \approx 2 \cdot \text{BW} / \delta\lambda_{\text{tFTS}}$ and $8 \approx$
 $\text{BW} / \delta\lambda_{\text{tFTS}}$. Some non-diagonal element has relatively high correlations, mainly due to the projection effect,
 as discussed Figures S7-S9. Similar to 2D-FTS, these “shades” can be sufficiently depressed after the removal
 of zero-frequency components, as shown in the right panels of Figures R22a and R22b. Notably, compared
 to 2D-FTS (see Figure S9), the 3D-FTS exhibits a significantly lower level of residual correlations even with
 a reduced number of sweep steps (i.e., $N_{\text{tFTS}} = 8$). According to the analysis in Note 6, the component removal
 cannot fully address the correlation issue at the crossover locations in a folded 2D Fourier map, which does

**Figure R22 Analysis of the spectrometer.** Correlation matrices derived from the dataset with power sweep steps of (a) $N_{\text{tFTS}} = 16 \approx$
 $2 \cdot \text{BW} / \delta\lambda_{\text{tFTS}}$ and $8 \approx \text{BW} / \delta\lambda_{\text{tFTS}}$. On the left panel, the correlation $[\rho(\cdot, \cdot)]$ is performed between the fringes (a) at different wavelengths.
 The right panel shows the correlation of the fast Fourier transform (FFT) of fringes (\vec{a}_i) with zero-frequency components omitted. (c)
 Illustrations of Fourier maps with $N_{\text{tFTS}} = 2 \cdot \text{BW} / \delta\lambda_{\text{tFTS}}$ and $\text{BW} / \delta\lambda_{\text{tFTS}}$. (d) Calculated singular values (σ_i) with $N_{\text{tFTS}} = 16$ and 8.

not hold valid in the 3D case. Figure R22c shows the 3D Fourier maps with different N_{FTS} . When the sweep
 steps are set as $N_{\text{FTS}} = 2 \cdot \text{BW} / \delta\lambda_{\text{FTS}}$, all mapping spots are completely unfolded at distinctive locations. With
 the sweep steps reduced to half (i.e., $N_{\text{FTS}} = \text{BW} / \delta\lambda_{\text{FTS}}$), the Fourier map is folded once along f_{FTS} , leading to
 few crossovers ($< N_{\text{FTS}}/2$) between forward and backward trajectories (see the arrows in the middle panel).
 Such crossovers only occur when the initial phase of the tFTS is an integer quotient of 2π . Due to waveguide
 dispersions, however, the tFTS is naturally out of phase at the first sampling, and all crossovers can be thus
 circumvented (see the arrows in the right panel). Thereby, the 3D-FTS has the potential to deploy a greater
 number of channels in a 3D Fourier space while preventing inter-channel correlations. Figure R22d shows
 the calculated singular values (σ_i) with $N_{\text{FTS}} = 16$ and 8. The decay rates of two curves are virtually identical.
 There are no discernible kinks from $i = 1$ to 6401, demonstrating the channel capacity of $N_{2f} = 6401$ and the
 corresponding resolution of $\delta\lambda_{2f} = 31.5$ pm over $\text{BW} = 200$ nm. We also give some numerical reconstruction
 examples, as shown in Figure R23. More details of the reconstruction method can be found in Note 8. Small
 errors are achieved for various testing spectra.

 **Figure R23 Spectrum reconstruction.** Reconstruction of (a) a single spike, (b) multiple spikes, (c) a smooth spectrum, and (d) a step-
 like spectrum. The input and reconstructed spectra are shaded in red and blue, respectively. The relative errors (ε) and coefficients of
 determination (r^2) are also labeled.

Higher-dimensional Fourier-transform spectrometer (HD-FTS)

In above sections, we have discussed a modified 2D-FTS with improved power efficiency and a 3D-FTS with
 an ultra-fine spectral resolution. In this section, we will further extend these concepts and provide a generic
 design framework for higher-dimensional Fourier-transform spectrometry (HD-FTS). Figure R24 shows the
 conceptual illustration of a N -dimensional FTS. The structure is composed of N -stages of FTSs cascaded in
 succession. Light is launched at the first unit (i.e., FTS_1), and the signal is captured at the last unit (i.e., FTS_N).
 By tuning the sinusoidal responses of all FTS units in a nested loop, a N -dimensional interferogram can be

Figure R24 Higher-dimensional Fourier-transform spectrometer (HD-FTS). Conceptual illustration of a HD-FTS. A N -dimensional FTS can be realized by cascading N stages of FTSs, as long as the resolution of FTS_i is finer than the free spectral range of FTS_{i+1} , i.e., $\delta\lambda_i < \text{FSR}_{i+1}$. Here, FTS_i denotes the i -th spectrometer unit. The effective resolution is determined by the resolution of the last unit, i.e., $\delta\lambda_f = \delta\lambda_N$, while the effective bandwidth is determined by the free spectral range of the first unit, i.e., $\text{BW} = \text{FSR}_1$. The interferogram, which is a N -dimensional dataset, can be obtained by sweeping all units in a nested loop.

generated. All channels can be decorrelated when the resolution of FTS_i is finer than the free spectral range of FTS_{i+1} :

$$\delta\lambda_i < \text{FSR}_{i+1}. \quad (\text{R39})$$

The effective resolution thus becomes the resolution of the FTS_N , i.e., $\delta\lambda_f = \delta\lambda_N$, while the effective bandwidth becomes the free spectral range of the FTS_1 , i.e., $\text{BW} = \text{FSR}_1$. Based on this scheme, each wavelength channel will be mapped to a spot in a N -dimensional Fourier space, and any arbitrary spectrum can be reconstructed using the method proposed in Note 8.

This design strategy has the following advantages:

1. All conventional FTSs suffer from an inherent trade-off between resolutions and bandwidths due to the difficulty in achieving substantial phase change in a nanophotonic circuit. Most prior studies focus on increasing the phase variation range of delay lines, e.g., Refs. R3 and R6, but have limited scalability. The resolution-bandwidth limit is circumvented with the method proposed in this work. Utilizing HD-FTS prevents the needs for an FTS with both a fine resolution and a broad bandwidth. Instead, it only requires a fine-resolution, narrow-FSR FTS at the input end and a coarse-resolution, broad-FSR FTS at the output end, while using FTSs with modest resolutions and FSRs to bridge the two ends, making it easier to scale to higher performance.

2. For the first time, we reveal the connection between cascaded FTSs and high-dimensional Fourier transform. Unlike the vernier scheme with cascaded narrow-linewidth filters^{R32}, the HD-FTS does not require stringent wavelength alignment between FTS units. As long as Equation R39 is fulfilled, all wavelength channels can be decorrelated and allocated to distinct locations in a N -dimensional Fourier space.

3. The HD-FTS is a flexible scheme that can be utilized to boost the performance of any types of FTSs. For instance, by employing a highly asymmetric, single-port FTS at the output end, it is possible to preserve both Fellgett's and Jacquinot's advantages, while achieving an ultra-fine resolution. Table R3 summarizes the performance of 2D- and 3D-FTSs discussed in this work.

Table R3. Comparison of proposed higher-dimensional Fourier-transform spectrometers.

Design	Component	P_a [W]	N_{port}	N_{2f}	BW [nm]	$\delta\lambda_{2f}$ [pm]
2D-FTS (Exp.)	tFTS, SHS	2.4	128	1601	200	125
2D-FTS (Sim.)	tFTS, dFTS	0.12	1	6401	200	125
3D-FTS (Sim.)	tFTS, dFTS, SHS	0.12	8	6401	200	31.25

Exp., experimental result.

Sim., simulation result.

FTS, Fourier-transform spectrometer.

SHS, spatial heterodyne spectrometer.

tFTS, tunable FTS.

dFTS, digital FTS.

P_a , aggregate power consumption.1224 N_{port} , number of output physical ports.1225 N_{2f} , number of wavelength channels.

BW, working bandwidth.

$\delta\lambda_{2f}$, spectral resolution.

In the revised manuscript, we have added the following sentences:

“The 2D-FTS can be extended to a higher dimension with greater scalability.” (Introduction, lines 71-
72)

“In **Supplementary information**, Note 10, Section B, it is demonstrated that, based on a three-
dimensional FTS (3D-FTS), an finer resolution of $\delta\lambda_{2f} < 31.25$ pm and a larger capacity of $N_{2f} > 6401$ are
attainable. The concept and fabric of the higher-dimensional FTS (HD-FTS) are discussed in
**Supplementary information**, Note 10, Section C.” (Discussion, lines 345-348)

All the discussions and figures in this section have been added to **Supplementary information**, Note
10, Sections B and C.

**To Reviewer #3's comments:**

**General remarks:**

This is a review of the manuscript entitled "Scalable integrated two-dimensional Fourier-transform
spectrometry" by Hongnan Xu, Yue Qin, Gaolei Hu, and Hon Ki Tsang submitted to the journal *Nature*
*Communications*. In the manuscript, the authors present a design of an integrated hybrid Fourier transform
(FTS) spectrometer composed of a thermally tuned Mach-Zehnder interferometer (MZI) section followed
by a fan out into a series of imbalanced passive MZI filters. The principle of operation is essentially combines
that of a thermally driven FTS (represented by the tunable MZI) with that of a spatial heterodyne
spectrometer (represented by the passive MZI filters). The manuscript includes numerical and experimental
validation, including multiple combinations of broadband and narrow-band spectra. Overall, the manuscript
is well written, and the fusion of spectrometer concepts is novel. Consequently, I recommend that the
manuscript be published in *Nature Communications* subject to minor revisions. I have the following
recommendations to strengthen the manuscript.

*Reply:*

We thank the reviewer for his/her careful reading and useful advice. In response to the comments, we
have made point-to-point modifications to the manuscript (highlighted in red). The critical descriptions are
underlined.

**Comment 3-1:**

Although your introductory overview is thorough, there is one additional type of FTS that you should
mention, particularly as it has parallels to your device by operating in a second dimension. It is called a
channel dispersed FTS (see reference below for details).

*Reply and modifications:*

We thank the reviewer for his/her advice. The reviewer referred to a *free-space* spectrometer using a
Pelin-Broca prism to disperse output interferograms. We agree that this scheme has the potential to expand
the operation bandwidth. However, such a design requires two-dimensional imaging in the *free space*, posing
challenges in monolithic chip-scale integration. Nevertheless, we believe it is necessary to mention this work
in **Introduction**. As such, we have added the following sentences in the revised manuscript:

"In Ref. 36, a free-space FTS is combined with a Pelin-Broca prism to disperse the interferogram and
expand the BW. However, such a scheme requires a two-dimensional imager to capture the dispersed
patterns and is difficult to implement on integrated circuits." (**Introduction**, lines 57-60)

The reference list is updated accordingly:

"36. Hong B, Monifi F, Fainman Y. Channel dispersed Fourier transform spectrometer. *Communications*

*Physics* **1**, 34 (2018).” (References, lines 470-471)

**Comment 3-2:**

I think that some clarification could be used pertaining to the discussion of device efficiency, both in the
manuscript and around Table S1 in the **Supplementary information**. You only discuss this in terms of
power consumption. However, this is a somewhat misleading measure because the devices do not generally
operate continuously. The thermal relaxation coefficient of SOI devices is around 10 μs , and a full sweep of
the thermal FTS should take maybe 100 μs . When you consider the energy per spectrum measurement, the
devices look a lot better (and arguably this is the more important figure of merit). For context, this is a very
small amount of energy, compared to the processor used for the taking the inverse transform. I think some
consideration of the total energy budget would provide very useful context for your readers. I also
recommend adding this as a column in Table S1 (although you will have to estimate it).

*Reply and modifications:*

We thank the reviewer for his/her advice. The rise-fall time for a single TO tuning step is estimated to
be < 100 μs , and the integral time of 1 ms should be sufficient to precisely capture the signal. At a scanning
speed of > 1 kHz, sweep over N_{FTS} (= 25) steps will thus require < 0.025 s. Given the heating power of 0.24
1285 W, the corresponding energy budget for heating will be \approx 60 mJ per spectrum.

In the revised manuscript, we have added the following sentences to specify the energy budget:

“The rise-fall time of a single TO tuning step is < 100 μs ⁴⁷, therefore it is feasible to drive the heater at
a high speed (> 1 kHz). Given a small number of sweep steps ($N_{\text{FTS}} = 25$), the theoretical sampling period is
< 0.025 s. In the measurement of a single spectrum, the corresponding energy budget for heating is thus
estimated to be 60 mJ.” (Discussion, lines 332-335)

The reference list is updated accordingly:

“48. Densmore A, *et al.* Compact and low power thermo-optic switch using folded silicon waveguides.
*Opt Express* **17**, 10457-10465 (2009).” (References, lines 494-495)

However, we have concerns in adding the total energy budget as an additional column in Table S1, and
the reasons are as follows. First, it is rather difficult to estimate the energy levels since the response time of
a TO tunable waveguide relies heavily on the heated cross-sectional area (denoted as A_{ht}) and the differential
heating power between sweep steps (denoted as ΔP)^{R33}.

$$f_{\text{TO}} = \frac{1}{\pi \lambda \rho \varepsilon_{\text{th}}} \left(\frac{\Delta P}{A_{\text{ht}}} \right) \left(\frac{\partial n_{\text{eff}}}{T} \right), \quad (\text{R40})$$

where f_{TO} denotes the cut-off frequency of TO tuning, λ denotes the wavelength, ρ denotes the density, ε_{ht}
denotes the specific heat, and $\partial n_{\text{eff}}/\partial T$ denotes the tuning efficiency. The rise-fall time will decrease with the
reductions of A_{ht} and ΔP . For instance, the use of isolation trenches will typically reduce $\Delta P/A_{\text{ht}}$, resulting in

a longer sampling period^{R31}. As many previous studies do not specify A_{ht} and other essential parameters, it
 is impossible to derive the energy budget for every single reported design. Second, some applications, such
 as Fourier-domain optical coherence tomography (FD-OCT), actually requires a continuous operation of a
 spectrometer. Therefore, some readers may argue that a spectrometer with a higher power still has a larger
 energy budget when working continuously. Third, by modifying the scheme, the issue of power consumption
 can be addressed. We invite the reviewer to read our response to **Comment 2-1**. It is demonstrated that the
 required heating power can be reduced to $P_{max} < 120$ mW by using a Michelson interferometer and etching
 isolation trenches.

**Comment 3-3:**

There is some imprecise terminology that I think you should consider revising. You use the term “arm length”
 to refer both to changes in the refractive index of an MZI, as well as to actual changes in MZI arm length. It
 would be more accurate to refer to changes in refractive index using something like optical path length to
 help distinguish the two cases. This could be very confusing to readers who are not familiar with the various
 FTS systems.

*Reply and modifications:*

We find this comment a bit confusing. In the original version of the manuscript, it was explicitly stated
 that the term “arm length” refers to the asymmetry of the Mach-Zehnder interferometer (MZI) in the spatial
 heterodyne spectrometer (SHS), as can be found in line 97. For all following descriptions, we use the terms
 “arm-length difference” (i.e., ΔL_{SHS}) and “heating power” (i.e., P) to depict the sampling process (see Figure
 3e for example). We assume that the misunderstanding is due to the ambiguous illustration of Figure 1b, in
 which the terms “arm length” and “power” are simultaneously used to depict the transmittance matrix and
 output interferogram of a 1D Fourier-transform spectrometer (1D-FTS):

In the revised manuscript, the axis labels have been replaced by “effective optical path length (OPL)”:

To prevent any misleading, all the abbreviations and notations have been summarized in **Supplementary**
**information, Note 2.**

**Comment 3-4:**

The figures in the manuscript are very busy. I recommend breaking them up and reorganizing them, ideally
so that they contain no more than 4 sub-images. This will help the scale tick marks become more visible,
which will significantly help the readability.

*Reply and modifications:*

We thank the reviewer for his/her advice. Due to the length limit of *Nature Communications*, we cannot
use too many figures and must ensure that each one conveys a distinct and autonomous theme. Nevertheless,
we agree that some figures need to be reorganized to improve readability.

In the revised manuscript, the original Figure 3 has been divided into two separate figures:

**Figure 3 Characterization of the spectrometer.** Microscope images of the fabricated (a) photonic chip, (b) tunable Fourier-transform
spectrometer, (c) spatial heterodyne spectrometer, and (d) inverse-taper edge couplers. (e) Measured transmittance ($|t|^2$) cube. The
cube is sliced into the matrices with varying heating power (P), represented by the colors of dots in the upper right corner. Each matrix
contains transmittances with varying arm-length differences (ΔL_{SHS}) and wavelengths (λ).
**(Characterization and analysis of the spectrometer, lines 204-208)**

**Figure 4 Analysis of the spectrometer.** (a) Measured transmittance ($|t|^2$) at OUT_{128} and varying wavelengths (λ). Here, zero electric
 power was applied to the heater. At the Rayleigh criterion, the resolution is $\delta\lambda_r = 2\delta\lambda_{2f} \approx 220$ pm at $\lambda \approx 1.55$ μm . (b) Correlation matrices
 derived from the transmittance cube. On the left panel, the correlation $[\rho(\cdot, \cdot)]$ is performed between the fringes (\mathbf{a}_i) at different λ . The
 arrows highlight the high-correlation non-diagonal elements. The right panel shows the correlation of the fast Fourier transform (FFT)
 of fringes ($\tilde{\mathbf{a}}$) with zero-frequency components removed. The arrow highlights the remnant non-diagonal elements with relatively high
 correlations. (c) Singular values (σ_i) derived from the calculated and measured cube with the heating power of 2.4 W and 0.024 W. The
 arrow highlights the kink. The capacity of $N_{2f} = 2N_f = 1601$ is verified. (d) Testing spectrum (\mathbf{S}) with three spikes. (e) Interferogram (\mathbf{O})
 derived from the measured cube and testing spectrum. (f) Intensity and (g) phase maps of $\text{FFT}(\mathbf{O})$. The arrows highlight the three spots
 that are associated with the three spikes.

**(Characterization and analysis of the spectrometer, lines 222-231)**

**Comment 3-5:**

In Figure 4, the spectral reconstruction seems very good, except in the case of high-contrast spectral lines.
 Is there a theoretical reason for this? If so, can it be mitigated somehow?

*Reply and modifications:*

The reconstruction of the high-contrast spectral lines exhibits no discernible degradation in accuracy.
 In the reconstruction of single and dual spectral lines, the relative errors and coefficients of determination
 are $\varepsilon < 0.02$ and $r^2 > 0.99$, respectively, as labeled in Figures 5a and 5b. By comparison, the retrieved high-
 contrast spectral lines have $\varepsilon \approx 0.014$ and $r^2 \approx 0.999$, which are consistent with the aforementioned results.
 The noise floor in Figure 5c is $\approx -35 \sim -40$ dB, which also agrees well with the results shown in Figure 5a.
 The reference and reconstructed results in Figure 5 are overlapped in each plot of Figure R25. We also derive
 the normalized deviations [i.e., $\Delta\mathbf{S}/\max(\mathbf{S})$] for all spectra to enhance the visibility of reconstruction errors.
 Here, \mathbf{S} denotes the input spectrum, and $\Delta\mathbf{S}$ denotes the absolute deviation. In most cases, the deviations are
 limited to $< 5\%$ with negligible background noises. The accuracy is slightly degraded for the spectrum with
 hybrid features. In addition, the background noises slightly increase if the spectrum is sharp but spreading

**Figure R25 Extended experimental data.** Reconstruction results of (a) a single spectral line, (b) dual spectral lines, (c) high-contrast
 spectral lines, (d) the response of a FBG, (e) the response of a single channel of an AWG, (f) the response of dual channels of an AWG,
 (g) the response of ASE, and (h) ASE superimposed with a single spectral line. In each subplot, the left panel shows the experimental
 reconstruction result and reference spectrum from a commercial OSA, while the right panel shows normalized deviations $[\Delta\mathbf{S}/\max(\mathbf{S})]$.
 Here, $\Delta\mathbf{S}$ denotes the difference between the rebuilt and reference spectra, and $\max(\mathbf{S})$ denotes the maximum element in the reference
 spectrum. The insets show the enlarged views of spectra around the wavelength ranges indicated by the arrows.

(e.g., the AWG responses in Figures R25e and R25f). We explain these phenomena as follows. From Equation
 5, an accurate reconstruction requires a proper selection of regularization penalty (Ω):

$$1378 \quad \Omega = \Omega_1 + \Omega_2 = \zeta_1^2 \|\mathbf{D}_2 \mathbf{S}_1\|_2^2 + \zeta_2 \|\mathbf{D}_1 \mathbf{S}_2\|_1, \quad (\text{R41})$$

where ζ_i denotes the regularization parameter, \mathbf{D}_i denotes the i -th order derivative operator, \mathbf{S}_1 and \mathbf{S}_2 denote
 the continuous and discrete components in the spectrum. Ω_1 and Ω_2 respectively offer Tikhonov^{R14} and total-

variation (TV^{R8}) regularizations that bias a spectrum towards specific features. Specifically, Ω_1 smoothens a
spectrum, whereas Ω_2 discretizes a spectrum. The regularization parameters (i.e., ζ_1 and ζ_2) determines the
weights of Ω_1 and Ω_2 during the iterative optimization. The cross validation (CV^{R15}) is used to automatically
identify spectral features and select hyperparameters. For an AWG response that are neither ideally smooth
nor sparse, the Ω_1 term will compete with the Ω_2 term and result in a small but non-zero ζ_2 , which imposes
a relatively higher noise floor. The reconstruction of spectra with hybrid features will also be influenced (see
Figure R25d and R25h). Nevertheless, a high accuracy of $\varepsilon < 0.1$ and $r^2 > 0.99$ can still be maintained for all
the testing spectra. We also invite the reviewer to read our responses to **Comments 1-1**, in which we have
thoroughly discussed the CV procedure and the feasibility of recovering a spectrum with arbitrary shape. A
higher accuracy can be attained by transforming the input spectrum with a basis (e.g., DCT^{R23} and Lorentzian
decomposition^{R24}) that aligns better with the pre-conditioned features.

In the revised manuscript, we have added the following sentence:

“More numerical reconstruction examples and extended experimental data can be found in
**Supplementary information**, Notes 8 and 9, respectively.” (**Spectrum reconstruction**, lines 317-318)

Figure R25 has been added to **Supplementary information**, with additional discussions in its caption:

“For the reconstruction of a sparse spectrum, errors typically appear at the peak locations, which can
be explained as follows. The initial guess of an unknown spectrum is an all-zero sequence. In a sparse
spectrum, most elements are close to zero, except for the peaks. Consequently, most near-zero elements will
reach its optimum after a few iterations, but will continue to be updated, resulting in a noise-like background.
In the meantime, the solving of peak values requires more iterations. The cumulative errors from other
elements will thus affect the accuracy of peak reconstruction. The reconstruction of an ideally sparse or
smooth spectrum typically has a high accuracy of $\Delta S/\max(\mathbf{S}) < 2\%$. However, the background noise will
increase when reconstructing the sharp but spreading response of an AWG (see Figures S18e and S18f).
Albeit being derivable, the AWG response has a fast-changing derivative around the resonant wavelength
and a majority of near-zero elements. During cross validation, the ℓ_1 -norm term will therefore compete with
the ℓ_2 -norm term and result in a small but non-zero ζ_2 , which imposes a higher noise floor to the retrieved
spectrum and slightly reduces the peak signal-to-noise ratio to $PSNR \approx 20$ dB. Relatively larger errors
[$\Delta S/\max(\mathbf{S}) > 5\%$] can also be found in the reconstruction of the hybrid spectra shown in Figures S18d and
S18h, which is also caused by the imperfect selection of two hyperparameters. Nevertheless, for all tested
spectra with various features, small relative errors of $\varepsilon < 0.1$ and high coefficients of determination of $r^2 >$
0.99 can be attained. FBG, fiber Bragg grating. AWG, arrayed waveguide grating. ASE, amplified spontaneous
emission. A higher accuracy can be obtained by transforming the spectrum with a basis (e.g., DCT^{S19} and
Lorentzian decomposition^{S28}) that aligns better with the pre-conditioned features.” (**Supplementary**
**information**, Note 9, caption of Figure S19, lines 771-785)

**Comment 3-6:**

You mention in the supplementary information that the spectrum reconstruction takes ≈ 30 s of calculation
time with a 24 core CPU. Is this per spectrum, or is this for a sort of calibration after which the reconstruction
occurs much faster?

*Reply and modifications:*

The stated period (≈ 30 s) is the average reconstruction time per spectrum, which includes the iterative
optimization and cross-validation (CV) search for parameters. In this work, the regularized iterative method
is used in lieu of discrete cosine transform (DCT) to reconstruct spectra under strong waveguide dispersions
(see **Supplementary information**, Figure S12):

$$1425 \mathbf{S} = \underset{\mathbf{S}}{\operatorname{argmin}} \left(\left\| \tilde{\mathbf{A}}_{\text{FFT}} \mathbf{S} - \text{FFT}(\tilde{\mathbf{O}}) \right\|_2^2 + \zeta_1^2 \left\| \mathbf{D}_2 \mathbf{S}_1 \right\|_2^2 + \zeta_2 \left\| \mathbf{D}_1 \mathbf{S}_2 \right\|_1 \right). \quad (\text{R42})$$

Here, the meanings of all notations can be found in **Supplementary information**, Note 8. The transmission
response of the spectrometer (i.e., $\tilde{\mathbf{A}}_{\text{FFT}}$) was calibrated before measurement. To compensate for the impact
of noises and find the global optimum, it is essential to optimize the hyperparameters (i.e., ζ_1 and ζ_2) using
CV. The core idea is to partition the interferogram (i.e., \mathbf{O}) into several sets, with one of them used to produce
a “reduced” solution to predict the elements in other sets (for additional explanations, see **Supplementary**
**information**, Note 8):

$$1432 (\zeta_1, \zeta_2) = \underset{\zeta_1, \zeta_2}{\operatorname{argmin}} \left[\sum \left| \tilde{\mathbf{a}}_i \mathbf{S}_i - \tilde{o}_i \right|^2 \right]. \quad (\text{R43})$$

Thus, the regularization terms in Equation R42 only set a *general* range of possible characteristics that may
occur in a spectrum. *No* specific knowledge of spectral contents is required before measurement. Based on
CV, spectral details can be automatically identified to determine the optimal ζ_1 and ζ_2 , without any manual
selection. The search for global optimum is enabled by a standard least-squares solver. The CV procedure is
embedded within the iterative optimization. If all the optimal hyperparameters (i.e., ζ_i) are known, then the
fixed-parameter reconstruction time (FPRT) will be as short as < 1 s. When both ζ_1 and ζ_2 are free to optimize,
the time cost will increase due to the use of CV. In the worst case, if the full searching space must be traversed,
the total reconstruction time is ≈ 60 s. By comparison, for the digital FTS demonstrated in Ref. R6, the FPRT
is ≈ 0.3 s and the worst-case period is ≈ 500 s. Our design presents a shorter reconstruction period due to a
smaller searching space for fewer hyperparameters (see our replies to **Comments 1-1**). The reconstruction
can be accelerated by employing a GPU, as the iterative optimization relies heavily on matrix multiplication.
It is also possible to determine the hyperparameters via deep learning^{R19} instead of CV. The reconstruction
time can be reduced to a single FPRT given a well-trained network.

In the revised **Supplementary information**, we have modified and added the following sentences:

**“The reconstruction was implemented with MATLAB on a 24-core 3-GHz Intel Xeon Gold CPU. If all the**
**optimal hyperparameters (i.e., ζ_i) are known, then the fixed-parameter reconstruction time (FPRT) will be**
**as short as < 1 s. When ζ_1 and ζ_2 are free to optimize, the time cost will increase due to the CV procedure. In**

the worst case, when the full 8×8 searching space must be traversed, the total reconstruction time is < 60 s.
There are several strategies to expedite the reconstruction. First, since the iterative least-squares solver
relies heavily on matrix multiplication, the reconstruction can be drastically accelerated by employing a GPU.
Second, it is possible to train a deep-learning network to identify spectral features directly from the
interferogram and determine ζ_i without CV^{S25} , which may reduce the reconstruction period to a single FPRT.”
**(Supplementary information, Note 8, lines 684-692)**

Table R4. Comparison of reported integrated spectrometers.

Design	Platform	Footprint [μm^2]	$\delta\lambda_{\text{res}}$ [pm] ^(a)	BW [nm] ^(b)	P_a [mW]	N_{ch} ^(c)
AWG ^{R34}	SOI	8000×8000	150	7.5	/	50
EDG ^{R35}	SOI	6000×9000	500	30	/	60
EDG ^{R36}	SOI	2×10 ¹²	150	148	/	926
MRR ^{R37}	SOI	1×10 ⁶	600	50	/	81
PhC cavity ^{R38}	SOI	6×111	1000	35	/	38
PhC cavity ^{R39}	SOI	18×250	160	16	30×3	101
WBG ^{R40}	SOI	3.6×10 ⁴	510	102.7	873	201
AWG/MRR ^{R41}	SOI	3000×3000	100	25.4	30×9	255
AWG/MRR ^{R42}	SNOI	1150×1250	750	57.5	/	70
AWG/MRR ^{R43}	SOI	200×270	200	70	NM ^(d)	350
MRR ^{R32}	SOI	3.5×10 ⁵	5	10	50×10	1941
MRR ^{R44}	SOI	20×35	80	100	45	1251
MRR ^{R45}	SOI	60×60	40	100	75	2501
MDR ^{R46}	SOI	200×200	200	20	160	101
SS ^{R47}	SOI	100×50	600	25	/	42
SS ^{R48}	SNOI	200×50	3400	40	/	13
SS ^{R49}	SOI	12.8×30	250	30		121
SS ^{R50}	SOI	35×260	450	180	/	401
SS ^{R23}	SOI	500×500	10	2	/	332
SS ^{R51}	SOI	1600×2100	16	2	/	126
SS ^{R52}	SOI	> 2000×2000	100	6.3	/	64
SS ^{R53}	SOI	> 2000×2000	20	100	64×6	5001
SS ^{R28}	SNOI	220×520	20	12	/	600
SS ^{R29}	SOI	88×300	100	120	/	160
SS ^{R19}	SOI	1000×1000	15	40	/	> 2500
SS ^{R30}	SOI	2000×7600	30	115	NM ^(d)	> 3800
SS ^{R54}	SOI	1000×1500	5	100	50×14/2 ^(e)	2×10 ⁴
SS ^{R24}	SOI	310×215	200	60	33	301
SHS ^{R1}	Silica	2500×4300	320	5.12	/	16
SHS ^{R21}	Silica	NM ^(d)	16	0.512	/	32
SHS ^{R11}	SOI	NM ^(d)	50	≈ 0.8	/	16
SHS ^{R4,5}	SOI	1.2×10 ⁷	40	≈ 0.64	/	16
SHS ^{R22}	SOI	NM ^(d)	48	0.78	/	16
SHS ^{R55}	SOI	1100×1500	6000	600	/	100

SHS ^{R26}	SNOI	NM ^(d)	≈ 49	≈ 0.39	/	8
SHS ^{R56}	SOI	7100×18000	40000 ^(e)	400 ^(e)		≈ 10 ^(f)
SHS ^{R27}	SNOI	650×4700	5000 ^(f)	400 ^(f)	/	> 80 ^(g)
SHS ^{R57}	SNOI	4600×5800	400	≈ 6.4	/	16
tFTS ^{R58}	LNOI	NM ^(d)	≈ 70000	450	NM ^(d)	> 6
tFTS ^{R3}	SOI	1×10 ⁶	3000	> 50	5100	≈ 20
tFTS ^{R59}	SOI	NM ^(d)	320 ^(g)	180 ^(g)	5000	> 560 ^(h)
dFTS ^{R6}	SOI	NM ^(d)	400	> 20	33×6	32
dFTS ^{R7}	SOI	2500×3500	50	3.2	30×7	≈ 64
SWIFTS ^{R60}	SOI	22×512	4000	96	/	25
SWIFTS ^{R61}	SNOI	1×10 ⁵	6000	> 100	/	> 16
SWIFTS ^{R62}	LNOI	1×10 ⁷	≈ 5000	500	NM ^(d)	101
tFTS & MRR ^{R31}	SOI	NM ^(d)	470 ⁽ⁱ⁾	90	> 1800	> 190
This work	SOI	5500×6000	250 125 ⁽ⁱ⁾	200	2.4	801 1601 ⁽ⁱ⁾

The blue shade represents all reported Fourier-transform spectrometers. The grey shade represents other types of spectrometers. The
red shade represents this work.

AWG, arrayed waveguide grating.

EDG, echelle diffraction grating.

MRR, microring resonator.

MDR, microdisk resonator.

PhC, photonic crystal.

WBG, waveguide Bragg grating.

SS, speckle spectrometer.

FTS, Fourier-transform spectrometer.

SHS, spatial heterodyne spectrometer.

tFTS, tunable FTS.

dFTS, digital FTS.

SWIFTS, stationary-wave integrated FTS.

SOI, silicon on insulator.

SNOI, silicon nitride on insulator.

LNOI, lithium niobate on insulator.

$\delta\lambda_{\text{res}}$, spectral resolution.

BW, working bandwidth.

N_{ch} , channel capacity defined by $N_f = \text{BW}/\Delta\lambda_{\text{res}}$.

P_a , aggregate power consumption.

1480 ^(a) $\delta\lambda_{\text{res}}$ is determined based on the Rayleigh criterion (i.e., $\delta\lambda_f$).

1481 ^(b)BW is derived using $\text{BW} = N_{\text{ch}}\delta\lambda_{\text{res}}$.

1482 ^(c)For the Fourier-transform spectrometers in which the number of Mach-Zehnder interferometers (MZI) is stated, N_{ch} is half of the MZI
number; otherwise, N_{ch} is derived from half of the grid number in the computationally reconstructed spectrum. For other spectrometer
schemes, N_{ch} is defined as the bandwidth-to-resolution ratio.

1485 ^(d)Not mentioned.

- 1486 ^(e)Since the applied heating power is a random sequence with the average of 0.5, the power consumption is halved here.
- 1487 ^(f)Operated at mid-infrared wavelengths.
- 1488 ^(g)Several narrow bands are bridged to form the whole BW by changing the polarization and incident angle of light.
- 1489 ^(h)Three parallel Michelson interferometers (MI) are employed to improve $\delta\lambda_{\text{res}}$ and expand BW.
- ⁽ⁱ⁾Here, $\delta\lambda_{\text{res}}$ is defined as the linewidth of the MRR.
- 1491 ^(j)The resolution and capacity can be improved to $\delta\lambda_{2f} = \delta\lambda_f/2$ and $N_{2f} = 2N_f$, respectively, using the computational method.

**Appendix B: additional modifications**

1. The spot trajectory shown in Figure 1d is modified:

2. The following sentences have been modified and added in the revised manuscript:

“The shift direction relies on the phases of sinusoidal responses.” (**Design principle**, caption of Figure
1, line 88)

“The shift direction along f_{tFTS} depends on the initial phase of the sinusoidal response of the tFTS at the
first sweep step. In this work, the spot on the blue end shifts towards $f_{tFTS} = 0$.” (**Design principle**, lines 119-
121)

“Each folded segment contains the spectral information within a single free spectral range (FSR_{SHS}) of
the SHS.” (**Design principle**, lines 121-122)

“To identify adjacent FSR_{SHS} , the tFTS must have a resolution ($\delta\lambda_{tFTS}$) finer than FSR_{SHS} , which yields:”
(**Design principle**, lines 124-125)

“The abbreviations and notations used in this work are summarized in **Supplementary information**,
**Note 2.**” (**Design principle**, lines 147-148)

**References:**

- R1. Okamoto K, Aoyagi H, Takada K. Fabrication of Fourier-transform, integrated-optic spatial heterodyne
spectrometer on silica-based planar waveguide. *Opt Lett* **35**, 2103-2105 (2010).
- R2. Grotevent MJ, *et al.* Integrated photodetectors for compact Fourier-transform waveguide
spectrometers. *Nat Photon* **17**, 59-64 (2023).
- R3. Souza M, Grieco A, Frateschi NC, Fainman Y. Fourier transform spectrometer on silicon with thermo-
optic non-linearity and dispersion correction. *Nat Commun* **9**, 665 (2018).
- R4. Velasco AV, *et al.* High-resolution Fourier-transform spectrometer chip with microphotonic silicon
spiral waveguides. *Opt Lett* **38**, 706-708 (2013).
- R5. Herrero-Bermello A, *et al.* On-chip Fourier-transform spectrometers and machine learning: a new
route to smart photonic sensors. *Opt Lett* **44**, 5840-5843 (2019).
- R6. Kita DM, *et al.* High-performance and scalable on-chip digital Fourier transform spectroscopy. *Nat*
*Commun* **9**, 4405 (2018).
- R7. Du J, *et al.* High-resolution on-chip Fourier transform spectrometer based on cascaded optical
switches. *Opt Lett* **47**, 218-221 (2022).
- R8. Jensen TL, Jørgensen JH, Hansen PC, Jensen SH. Implementation of an optimal first-order method for
strongly convex total variation regularization. *BIT Numerical Mathematics* **52**, 329-356 (2011).
- R9. Becker S, Bobin J, Candès EJ. NESTA: a fast and accurate first-order method for sparse recovery. *SIAM*
*Journal on Imaging Sciences* **4**, 1-39 (2011).
- R10. Hamming RW. *Digital filters*. Courier Corporation (1998).
- R11. Bock PJ, *et al.* Subwavelength grating Fourier-transform interferometer array in silicon-on-insulator.
*Laser Photon Rev* **7**, L67-L70 (2013).
- R12. Hansen PC. *Rank-deficient and discrete ill-posed problems: numerical aspects of linear inversion*. SIAM
(1998).
- R13. Hansen PC, Nagy JG, O'leary DP. *Deblurring images: matrices, spectra, and filtering*. SIAM (2006).
- R14. Buccini A, Donatelli M, Reichel L. Iterated Tikhonov regularization with a general penalty term.
*Numerical Linear Algebra with Applications* **24**, e2089 (2017).
- R15. Golub GH, Heath M, Wahba G. Generalized cross-validation as a method for choosing a good ridge
parameter. *Technometrics* **21**, 215-223 (1979).
- R16. Gazzola S, Hansen PC, Nagy JG. IR Tools: a MATLAB package of iterative regularization methods and
large-scale test problems. *arXiv preprint arXiv:171205602* (2017).
- R17. Ravasi M, Vasconcelos I. PyLops—A linear-operator Python library for scalable algebra and
optimization. *SoftwareX* **11**, 100361 (2020).
- R18. Hansen PC. The discrete Picard condition for discrete ill-posed problems. *BIT Numerical Mathematics*
**30**, 658-672 (1990).
- R19. Lin Z, *et al.* High-performance, intelligent, on-chip speckle spectrometer using 2D silicon photonic

disordered microring lattice. *Optica* **10**, 497 (2023).

R20. Filip S, Javeed A, Trefethen LN. Smooth random functions, random ODEs, and Gaussian processes.
*SIAM Review* **61**, 185-205 (2019).

R21. Fontaine NK, Okamoto K, Su T, Yoo SJ. Fourier-transform, integrated-optic spatial heterodyne
spectrometer on a silica-based planar waveguide with 1 GHz resolution. *Opt Lett* **36**, 3124-3126 (2011).

R22. Podmore H, *et al.* Demonstration of a compressive-sensing Fourier-transform on-chip spectrometer.
*Opt Lett* **42**, 1440-1443 (2017).

R23. Redding B, Fatt Liew S, Bromberg Y, Sarma R, Cao H. Evanescently coupled multimode spiral
spectrometer. *Optica* **3**, 956-962 (2016).

R24. Sun C, *et al.* Integrated microring spectrometer with in-hardware compressed sensing to break the
resolution-bandwidth limit for general continuous spectrum analysis. *Laser Photon Rev*, 2300291 (2023).

R25. Wang Y, *et al.* Polarization-independent mode-evolution-based coupler for the silicon-on-insulator
platform. *IEEE Photon J* **10**, 1-10 (2018).

R26. Gonzalez-Andrade D, *et al.* Broadband Fourier-transform silicon nitride spectrometer with wide-area
multiaperture input. *Opt Lett* **46**, 4021-4024 (2021).

R27. Yoo KM, Chen RT. Dual-polarization bandwidth-bridged bandpass sampling Fourier transform
spectrometer from visible to near-infrared on a silicon nitride platform. *ACS Photon* **9**, 2691-2701 (2022).

R28. Zhang Z, Li Y, Wang Y, Yu Z, Sun X, Tsang HK. Compact high resolution speckle spectrometer by using
linear coherent integrated network on silicon nitride platform at 776 nm. *Laser Photon Rev* **15**, 2100039
(2021).

R29. Li Y, *et al.* Inverse-designed linear coherent photonic networks for high-resolution spectral
reconstruction. *ACS Photon* **10**, 1012-1018 (2023).

R30. Yao C, Chen M, Yan T, Ming L, Cheng Q, Penty R. Broadband picometer-scale resolution on-chip
spectrometer with reconfigurable photonics. *Light Sci Appl* **12**, 156 (2023).

R31. Zheng SN, *et al.* Microring resonator-assisted Fourier transform spectrometer with enhanced
resolution and large bandwidth in single chip solution. *Nat Commun* **10**, 2349 (2019).

R32. Zhang L, Zhang M, Chen T, Liu D, Hong S, Dai D. Ultrahigh-resolution on-chip spectrometer with
silicon photonic resonators. *Opto-Electronic Advances* **5**, 210100-210100 (2020).

R33. Espinola RL, Tsai MC, Yardley JT, Osgood RM. Fast and low-power thermo-optic switch on thin silicon-
on-insulator. *IEEE Photon Technol Lett* **15**, 1366-1368 (2003).

R34. Cheben P, *et al.* A high-resolution silicon-on-insulator arrayed waveguide grating microspectrometer
with sub-micrometer aperture waveguides. *Opt Express* **15**, 2299-2306 (2007).

R35. Xiao M, Mingyu L, Jian-Jun H. CMOS-compatible integrated spectrometer based on Echelle diffraction
grating and MSM photodetector array. *IEEE Photon J* **5**, 6600807-6600807 (2013).

R36. Calafiore G, *et al.* Holographic planar lightwave circuit for on-chip spectroscopy. *Light Sci Appl* **3**,
e203-e203 (2014).

R37. Xia Z, *et al.* High resolution on-chip spectroscopy based on miniaturized microdonut resonators. *Opt*
*Express* **19**, 12356-12364 (2011).

R38. Cheng Z, *et al.* Generalized modular spectrometers combining a compact nanobeam microcavity and
computational reconstruction. *ACS Photon* **9**, 74-81 (2021).

R39. Zhang J, Cheng Z, Dong J, Zhang X. Cascaded nanobeam spectrometer with high resolution and
scalability. *Optica* **9**, 517-521 (2022).

R40. Sun C, *et al.* Broadband and high-resolution integrated spectrometer based on a tunable FSR-free
optical filter array. *ACS Photon* **9**, 2973-2980 (2022).

R41. Zheng S, *et al.* A single-chip integrated spectrometer via tunable microring resonator array. *IEEE*
*Photon J* **11**, 1-9 (2019).

R42. Zhang Z, Wang Y, Tsang HK. Tandem configuration of microrings and arrayed waveguide gratings for a
high-resolution and broadband stationary optical spectrometer at 860 nm. *ACS Photon* **8**, 1251-1257
(2021).

R43. Zhang Z, *et al.* Integrated scanning spectrometer with a tunable micro-ring resonator and an arrayed
waveguide grating. *Photon Res* **10**, A74 (2022).

R44. Xu HN, Qin Y, Hu GL, Tsang HK. Integrated single-resonator spectrometer beyond the free-spectral-
range limit. *ACS Photon* **10**, 654-666 (2023).

R45. Xu H, Qin Y, Hu G, Tsang HK. Breaking the resolution-bandwidth limit of chip-scale spectrometry by
harnessing a dispersion-engineered photonic molecule. *Light Sci Appl* **12**, 64 (2023).

R46. Sun CL, *et al.* Scalable on-chip microdisk resonator spectrometer. *Laser Photon Rev* **17**, 2200792
(2023).

R47. Redding B, Liew SF, Sarma R, Cao H. Compact spectrometer based on a disordered photonic chip. *Nat*
*Photon* **7**, 746-751 (2013).

R48. Hartmann W, *et al.* Waveguide-integrated broadband spectrometer based on tailored disorder.
*Advanced Optical Materials* **8**, 1901602 (2020).

R49. Hadibrata W, Noh H, Wei H, Krishnaswamy S, Aydin K. Compact, high-resolution inverse-designed on-
chip spectrometer based on tailored disorder modes. *Laser Photon Rev* **15**, 2000556 (2021).

R50. Li A, Fainman Y. On-chip spectrometers using stratified waveguide filters. *Nat Commun* **12**, 2704
(2021).

R51. Piels M, Zibar D. Compact silicon multimode waveguide spectrometer with enhanced bandwidth. *Sci*
*Rep* **7**, 43454 (2017).

R52. Yi D, Zhang Y, Wu X, Tsang HK. Integrated multimode waveguide with photonic lantern for speckle
spectroscopy. *IEEE J Quant Electron* **57**, 1-8 (2021).

R53. Yi D, Tsang HK. High-resolution and broadband two-stage speckle spectrometer. *J Lightwave Technol*
**40**, 7969-7976 (2022).

R54. Xu H, Qin Y, Hu G, Tsang HK. Cavity-enhanced scalable integrated temporal random-speckle

spectrometry. *Optica* **10**, 1177-1188 (2023).

R55. Yang M, Li M, He JJ. Static FT imaging spectrometer based on a modified waveguide MZI array. *Opt*
*Lett* **42**, 2675-2678 (2017).

R56. Duong Dinh TT, *et al.* Mid-infrared Fourier-transform spectrometer based on metamaterial lateral
cladding suspended silicon waveguides. *Opt Lett* **47**, 810-813 (2022).

R57. Lu L, Zhang H, Li X, Chen J, Zhou L. Low temperature sensitivity on-chip Fourier-transform
spectrometer based on dual-layer Si₃N₄ spiral waveguides. *Photon Res* **11**, 591-599 (2023).

R58. Li J, Lu DF, Qi ZM. Miniature Fourier transform spectrometer based on wavelength dependence of
half-wave voltage of a LiNbO₃ waveguide interferometer. *Opt Lett* **39**, 3923-3926 (2014).

R59. Li A, Fainman Y. Integrated silicon Fourier transform spectrometer with broad bandwidth and ultra-
high resolution. *Laser Photon Rev* **15**, 2000358 (2021).

R60. le Coarer E, *et al.* Wavelength-scale stationary-wave integrated Fourier-transform spectrometry. *Nat*
*Photon* **1**, 473-478 (2007).

R61. Nie X, Ryckeboer E, Roelkens G, Baets R. CMOS-compatible broadband co-propagative stationary
Fourier transform spectrometer integrated on a silicon nitride photonics platform. *Opt Express* **25**, A409-
A418 (2017).

R62. Pohl D, *et al.* An integrated broadband spectrometer on thin-film lithium niobate. *Nat Photon* **14**, 24-
29 (2019).

REVIEWER COMMENTS

Reviewer #1 (Remarks to the Author):

The authors addressed most of the comments giving extensive answers and providing additional data to support their study. This surely helped clarify many aspects. However, there are still some concerns and some further comments listed below separated in sections relative to their division of comments:

Comments 1-1

In general, the authors explained well the reconstruction procedure, especially with their response. The explanation is very long indeed, but needed to explain the procedure and their implementation. It is clear that the computational methods works, and it is now convincing that there is no need to know any detail about the spectrum a-priori, but a visual connection between the experimental raw data and the final outcome of the reconstruction process is still missing (i.e. experimental data points superimposed to the reconstructed spectrum after optimization), especially for narrow spectral features. Coefficients of determinations show a nearly perfect fit to the reference spectra (the slight deviations from the new figure R8 should also be available to the readers and not only the reviewers, for instance in the supplementary section), but the lack of experimental data-points makes it anyway impossible to distinguish between actual experimental results or numerical tests.

Comments 1-3

Obviously both amplitude and phase are needed to reconstruct the spectrum, and phase is actually more important than amplitude when performing Fourier analysis. However, amplitude is typically of much more intuitive interpretation (also in this case, see for instance the spots in the 2D Fourier space correspondent to laser lines in Fig. R9, they clearly represent very distinct frequencies), and that is the reason why that is typically what is shown and commented, while phase is overlooked. Maybe the question was not well formulated: showing the phase profile of each Fourier transform seems to overcrowd the figures with information that is not commented in the text. What value does it add to the discussion? If shown, it should be described more thoroughly (highlighting some peculiar feature for instance), otherwise, any reader interested in this topic would know that a Fourier transform is a complex number and it is not necessary to specify it.

Comment 1-5

A possible explanation for the defects in the patterned circuits: they may be introduced in the lithography step in form of stitching errors (by the beam itself if waveguides are written with electron-beam lithography, or present in the mask if DUV lithography is used).

Comments 1-6

For a proof-of-concept this approach is valid even though it is clearly slow and cumbersome to perform, especially considering that the experiment needs to be performed many times for different sources and spectra. Each port needs a few minutes for the collection of the signal, including sweeping the heater power. Let's say this "a few minutes" is 5, for 128 outputs this means around 10h for acquiring the data necessary to retrieve a spectrum? And with manual movement of the fibre required.

The claim that a straightforward solution to this is to monolithically integrate PDs on the platform to have simultaneous data acquisition is very strong and in disagreement with what is mentioned in previous paragraph. Monolithic integration would mean fabricating PDs on silicon for efficient detection at 1550 nm, which is possible but not straightforward and not conventional. The proposed solution is put in a too simplistic way and a more thorough discussion on the possible implementation is required, including references.

Comments 1-7

Regarding the rephrased sentence about SWIFTS, again the statement is partially incorrect, since in the work cited as Ref. 32, electro-optic tuning of the device allowed to achieve broad bandwidth with a fairly low density of samplers thanks to active shifting of the waveform.

Reviewer #2 (Remarks to the Author):

I appreciate the authors' efforts to address my concern and I agree that using this hybrid architecture could release the critical problems of previously demonstrated FTS. However, I am not convinced by the value and the novelty of this work brought to the community of spectrometer. Technically, this work utilizes mature SHS and thermal tuning FTS technique. In terms of performance, the incident light has to be split into 128 channels, leading to very low intensities at each detector. Moreover, the spectrometer has to consume over 2.5 Watts power, which is impractical for the working scenarios of integrated spectrometers. Despite improvement upon certain aspects compared with standalone SHS or thermal tuning FTS, this work couldn't notably advance the research or applications of spectrometers, in my viewpoint.

Reviewer #3 (Remarks to the Author):

The authors have adequately addressed my concerns. Consequently I recommend publication of the manuscript.

To Reviewer #1's comments:

We thank the reviewer for his/her reading and comments. In the following sections, we will give a point-to-point response to address the reviewer's concerns. The modifications to the manuscript are highlighted in red. The crucial descriptions are shaded in gray.

Comment 1-1:

In general, the authors explained well the reconstruction procedure, especially with their response. The explanation is very long indeed but needed to explain the procedure and their implementation. It is clear that the computational method works, and it is now convincing that there is no need to know any detail about the spectrum a-priori, but a visual connection between the experimental raw data and the final outcome of the reconstruction process is still missing (i.e., experimental data points superimposed to the reconstructed spectrum after optimization), especially for narrow spectral features. Coefficients of determinations show a nearly perfect fit to the reference spectra (the slight deviations from the new Figure R8 should also be available to the readers and not only the reviewers, for instance in the supplementary section), but the lack of experimental data-points makes it anyway impossible to distinguish between actual experimental results or numerical tests.

Reply and modifications:

1. Actually, Figure R8 in the *prior* response letter has already been added to the revised **Supplementary information**, Note 9 (see Figure S21) in the *last* revision cycle.
2. It is assumed that the stated "experimental raw data" pertains to the recorded interferograms. We agree that the measurements of interferograms must be included to distinguish them from the results shown in Figures 4d-4g. However, the interferogram is a 2D patterns with variations in length differences and heating power, whereas the reconstructed spectrum is a 1D vector as a function of wavelengths. Hence, it is impossible to "superimpose" an interferogram to a spectrum. We have added the following figures to **Supplementary information**, Note 9, in which interferograms are positioned in parallel with the corresponding reconstruction results. As there are too many reconstruction examples in this work, the results are displayed in two independent figures:

Figure S19 Extended experimental data (part I). Left panels: experimentally measured interferograms. Right panels: reconstructed spectra. Insets: enlarged views of spectra around the wavelength ranges indicated by the arrows. The two-dimensional interferograms and retrieved spectra shown in Figures S19a-S19h correspond to the results shown in Figures 5a-5b.

(Supplementary information, Note 9, lines 770-773)

Figure S20 Extended experimental data (part II). Left panels: experimentally measured interferograms. Right panels: reconstructed spectra. Insets: enlarged views of spectra around the wavelength ranges indicated by the arrows. The two-dimensional interferograms and retrieved spectra shown in Figures S20a-S20h correspond to the results shown in Figures 5b-5h.

(Supplementary information, Note 9, lines 774-777)

In the revised manuscript, we have added the following sentence:

“The recorded interferograms are shown in **Supplementary information**, Figures S19 and S20.”

(Spectrum reconstruction, lines 321-322)

A minor mistake of axis labels in Figure S21 has also been corrected.

Comment 1-2:

Obviously, both amplitude and phase are needed to reconstruct the spectrum, and phase is actually more important than amplitude when performing Fourier analysis. However, amplitude is typically of much more

intuitive interpretation (also in this case, see for instance the spots in the 2D Fourier space correspondent to laser lines in Figure R9, they clearly represent very distinct frequencies), and that is the reason why that is typically what is shown and commented, while phase is overlooked. Maybe the question was not well formulated: showing the phase profile of each Fourier transform seems to overcrowd the figures with information that is not commented in the text. What value does it add to the discussion? If shown, it should be described more thoroughly (highlighting some peculiar feature for instance); otherwise, any reader interested in this topic would know that a Fourier transform is a complex number, and it is not necessary to specify it.

Reply and modifications:

We thank the reviewer for his/her clarification. To rephrase this remark, the reviewer requests us to specify whether a distinguishable feature in phase exists at the spot location of the intensity distribution of a Fourier map. Figure R1 shows the *simulation* results of intensity and phase distributions of a Fourier map. The FFT results are derived from a spectrum with a spike at $\lambda = 1.51 \mu\text{m}$. It can be clearly observed that, for a single-peak input, the phase map exhibits an abrupt discontinuity in each quadrant. Moreover, the location of phase hopping is precisely the spot location in the intensity map, demonstrating that the phase map also carries spectral information. Figure R2 is a truncation of Figure 4 in the main manuscript, showing the Fourier maps derived from the *measured* transmittance cube. The phase hopping becomes less visible in Figure R2d since the perturbation during measurement imposes a chaotic background on the phase map. Nevertheless, the impact of noise components can be mitigated using the iterative method discussed previously.

Figure R1 Phase hopping. (a) Intensity and (b) phase distributions of the fast Fourier transform (FFT) result before removing zero-frequency components. (a) Intensity and (b) phase distributions of the FFT result after the component removal. The input is a single spike at the 1.51- μm wavelength. The arrows highlight the phase hopping in the FFT maps.

Figure R2 Fourier maps. (a) Testing spectrum with three spikes. (b) Interferogram (**O**) derived from the measured cube and testing spectrum. (f) Intensity and (g) phase distributions of FFT(**O**).

Based on the simulation results, we confirm that the phase map does indeed contain peculiar features. It is also essential to explain the degraded visibility of phase hopping in the experimental results. **Figure R1** has already been presented in **Supplementary information**, Note 6 (see Figure S10). We have added some arrows to highlight the hopping location. In the revised manuscript, we have added the following sentences:

“The simulation results show that the phase distribution of FFT(**O**) has an abrupt discontinuity at the spot location (see **Supplementary information**, Figures S10b and S10d, for instance), indicating that the phase map also carries information. Such a phase hopping is less visible in the experimental results (see Figure 4g) since the environmental perturbation during measurement imposes a chaotic background on the phase map. Nevertheless, the impact of noise components can be mitigated using the iterative optimization method.” (**Characterization and analysis of the spectrometer**, lines 265-270)

In the revised **Supplementary information**, we have added the following sentences:

“Remarkably, for a single-peak input, the phase distribution of the FFT map exhibits a hopping in each quadrant (see the arrows in Figures S10b and S10d), and the hopping location is precisely the spot location in the intensity map. Such a phase hopping is less visible in the experimental results (see Figure S4g for instance) due to the presence of noises.” (**Supplementary information**, Note 6, lines 494-497)

Comment 1-3:

A possible explanation for the defects in the patterned circuits: they may be introduced in the lithography step in form of stitching errors (by the beam itself if waveguides are written with electron-beam lithography, or present in the mask if DUV lithography is used).

Reply and modifications:

We thank the reviewer for his/her insightful advice. We agree that the stitching error may serve as a possible explanation for defects and be added to **Supplementary information**:

“These defects may result from the stitching errors between writing fields during electron-beam lithography.” (**Supplementary information**, Note 5, lines 275-276)

Comment 1-4:

For a proof-of-concept this approach is valid even though it is clearly slow and cumbersome to perform, especially considering that the experiment needs to be performed many times for different sources and spectra. Each port needs a few minutes for the collection of the signal, including sweeping the heater power. Let's say this “a few minutes” is 5, for 128 outputs this means around 10 h for acquiring the data necessary to retrieve a spectrum? And with manual movement of the fiber required.

The claim that a straightforward solution to this is to monolithically integrate PDs on the platform to

have simultaneous data acquisition is very strong and in disagreement with what is mentioned in previous paragraph. Monolithic integration would mean fabricating PDs on silicon for efficient detection at 1550 nm, which is possible but not straightforward and not conventional. The proposed solution is put in a too simplistic way and a more thorough discussion on the possible implementation is required, including references.

Reply and modifications:

1. In the experiment, each measurement step includes the acquisitions of interferograms for *all* the testing spectra together with the corresponding column in the transmittance cube at one of the output channels. In other words, we moved the fiber probe once and then captured the signals for totally 16 spectra (as shown in Figure 5) before moving it again. All light paths were connected by fibers and all optical and electric sources were controlled by a host computer, enabling automatic switch of optical input and fast wavelength/power sweep. The time consumption is for 16 spectra as opposed to one spectrum, because the acquisition at each step (≈ 80 s) covers all the spectrum examples, and we only need to perform the measurement *once* from 1st to 128th ports. In the experiment, the *average* period per spectrum is ≈ 10 min. The principle of this method has been thoroughly discussed in the *prior* response letter (see Figure R14), and the related contents have been modified. This proof-of-concept method is widely applied, and the reviewer has agreed on its validity.
2. As pointed out by the reviewer, the detection wavelength (≈ 1.55 μm) is within the transparent window of silicon. However, it is commonly known that PDs can be monolithically integrated on silicon circuits via epitaxy growth of germanium. The first waveguide-based silicon-germanium PD was demonstrated in 2007 by Laurent Vivien et. al.^{R1} The more prevalent lumped PD design was demonstrated in 2009 by the same group^{R2}. After years of research and development, Si-Ge PDs have become a *standard* building block in silicon nanophotonic circuits and are supported by most *commercial* silicon photonic foundries in their multi-project wafer (MPW) service, such as IMEC^{R3}, AMFR^{R4}, AIM^{R5}, and Cornerstone^{R4}. Table R1 summarizes the performance of process-design-kit (PDK) devices in different foundries. This table is a snapshot from Ref. R6. Foundry-fabricated Si-Ge PDs typically have a large electric bandwidth of > 20 GHz and a high responsivity of > 1 A/W, as shown in the third column of Table R1. The first PD-integrated silicon spectrometer was reported in 2013 by Xiao Ma et. al.^{R7} In this research, totally 60 Si-Ge PDs are integrated to a arrayed waveguide grating (AWG). Recently, the ultra-fast spectrum acquisition has also been realized using a ring-AWG architecture with monolithic Si-Ge PDs^{R8}. We did not integrate PDs to our spectrometer chip since we chose to use a “passive-plus” foundry (Applied Nanotools Inc.^{R9}) for fast prototyping. Nevertheless, it should be easy to transfer our scheme to a PD-integrated implementation.

Institution (Platform)	Mach-Zehnder Modulators	SiGe Detector	Passive Devices	Edge Coupling
AMF (MPW)	BW: > 40 GHz @ -2V IL: 4 dB V π : 6.29V Length: 4 mm	BW: 35 GHz @ -2V R: 0.85 A/W ID: 30 nA Bias: -2V	Loss: 1.4 dB/cm (500 nm \times 220 nm Rib) 1 \times 2 MMI: 0.3 dB	1.2 dB SMF
AIM (MPW)	Speed: 50 Gbps IL: 2.7 dB V π L: 0.8 V-cm @ -1V Length: 1 mm	BW: 50 GHz R: 1 A/W ID: 40 nA Bias: 1V	Loss: 0.9 dB/cm (Low Loss Rib)	1.5 dB SMF
CEA-Leti (Si310-PHMP2M)	BW: 40GHz @ -2V IL: 2.4 dB V π L: 2 V-cm Length: 3 mm	BW: 35 GHz R: 0.75 A/W ID: 50 nA Bias: -1V	Loss: 2dB/cm (Rib) 1 \times 2MMI: 0.5 dB	-
GF (90WG)	BW: 27 BHz @ -2V IL: 3.5 dB V π L: 1.54 V-cm Length: 3 mm	BW: 39 GHz R: 1 A/W ID: 70 nA Bias: -1V	Loss: 2.2 dB/cm (170 nm \times 350 nm Rib) Multimode Rib	0.7 dB Metamaterial waveguide SMF
IHP (SG25H5-ePIC)	BW: - IL: - V π L: 3.0 V-cm Length: 7 mm	BW: 60 GHz R: 0.8 A/W ID: 200 nA Bias: -2V	Loss: 2.5 dB/cm (500 nm \times 220 nm Rib)	-
IMEC (iSiPP50)	BW: 37 GHz @ -2V IL: 2.5 dB V π : 12V Length: 1.5 mm	BW: 50 GHz R: 0.9 A/W ID: 50 nA Bias: -1V	Loss: 1.4 dB/cm (450 nm \times 220 nm Rib)	2 dB Lensed fiber
VTT (MPW)	BW: Up to 2.5 MHz	BW: 35 GHz R: 0.8 A/W ID: 1 μ A Bias: -1V	Loss: 0.1 dB/cm (3 μ m \times 3 μ m Strip) 1 \times 2 MMI: 0.2 dB	1 – 1.4 dB SMF

Table R1 Performance comparison of process-design-kit (PDK) devices in different silicon photonic foundries. This table is a snapshot from Ref. R6.

- Based on the analysis above, we give an estimate of sampling period (excluding the optimization time) per spectrum with the use of monolithic PDs. As discussed, the rise/fall time of the tunable section is < 100 μ s (see **Comment 3-2** in the *prior* response letter). An analog-to-digital converter typically has 16 channels, so it requires 8 conversion steps for 128 output ports. Since Si-Ge PDs can operate at a high speed (> 20 GHz), it is feasible to set the sampling time as \approx 1 ms at each power sweep step containing 8 times of electric-signal readout. **With 25 sweep steps, the theoretical sampling period per spectrum is thus estimated to be \approx 0.025s.**
- The reviewer’s claim on “disagreement with what is mentioned in previous paragraph” is confusing to us. We presume that this statement may relate to our overview description of SWIFT in **Introduction**. **The PD mentioned in Ref. R10 is embedded in a waveguide to capture the field distribution of a standing wave. In contrast, Si-Ge PDs are integrated at the terminal of a waveguide, as a receiver of total output power, which is easier to realize in foundry processes.** For clarity, we have replaced the term “monolithic PDs” with “embedded PDs” in the revised introductory overview (see **Comment 1-5** on the next page). In the revised manuscript, we have added the following sentence:

“Practical multiport acquisition can be realized by integrating silicon-germanium PDs to each output channel and reading their signals under a synchronized clock⁵⁰. The monolithic integration of PDs can be supported by most commercial silicon photonic foundries⁵¹. Integrated PDs typically have an electric bandwidth of > 20 GHz⁵¹, so it is feasible to capture signals at all ports within the theoretical time span of a tuning step (< 1 ms). With 25 tuning steps, the theoretical sampling period per spectrum is < 0.025 s.” **(Measurement details, lines 381-385)**

The reference list has also been updated:

“50. Zhang Z, Rony KTA, Wang Y, Cheng Z, Tsang HK. Integrated spectrometer with fast wavelength scanning using current injection in PIN diode. In: *2023 Opto-Electronics and Communications Conference (OECC) (2023)*.

51. Siew SY, *et al.* Review of silicon photonics technology and platform development. *J Lightwave Technol* **39**, 4374-4389 (2021).” (**References**, lines 507-511)

Other discussions regarding sampling period have already been added to the manuscript in response to Reviewer #3’s **Comment 3-2** in the *prior* response letter.

Comment 1-5:

Regarding the rephrased sentence about SWIFTS, again the statement is partially incorrect, since in the work cited as Ref. 32, electro-optic tuning of the device allowed to achieve broad bandwidth with a fairly low density of samplers thanks to active shifting of the waveform.

Reply and modifications:

We thank the reviewer for his/her careful reading. In the revised manuscript, we have deleted the sentence regarding sampling density and simplified the description as follows. For clarity, the term “monolithic PDs” has been replaced with “embedded PDs”.

“However, it remains challenging for a SWIFT to probe the field distribution of a guided mode even with embedded PDs.” (**Introduction**, lines 50-51)

To Reviewer #2's comments:

General remark:

I appreciate the authors' efforts to address my concern and I agree that using this hybrid architecture could release the critical problems of previously demonstrated FTS. However, I am not convinced by the value and the novelty of this work brought to the community of spectrometry. Technically, this work utilizes mature SHS and thermal tuning FTS technique. In terms of performance, the incident light has to be split into 128 channels, leading to very low intensities at each detector. Moreover, the spectrometer has to consume over 2.4 Watts power, which is impractical for the working scenarios of integrated spectrometers. Despite improvement upon certain aspects compared with standalone SHS or thermal-tuning FTS, this work could not notably advance the research or applications of spectrometers, in my viewpoint.

Reply:

The reviewer #2's three remarks (i.e., etendue issue, power efficiency, and scheme novelty) have been raised and replied in the *prior* review cycle. In the *prior* response letter (see pages 32-48), we have provided two additional design examples and substantial revisions to address these issues (see pages 40-55 in the revised **Supplementary information**). After *previous* revision, it should be clear that the concept of 2D-FTS can be extended to a *single*-port configuration with a low power of < 120 mW. Using a 2D-FTS prevents the difficulty in realizing a single FTS with both fine resolution and broad bandwidth. Instead, it only requires cascading a fine-resolution, narrow-band FTS and a coarse-resolution, broad-band FTS, leading to a greater scalability and a paradigm shift from 1D to 2D (or higher-dimensional) Fourier transform. The 2D-FTS concept can be realized with any existing FTS type. As a proof of concept, the presented design is implemented with a SHS and a tunable FTS; however, the main novelty of this work resides in the effect offered by the combination rather than the separated elements. For the first time, the connection between the cascaded interferometers and high-dimensional Fourier transform is revealed. It is noticed that the reviewer #2 did not mention these contents in his/her comments. We assume that the *prior* response letter may be too lengthy for the reviewer to complete reading. Herein, we summarize the key points in our *prior* responses and revisions:

1. **Power consumption.** This issue has been addressed in the *revised Supplementary information*, Note 10, Section, pages 42-48. It has been demonstrated that, by introducing thermal-isolation trenches, the heating power can be reduced from 2.4 W to < 120 mW, which is applicable in most scenarios (see Refs. R11-R15). The undercut etching of thermal isolation has been demonstrated in many previous studies (see Refs. R16,R17).
2. **Output etendue.** This issue has been addressed in the *revised Supplementary information*, Note 10, Section A (see pages 42-48), and Note 11 (see pages 55-56). A small number of output ports is essential to attain a higher etendue, and thus a higher signal-to-noise ratio (SNR). The presented 2D-FTS supports 1601 wavelength channels with 128 output ports. Thus, compared to a 1601-channel filter, the received

etendue has already been improved by around one order of magnitude. Moreover, the measured peak SNR (> 25 dB) and noise floor ($\approx -35 \sim -40$ dB) are already better than most previous results, as can be found in **Supplementary information**, Table S4, which indicates that the Jacquinot's advantage is only *partially* diminished. In addition, the number of output ports can be reduced from 128 to 1 by replacing the SHS with the well-established digital FTS (see the design example in **Supplementary information**, Figures S22-S25).

3. **Advance in performance.** This issue has been addressed in the *revised Supplementary information*, Note 10, Sections B-C (see pages 48-54), and Note 11 (see pages 55-56). The results shown in this work represent the *largest* channel capacity (> 1601) over all reported integrated FTSs (see Figure 1e). It is also proved that, compared to other spectrometry schemes (e.g., filters and speckle spectrometers), the FTS has the potential to attain a higher level of SNR (see **Supplementary information**, Figure S31). By extending the 2D-FTS to a higher dimension, a finer resolution and a larger capacity can be attained, as discussed in **Supplementary information**, Figures S26-S29.
4. **Principal novelty.** The most significant breakthrough in this research is the transition of the principle from 1D to 2D Fourier transform, which is effectuated by the cascading of multiple FTSs. The concept of 2D-FTS offers a flexible framework that can be applied to enhance the performance of any existing type of FTSs. Such a paradigm shift in FTS design also paves the path towards higher-dimensional FTSs with great scalability (see **Supplementary information**, Figures S30).
5. **Other statements.** All the additional designs are realized by utilizing the building blocks *experimentally* verified in this work, and these components are modeled with measurement results to ensure viability. The only exception is the modelling of thermal isolation, which is a well-established technology that has been widely applied in many reported devices^{R16,R17}.

References:

- R1. Vivien L, *et al.* High speed and high responsivity germanium photodetector integrated in a silicon-on-insulator microwaveguide. *Opt Express* **15**, 9843-9848 (2007).
- R2. Vivien L, *et al.* 42 GHz p.i.n germanium photodetector integrated in a silicon-on-insulator waveguide. *Opt Express* **17**, 6252-6257 (2009).
- R3. IMEC <https://www.imec-int.com/en/silicon-photonics-ICs-prototyping/>.
- R4. AMF <https://www.advmf.com/>.
- R5. Fahrenkopf NM, McDonough C, Leake GL, Su Z, Timurdogan E, Coolbaugh DD. The AIM photonics MPW: A highly accessible cutting edge technology for rapid prototyping of photonic integrated circuits. *IEEE J Sel Top Quant Electron* **25**, 1-6 (2019).
- R6. Siew SY, *et al.* Review of silicon photonics technology and platform development. *J Lightwave Technol* **39**, 4374-4389 (2021).
- R7. Xiao M, Mingyu L, Jian-Jun H. CMOS-compatible integrated spectrometer based on Echelle diffraction grating and MSM photodetector array. *IEEE Photon J* **5**, 6600807-6600807 (2013).
- R8. Zhang Z, Rony KTA, Wang Y, Cheng Z, Tsang HK. Integrated spectrometer with fast wavelength scanning using current injection in PIN diode. In: *2023 Opto-Electronics and Communications Conference (OECC)* (2023).
- R9. Applied Nanotools Inc. <https://www.appliednt.com/>.
- R10. Grotevent MJ, *et al.* Integrated photodetectors for compact Fourier-transform waveguide spectrometers. *Nat Photon* **17**, 59-64 (2023).
- R11. Zheng S, *et al.* A single-chip integrated spectrometer via tunable microring resonator array. *IEEE Photon J* **11**, 1-9 (2019).
- R12. Zhang L, Zhang M, Chen T, Liu D, Hong S, Dai D. Ultrahigh-resolution on-chip spectrometer with silicon photonic resonators. *Opto Electron Adv* **5**, 210100-210100 (2020).
- R13. Sun CL, *et al.* Scalable on-chip microdisk resonator spectrometer. *Laser Photon Rev* **17**, 2200792 (2023).
- R14. Zhang Z, Li Y, Wang Y, Yu Z, Sun X, Tsang HK. Compact high resolution speckle spectrometer by using linear coherent integrated network on silicon nitride platform at 776 nm. *Laser Photon Rev* **15**, 2100039 (2021).
- R15. Kita DM, *et al.* High-performance and scalable on-chip digital Fourier transform spectroscopy. *Nat Commun* **9**, 4405 (2018).
- R16. Zheng SN, *et al.* Microring resonator-assisted Fourier transform spectrometer with enhanced resolution and large bandwidth in single chip solution. *Nat Commun* **10**, 2349 (2019).
- R17. Zeqin L, Murray K, Jayatilleka H, Chrostowski L. Michelson interferometer thermo-optic switch on SOI with a 50- μ W power consumption. *IEEE Photon Technol Lett* **27**, 2319-2322 (2015).

REVIEWERS' COMMENTS

Reviewer #1 (Remarks to the Author):

The authors partially addressed my comments.

In particular, it is hard to distinguish between experimental results and numerical studies due to the extremely matching nature of the presented spectra.

Even if the study is well conducted and the manuscript well written, it is technically very difficult to read and understand due to the approach itself, which requires very specialized readers and thorough understanding of the computational method. Therefore it might be better suited for a more specific readership than this journal.

To Reviewer #1's comment:

Comment 1-1:

The authors partially addressed my comments.

In particular, it is hard to distinguish between experimental results and numerical studies due to the extremely matching nature of the presented spectra.

Even if the study is well conducted and the manuscript well written, it is technically very difficult to read and understand due to the approach itself, which requires very specialized readers and thorough understanding of the computational method. Therefore, it might be better suited for a more specific readership than this journal.

Reply and modifications:

1. The reconstruction results displayed in the revised manuscript are all based on measurement, as clearly specified in the caption of Figure 5 and related descriptions. The numerical results are mainly discussed in **Supplementary information**, so it should be easy for a reader to identify them. Nevertheless, in the revised manuscript, we have added the following sentence to improve clarity:

“The reference spectra were measured using a commercial optical spectrum analyzer (OSA), while the reconstructed spectra are derived from the measured interferograms (see **Supplementary information, Note 9, Figures S19 and S20**).” (caption of Figure 5, lines 518-520)

2. As stated in the *prior* response letter, the computational method is *not* central to this research since the solving of a linear inverse problem has been thoroughly studied and widely applied. The discussion on the numerical processing is detailed only in **Supplementary information**, in response to the Reviewer #1's *prior* request. The novelty of our proposal resides in the use of cascaded interferometers and the resulting establishment of a 2D Fourier space. The primary content of this manuscript is on the device design and result analysis, rather than numerical processing. In the revised manuscript, we have added the following sentences to emphasize the key information in this research and give a clearer guide for a broader readership:

“This article is structured into four sections: the concept of the 2D-FTS, the design of crucial components, the characterization of the device, and the measurement of spectra. We will focus on the concept, mechanism, and realization of the 2D-FTS. The computational details are covered in **Supplementary information**.” (**Introduction**, lines 72-74)